

**Dynamics, predictability, impacts, and climate change considerations of the catastrophic Mediterranean Storm Daniel (2023)**

Emmanouil Flaounas[1,2], Stavros Dafis[3], Silvio Davolio[4,5], Davide Faranda[6, 7, 8], Christian Ferrarin[9], Katharina Hartmuth[1], Assaf Hochman[10], Aristeidis Koutroulis[11], Samira Khodayar[12], Mario Marcello Miglietta[13, 14], Florian Pantillon[15], Platon Patlakas[2,16], Michael Sprenger[1], Iris Thurnherr[1]

1. Institute for Atmospheric and Climate Science, ETH Zurich, Zurich, Switzerland
2. Institute of Oceanography, Hellenic Centre for Marine Research, Athens, Greece
3. National Observatory of Athens, Institute for Environmental Research and Sustainable Development, I. Metaxa &Vas. Pavlou, P. Penteli (Lofos Koufou), 15236 Athens, Greece
4. Department of Earth Sciences "Ardito Desio", University of Milan, Milan, Italy
5. Institute of Atmospheric Sciences and Climate, National Research Council, Bologna, Italy
6. Laboratoire des Sciences du Climat et de l'Environnement, UMR 8212 CEA-CNRS-UVSQ, Université Paris-Saclay & IPSL, CE Saclay l'Orme des Merisiers, 91191 Gif-sur-Yvette, France
7. London Mathematical Laboratory, 8 Margravine Gardens, London W6 8RH, UK
8. LMD/IPSL, ENS, Université PSL, École Polytechnique, Institut Polytechnique de Paris, Sorbonne Université, CNRS, Paris France
9. CNR - National Research Council of Italy, ISMAR - Institute of Marine Sciences, Venice, Italy
10. Fredy and Nadine Herrmann Institute of Earth Sciences, Hebrew University of Jerusalem, Edmond Safra Campus, Jerusalem, Israel
11. School of Chemical and Environmental Engineering, Technical University of Crete, 73100 Chania, Greece
12. Mediterranean Centre for Environmental Studies (CEAM), Charles R. Darwin Street, 14 46980 Paterna, Valencia (Spain)
13. Institute of Atmospheric Sciences and Climate, National Research Council, Padua, Italy
14. Department of Geosciences, University of Bari, Bari, Italy
15. Laboratoire d'Aérologie, Université de Toulouse, CNRS, UPS, IRD, Toulouse, France
16. Department of Physics, National and Kapodistrian University of Athens, Athens, Greece

*Correspondence to*: Emmanouil Flaounas (emmanouil.flaounas@env.ethz.ch)

**Abstract**

In September 2023, storm Daniel formed in the centre of the Mediterranean Sea as an intense Mediterranean cyclone. Its formation was accompanied by significant socioeconomic impacts in Greece including several fatalities and severe damages to agricultural infrastructures. Within a few days, the cyclone evolved into a tropical-like storm, i.e., medicane, that made landfall in Libya, probably marking the most catastrophic and lethal weather event that was ever documented in the region. In this study, we place storm Daniel as the centrepiece of the catastrophic events in Greece and Libya. We thus consider that there is a direct link between the atmospheric processes that turned Daniel into a catastrophic storm and the actual socioeconomic impacts that a single weather system has produced in the two countries. We perform a holistic analysis that articulates between atmospheric dynamics, precipitation extremes, and quantification of impacts, i.e., floods and sea state. This is done by taking into account the predictability of Daniel at weather scales and the attribution of impacts to climate change.

Our results show that Daniel initially formed like any other intense Mediterranean cyclone. At this stage, the cyclone produced significant socioeconomic impacts on Greece, in an area far from the cyclone centre. In later times, Daniel attained tropical-like characteristics while gradually reaching its maximum intensity. Impacts over Libya coincided with the cyclone's landfall at its maturity stage. The predictability of the cyclone formation was rather low even in relatively short lead times -of the order of four days- while higher prediction skill was found when addressing the landfall in Libya for the same lead times. Our analysis of impacts shows the adequate capacity of numerical weather forecasting to capture the extremeness of precipitation amounts and floodings in Greece and Libya.



Therefore, state-of-the-art numerical weather prediction has provided information on the severity of the imminent flood events.

We also analyse the moisture sources contributing to extreme precipitation. Results show that moisture sources were majorly driven by large-scale atmospheric circulation, while in maturity, Daniel drew substantial amounts of water vapor from local maritime areas within the Mediterranean Sea. In a climatological context, Daniel was indeed shown to produce extreme precipitation amounts, and our analysis allows us to interpret Daniel's impacts as an event whose characteristics can be ascribed to human-driven climate change.

## 1. Introduction

In September 2023, a low-pressure system developed within the central Mediterranean Sea, close to Greece. Due to the expected severity of the event, on 4 Sep 2023, the Hellenic National Meteorological Service named the storm 'Daniel.' Within a few days, Daniel evolved into a deep cyclone that propagated southwards, making landfall at the coast of Libya (Fig. 1a). Daniel led to substantial, unprecedented socio-economic impacts in the Central-Eastern Mediterranean from 4 to 11 September 2023, all attributed to the same weather system.

In the cyclogenesis stage, on 5 September 2023, the weather station network of the National Observatory of Athens in Greece (NOAAN; Lagouvardos et al., 2017) measured more than 750 mm of accumulated daily rainfall and up to 1235 mm within four days. The eastern parts of Greece experienced flooding (Fig. 1b) that led to 17 fatalities, the loss of 25% of Greece's annual agricultural production, and the destruction of the local road network. About five days later, on 10 September 2023, the cyclone made landfall near Benghazi, Libya. Consequent flooding caused more than 4,000 fatalities, thousands of missing persons, and overwhelming damages, which were aggravated -among other reasons- by the collapse of city dams. Overall, more than 1150 km$^2$ and 1010 km$^2$ were flooded in Greece (He et al., 2024) and Libya (Qiu et al., 2023), respectively, including the densely populated city of Derna, Libya (Fig. 1c).

Daniel was an intense cyclone, preceded by Rossby wave breaking over the Atlantic and the consequent intrusion of an upper-level trough, as it typically occurs in the Mediterranean basin (Raveh-Rubin and Flaounas, 2017). From the perspective of atmospheric dynamics, upper tropospheric systems are often precursors of Mediterranean cyclogenesis. Indeed, troughs and cut-off lows correspond to stratospheric air intrusions that impose significantly high potential vorticity (PV) anomalies and thus trigger baroclinic instability (Flaounas et al., 2022). While the formation of Mediterranean cyclones is almost entirely dependent on baroclinic instability, the development and intensification of a cyclone into a deep low-pressure system is also a function of diabatic processes. More precisely, latent heat release close to the cyclone centre, mainly due to convection, is a source of positive PV anomalies at low levels, eventually translating into enhanced cyclonic circulation. Therefore, baroclinic instability and latent heat release are cyclone development's main forcings. Both processes are modulating factors of cyclones' intensification from the cyclogenesis stage until maturity, i.e., when the cyclone reaches its minimum pressure at the centre. A complete review of Mediterranean cyclone dynamics is available by Flaounas et al. (2022), while a recent thorough analysis of the dynamics of another intense cyclone in the central-eastern Mediterranean (Ianos, 2020) is provided by Pantillon et al. (2024).

As an environmental hazard, cyclones may produce heavy precipitation from the stage of genesis until their lysis, close to their centres but also in remote areas due to localised convective cells (Raveh-Rubin and Wernli, 2016), warm conveyor belts and frontal structures (Pfahl et al., 2012; Flaounas et al., 2018). Regardless of whether precipitation is stratiform or convective, the large-scale atmospheric circulation is essential for transporting water vapour toward the Mediterranean and thus "feeding" the cyclone-induced precipitation (Hochman et al., 2024). Indeed, the Mediterranean basin is composed of a relatively closed sea surrounded by high mountains. Consequently, Mediterranean cyclones have fewer water sources than their counterparts in the storm tracks over the open oceans. In these regards, large-scale ventilation of water vapour from the Atlantic Ocean and other remote regions towards the Mediterranean has been shown in numerous cases to enhance heavy precipitation, together with local



evaporation due to cyclone-induced high wind speeds (Duffourg and Ducrocq, 2011; Flaounas et al., 2019; Khodayar et al., 2021). Hence, identifying and quantifying the contribution of water sources to heavy precipitation is crucial for understanding socio-economic impacts in the Mediterranean (Hochman et al., 2022a).

In a climatological context, Mediterranean cyclones produce most of the wind and precipitation extremes in the region (Nissen et al., 2010; Flaounas et al., 2015; Hochman et al., 2022b). Therefore, cyclones play a central role in the compoundness of high-impact weather events (Catto and Dowdy, 2021; Rousseau-Rizzi et al., 2023; Portal et al., 2024), also considering that landfalling systems additionally produce storm surges and significant high waves (Patlakas et al., 2021; Ferrarin et al., 2023a; Ferrarin et al., 2023b;). Especially in the case of precipitation, recent results have shown that intense water vapour transport and PV streamers, as a proxy for Rossby-wave breaking, are two of the main features that lead to extreme Mediterranean events (de Vries, 2021; Hochman et al., 2023). Both of these large-scale atmospheric features favour the development of cyclones into deep, low-pressure systems (e.g., Davolio et al., 2020). Thus, their understanding is crucial for predicting socio-economic impacts on weather and climate scales.

Future trends in cyclone-induced hazards in the Mediterranean are mainly quantified through downscaling experiments (e.g., Reale et al., 2022) or statistical-deterministic methods that generate synthetic tracks (e.g., Romero and Emanuel, 2017). Nevertheless, additional investigation is needed to assess the role of climate change in the intensification of storms that occur in the current climate. While attributing extreme events, such as medicanes and high-impact extratropical storms, is a rather difficult task, recent studies based on analogues have suggested that several recent storms are more intense than expected (Faranda et al., 2022, 2023). Further investigation of this critical topic requires a case-to-case approach to take into account the particularities of each storm and to acquire a more holistic understanding of the specific processes that relate to cyclone intensity that are also affected by climate change.

The substantial socio-economic damages of storm Daniel in well-distinct locations call for further investigation into the predictability of the cyclone at different weather timescales and its placement in the context of climate change. In this study, we rely on the underlying processes of cyclone dynamics as the factor directly responsible for the socio-economic impacts of Daniel, and we mainly aim to address the following four questions:
1.      How did cyclone development stages relate to flooding in Greece and Libya?
2.      How reliable and accurate was numerical weather prediction of imminent hazards at different lead times?
3.      Are numerical weather models adequate for the prediction of climate extremes?
4.      Can we attribute Daniel to climate change?

The following section describes the datasets and methods, while section 3 briefly describes the storm dynamics. Section 4 analyses storm Daniel's predictability, and section 5 is devoted to Daniel's attribution to climate change.

**2. Datasets and methods**

**2.1 Datasets**

To analyze the evolution of the cyclone and assess its predictability, we use the operational analysis and the ensemble prediction system (EPS) products of the European Centre for Medium-Range Weather Forecasts (ECMWF). Since the last model upgrade at ECMWF (Cycle 48r1), operational analysis and medium-range ensemble forecast data have been available at a grid spacing of about 9 km. The increase in horizontal resolution and improvements in the data-assimilation system resulted in substantial improvements in skill (ECMWF Newsletter, 176, 2023). The EPS comprises 50 members, initialised with a perturbed analysis and using slightly altered model physics, and one control forecast. This probabilistic forecasting system has been designed to provide a range of possible weather conditions up to 15 days ahead, providing an estimation of predictability. Finally, to



assess Daniel's climatological aspects, we used ERA5 reanalysis (Hersbach et al., 2020) with hourly
atmospheric fields at a 0.25 degrees grid spacing.
We used river discharge data from the Global Flood Awareness System (GloFAS; Grimaldi et al.,
2022) to investigate the hydrological impacts of Daniel across Greece and Libya . GloFAS is an
integral component of the Copernicus Emergency Management Service (CEMS), focusing on
operational flood forecasting globally. It integrates the open-source LISFLOOD hydrological model
with ERA5 meteorological reanalysis data, interpolated to align with GloFAS's resolution (0.05° for
version 4.0), and produced with a daily temporal resolution. This dataset encompasses historical
discharge records crucial in establishing the discharge climatology from 1993 to 2023. We employed
the European Flood Awareness System (EFAS) data to assess the flood forecast potential. The EFAS
system utilises the open-source LISFLOOD hydrological model, calibrated to a refined spatial
resolution of approximately 1.5 km at European latitudes. Forecasts are generated twice per day,
based on initializations at 00 and 12 UTC, and extend lead times from 5 to 15 days to capture a broad
spectrum of potential weather conditions impacting river discharge volumes. These forecasts
incorporate data from the 51 EPS members, the Deutsches Wetter Dienst (DWD) high-resolution
forecasts, and the COSMO Local Ensemble Prediction System (COSMO-LEPS) with 20 ensemble
members, ensuring a comprehensive analysis of the forecast potential.
Finally, to evaluate Daniel's marine and coastal impacts, we analysed the wave results of the
Mediterranean Sea Waves Analysis and Forecast (Korres et al., 2023) available via the Copernicus
Marine Service (CMEMS). We also determined the wave climatology by analysing the Mediterranean
Sea wave reanalysis available via CMEMS (1993-2021; Korres et al., 2021).

**2.2 Methods**


**2.2.1 Object diagnostics**
We identify two-dimensional objects of extreme precipitation to assess the predictability of major
impacts in the EPS forecasts. These objects are defined separately for each member of the EPS as
neighbouring grid points where daily values of precipitation and wind speed exceed the 99th
percentile in the ERA5 climatology (1990-2020). With these objects, we define the probability of the
EPS to forecast extreme weather due to Daniel. Similarly, we define the probability of a cyclone
occurrence in EPS by identifying cyclone masks in each ensemble member as the outermost mean sea
level pressure (MSLP) contour that delimits a surface smaller than that of a circular disc with a radius
of 200 km.

**2.2.2 Air parcel trajectories and moisture source diagnostic**
Ten-day air parcel backward trajectories are calculated from a 30 km horizontal grid every 20 hPa
between 1000 and 300 hPa within boxes over Greece and Libya (as shown in Fig. 1a) using the
LAGRANTO tool (Wernli and Davies, 1997; Sprenger and Wernli, 2015). We calculated two sets of
backward trajectories: (i) the first concerns storm Daniel, where trajectories started every 6 hours on 5
September 2023 and 11 September 2023 from the Greece and Libya box, respectively, using the six-
hourly 3D wind fields from the ECMWF operational analysis data; (ii) the second concerns air parcel
trajectories based on the ERA5 reanalysis wind fields for the 100 most extreme daily precipitation
events in each of the two boxes, starting from the same locations as for the first set of trajectories.
These 100 extreme events were defined as the days with the highest number of grid points within the
Libya or Greece region experiencing daily surface precipitation exceeding the 90th percentile for
autumn in the years 1990 - 2023. Storm Daniel is among the 100 most extreme daily precipitation
events for both regions. We interpolated specific humidity, relative humidity, and the boundary layer
height pressure along all trajectories.
After calculating all the air parcel trajectories, we identified Daniel's moisture sources and those with
the 100 most extreme daily precipitation events using the moisture source diagnostic from Sodemann
et al. (2008). The changes in specific humidity along the trajectory are tracked for all trajectories that
precipitate upon arrival, i.e., showing a decrease in specific humidity during the last step before



arrival. If the specific humidity increases or decreases, a moisture uptake or loss, respectively, is
       recorded. All subsequent moisture uptakes or losses weight a moisture uptake. The identified moisture
uptakes along each trajectory were weighted by the decrease in specific humidity during the last step
       before arrival, and relative moisture uptakes over all trajectories were calculated for each six-hourly
time step. Relative moisture uptakes are then gridded to a 1° global latitude/longitude grid and
       averaged for each day. The relative moisture uptakes are given in $10^{-5}$ % $km^{-2}$, representing each grid
cell's relative contribution per $km^2$ to the precipitation in the target region. Finally, for the 100 most
       extreme events, the daily relative moisture sources are averaged over the 100 most extreme events and
used as a climatological reference for Daniel.

### 2.2.3 Attribution to climate change

We used the methodology developed in the rapid attribution framework Climameter (see Faranda et
       al. (2024) for more details). ClimaMeter offers a dynamic approach to contextualizing and analyzing
weather extremes within a climate context. This framework provides both easily understandable,
       immediate contextualization of extreme weather events and more in-depth technical analysis shortly
after the events. In particular, we analyse here how Mediterranean depressions landfalling in Greece
       and Libya have changed in the present (2001–2022) compared to what they would have looked like if
they had occurred in the past (1979–2000). To do so, we compute analogues of MSLP anomalies of
       Daniel from the MSWX database (Beck et al., 2022) and search for significant differences between
present and past analogues in terms of pressure, near-surface temperature (t2m), precipitation (tp), and
       wind speed (wspd). To account for the seasonal cycle in surface pressure and temperature data, we
remove the average pressure and temperature values for the corresponding calendar days at each grid
       point and each day. This removes the effect of varying surface elevation in space for surface pressure.
Total precipitation and wind-speed data are not preprocessed. If the duration of the event is longer
       than one day, we performed a moving average over the duration of the event on all datasets. We
examined all daily surface pressure data for each period and selected the best 15 analogues, i.e., the
       data minimizing the Euclidean distance to the event itself. The number of 15 corresponds
approximately to the smallest 1‰ Euclidean distances in each subset of our data. We tested the
       extraction of 10 to 20 analogues, without finding qualitatively significant differences in our results.
As customary in attribution studies, the event itself is excluded for the present period. Following
       Faranda et al. (2022), we defined quantities supporting our interpretation of analogue-based
assignments. We can then compare these quantities between the counterfactual and factual periods.
Analogue Quality (Q): Q is the average Euclidean distance of a given day from its 29 closest
       analogues. If the value of Q for the extreme event belongs to the same distribution as its analogues,
then the event is not unprecedented, and attribution can be performed. If the Q value is greater than its
       analogues, the event is unprecedented and, therefore, not attributable.


- Predictability Index (D): Using dynamical systems theory, we can compute the local dimension D of
       each SLP map (Faranda et al., 2017). The local dimension is a proxy for the number of active degrees
of freedom of the field, meaning that the higher D, the less predictable the temporal evolution of the
       SLP maps will be (Faranda et al., 2017). If the dimension D of the event analysed is higher or lower
than its analogues, then the extreme will be less or more predictable than the closest dynamical
       situations identified in the data.


- Persistence Index (Θ): Another quantity derived from dynamical systems theory is the persistence Θ
       of a given configuration (Faranda et al., 2017). Persistence estimates the number of days we will
       likely observe a map that is an analogue of the one under consideration. As with Q and D, we
compute the two values of persistence for the extreme event in the factual and counterfactual world
       and the corresponding distributions of persistence for the analogues.


Finally, to account for the possible influence of low-frequency modes of natural variability in
       explaining the differences between the two periods, we also considered the possible roles of the El
       Niño-Southern Oscillation (ENSO), the Atlantic Multidecadal Oscillation (AMO), and the Pacific
Decadal Oscillation (PDO). We performed this analysis using monthly indices produced by



NOAA/ERSSTv5. Data for ENSO and AMO were retrieved from the Royal Netherlands
Meteorological Institute (KNMI) Climate Explorer. At the same time, the PDO time series was
downloaded from the NOAA National Centers for Environmental Information (NCEI). The
significance of the changes between the distributions of variables during the past and present periods
was evaluated using a two-tailed Cramér-von Mises test at the 0.05 significance level. If the p-value is
smaller than 0.05, the null hypothesis that both samples are from the same distribution is rejected.
Namely, we interpret the distributions as being significantly different. We use this test to determine
the role of natural variability.
**3. Atmospheric processes leading to impacts**
**3.1 Cyclogenesis stage and impacts in Greece**
Before Daniel formed, an omega-blocking pattern and an anticyclonic Rossby wave-breaking
occurred over Europe. Wave breaking resulted in the intrusion of a PV streamer into the central
Mediterranean basin, triggering cyclogenesis in the Ionian Sea on 4 September 2023, which
eventually led to the formation of Daniel within 24 hours (northernmost, first track point in Fig. 1a).
Figure 2a shows that the cyclone on 5 September 2023 was located between Italy and Greece,
developing as a moderate low-pressure system with a minimum MSLP value of about 1004 hPa. The
PV streamer in the upper troposphere wrapped cyclonically around the cyclone centre (green contour
in Fig. 2a), pointing out an ongoing baroclinicity, forcing the cyclone's development. Accordingly, a
high wind speed pattern aligned with the PV streamer's orientation with larger values at the northwest
side of the cyclone (wind barbs in Fig. 2a). This configuration summarises a typical dynamical
structure of Mediterranean cyclones at a stage preceding maturity, i.e., the time of maximum intensity
(Flaounas et al., 2015).
Accumulated precipitation also follows the typical structure of Mediterranean cyclones, with higher
amounts on the northeast side of the cyclone centre (Flaounas et al., 2018). Figure 2a shows that at the
cyclone's initial stages, the highest precipitation accumulation was observed in central Greece
(Dimitriou et al., 2024). The NOAAN surface stations recorded more than 750 mm of daily rainfall
and up to 1,235 mm within four days in eastern parts of the Thessaly region (flooded areas are shown
in cyan colours in Fig. 1b). It is noteworthy that these peak values are underestimated by about 50%
in the ECMWF analysis (purple colours in Fig. 2a).
To quantify the contribution of local and remote areas to such an intense precipitation event in
Greece, Fig. 3a identifies the areas where moisture uptake has been significant for the air parcels that
reached the flooded area of Thessaly (blue square in Fig. 3a). Taking into consideration the largest
moisture uptakes that contribute by at least 50% to the catastrophic precipitation in Greece (second
inner black contour in Fig. 3a that mostly outlines green to red colours), major sources were found in
the Aegean and the Black Seas. This tilted southwest-to-northeast orientation of essential water
sources follows the pathway of strong winds blowing over the Balkans and the eastern Mediterranean
(wind barbs in Fig. 2a), concomitant to the upper-level PV streamer. The intense sea surface fluxes
induced by easterly winds are a precursor feature in common with other cyclones developing in the
same area (e.g., Miglietta et al., 2021). Further moisture (light blue colours in Fig. 3a) mainly
originated from the North Atlantic Ocean. This agrees well with the climatology of moisture sources
of the Mediterranean cyclones that produce the most heavy precipitation events (Flaounas et al.,
2019). The water sources shown in Fig. 3a come partly in contrast to the climatological moisture
sources of extreme precipitation in the same area. Indeed, Fig. 3b shows that the water sources that
typically contribute to extreme precipitation events in the region of Thessaly are mainly located in the
Aegean Sea, extending westwards over the Mediterranean Sea in areas that somewhat overlap with
the primary moisture sources for Daniel precipitation event in Greece (Fig. 3a).
The hydrological impacts of storm Daniel were profound and unprecedented. Figure 4 compares the
peak mean daily river discharge during Daniel with the historical records over three decades. Figure
4a shows the spatial distribution of the maximum peak discharge from January 1993 to August 2023
(i.e., before Daniel), demonstrating typical peak discharge patterns in the Eastern Mediterranean. On

12                                                                                                      6



the other hand, Fig. 4b compares the mean daily peak discharge during September 2023, when Daniel occurred, against the historical peak discharges of the last 30 years in Fig. 4a. Results reveal an
unprecedented magnitude of Daniel's impacts, with several areas experiencing discharges that exceeded the historical maximums by 300 to 500%. The darkest shades in Fig. 4b signify the most
heavily affected regions, where the river discharge during Daniel exceeded previous records by at least a factor of five, highlighting that Daniel was an unprecedented event of increased river discharge
levels (further discussed in section 5). At this cyclone stage, 17 human casualties were registered in Thessaly, along with a profound hydrological aftermath. The extreme rainfall from 3 to 8 September
2023 led to widespread flooding across 1,150 km² in the Thessalian plain, 70% of which constituted agricultural land. The inundation severely affected the cotton crops, with floodwaters covering more
than 282 km², roughly 30% of the region's total cotton fields. Over 35,000 farm animals were also affected (He et al., 2023).

**3.2 Mature stage and impacts in Libya**
Severe weather events gradually faded in Greece during the night of 6 September 2023 while the surface cyclone moved southwards in phase with the upper-tropospheric low. In the following three
days, Daniel lingered over the central Mediterranean Sea (circular part of the track in Fig. 1a), with minimum pressure remaining almost constant close to 1004 hPa (Fig 1a). During this period, the sea
surface temperature (SST) in the central Mediterranean has been anomalously high by roughly 2 K respect to the average September SST of the period 1982-2011 (Fig. 5a). The role of anomalously
high SSTs in intensifying cyclones has been previously shown in several studies based on numerical sensitivity experiments (Miglietta et al., 2011; Romaniello et al., 2015; Messmer et al., 2017;
Pytharoulis, 2018). In the case of Daniel, deep moist convection was favoured, as suggested by the great extent of the areas covered by cold cloud-tops and intense lightning activity close to the cyclone
centre (not shown). Afterward, on September 8, the cyclone started showing tropical-like features, like deep warm core, spiral cloud bands, and a maximum wind speed in the low levels a few tens of
km from the centre. Thus, the cyclone satisfies the phenomenological definition of a medicane recently proposed (Miglietta et al., 2025). Deep convection contributed to the rapid deepening of the
cyclone, reaching a minimum MSLP of 997 hPa on 9 September 2023. After that, Daniel made landfall at the northeastern coasts of Libya during the night hours of 9 September 2023. Comparison
of Figs. 2a and 2b shows that at the time of maturity, the upper-level PV streamer at 300 hPa was weaker than during cyclogenesis. At the same time, Daniel has developed a significantly stronger
MSLP gradient, leading to wind speeds that reached up to 40 kts (about 20 m s$^{-1}$). While weaker than earlier, the wrap-up of the upper-level PV streamer around the cyclone centre was proposed to be
responsible for its intensification just before the cyclone made landfall (Hewson et al., 2024). This reflects, on the one hand, the anomalous characteristics of this medicane (medicanes generally
intensify over the sea and weaken inland), on the other hand, the critical role of upper-level features for the evolution of Mediterranean cyclones.

The intense winds associated with the storm generated a severely disturbed sea in the Central
Mediterranean basin, with south-westerly propagating waves extending from the Aegean Sea to Libya following the strong winds pathway (Fig. 2b). Indeed, the analysis of the wave data from the
Mediterranean Sea Waves Analysis and Forecast evinces waves with significant height of about 5 m in the Gulf of Sirte and the northern Aegean Sea (Fig. 6a). Such values exceed the 99th percentile in
the Mediterranean Sea wave reanalysis. A peculiar aspect of Daniel is that strong winds blew in the Central Mediterranean Sea for many days. As a result, Daniel preserved a severe sea state over
northern Greece, in the Central Mediterranean basin, and along the Libyan coast. To evaluate the cumulative impact of the event, we computed the total storm wave energy (TSWE; Arena et al., 2015)
by integrating the wave power contribution of each sea state over the storm duration (Fig. 6b). TSWE reaches peak values of about 3000 kWh/m in the Gulf of Sirte, which is above the 99th percentile of
the total storm wave energy obtained from the Mediterranean Sea wave reanalysis. Such an energetic sea condition and the storm surge affected much of Libya's eastern coastal zones, causing coastal
flooding, erosion, and infrastructure damage (World Bank, 2023). However, due to a lack of detailed information about coastal damages, it is impossible to evaluate the relative socioeconomic impact of
each single threat (storm surge, waves, rain, river flood) driven by storm Daniel.



During the cyclone's mature stage, Bayda experienced about 414.1 mm of rainfall within less than 24 hours, equaling 80% of the city's mean annual accumulated precipitation and a new daily
precipitation record (Weather Meteorological Organisation, 2023). Figure 3c shows significant moisture sources (red colours in Fig. 3c) to encompass the cyclone centre. This suggests that the
cyclone-induced circulation played an essential role in moistening the atmosphere within the proximity of the cyclone. Nevertheless, the moisture sources that contribute by at least 75% to the
precipitation event in Libya (black contour encompassing green to red colours in Fig. 3c) still retain a southwest-to-northeast orientation as in Fig. 3a (i.e., during the precipitation event in Greece).
Comparing the moisture sources among the two precipitation events in Greece and Libya, it seems that in the latter case, the cyclone tends to attract more moisture from its surrounding area. In contrast,
in both cases, northern moisture sources tend to align with the large-scale circulation responsible for downstream cyclogenesis in the Mediterranean. This southwest-northeast orientation of moisture
sources contrasts with the climatological sources in Figs. 3b and 3d that mainly highlight the importance of local sources, especially from the Mediterranean Sea westwards of the two study
regions. Eventually, after landfall, Daniel dissipated fast over the Sahara Desert when it reached Egypt on 11 September 2023.

Daniel resulted in severe flash floods in northern Libya, with river discharges exceeding by 500% the
peak values of the last three decades in the region (Fig. 4b). As a result, northeastern Libya's population of 884,000 people has been affected directly in five provinces by the collapse of two dams.
About 30% of the city of Derna was flooded, and almost 900 buildings were destroyed, including damages to roads and other infrastructure in the area (OCHA 2023, UNICEF 2023). According to the
DTM update (IOM 2023), over 5,000 people were presumed dead, 3,922 deaths were registered in hospitals, 10,000 people were declared missing by the Libyan government and Red Crescent Society
while at least 30,000 people were recognized as internally displaced (UNICEF 2023, IOM 2023) in the Derna area. Extensive damage was shown to critical infrastructure such as hospitals and drinking
water supply systems. Many roads were rendered impassable, making it difficult for humanitarian aid and supplies to get through. At least $10 million budget was allocated from the UN Central
Emergency Response Fund to scale up intervention in response to the Libya disaster, and almost 72 million were requested to cope with the most urgent needs of around 250,000 people (OCHA 2023)
just for the first three months after the flooding.
**4. Weather forecasting of Daniel and implications to impacts**
Daniel's impacts took place in two distinct periods: during cyclogenesis and at maturity. In the former
stage, most precipitation was produced in areas remote to the cyclone centre, drawing moisture from the broader surrounding area. At the later stage, the cyclone impacts were relevant around landfall,
and precipitation and sea level rise were important close to the cyclone centre. Therefore, the two distinct stages of Daniel that provoked substantial impacts in Greece and Libya were linked to cyclone
stages of different dynamics, which also have different implications in Daniel's numerical prediction. In the case of Greece, i.e., at the initial stage of Daniel, it is the timely prediction of cyclogenesis that
would primarily provide useful information to civil protection, whereas, in the case of Libya, it is the accurate prediction of the cyclone track, intensification, and its landfall location. This section focuses
on the predictability of the environmental hazards linked to Daniel's socio-economic impacts, i.e., precipitation amounts, sea state, and cyclone track.

**4.1 Forecasting cyclogenesis stage**
Concerning the cyclogenesis stage, a forecast model has to predict the formation of the cyclone in order to provide valuable information regarding its impact. This suggests that numerical weather
prediction should accurately reproduce the large-scale atmospheric circulation, the Rossby wave breaking, and the consequent intrusion of the PV streamer within the Mediterranean, as shown in Fig.
2a by the green contour. At a lead time of 96 hours, Fig. 7a shows high uncertainty among the EPS members on the location of the PV streamer intrusion. Indeed, the average PV of all EPS members of
ECMWF at 300 hPa (outlined by blue contours in Fig. 7) depicts a much larger area of high PV values than the one in Fig. 2a. This is due to the limited agreement on the occurrence -or colocation-





of the intrusion of the PV streamer among the EPS members, of the order of 25 to 50%, as suggested by the blue crosses in Fig. 7a.

Following the uncertainty in the PV streamer occurrence, the MSLP spread is also high in Fig. 7a with
no clear local minimum in the average values (black contour in Fig. 7a). At subsequent lead times, the spread of MSLP decreases (e.g., Figs 7c and 7e) until it becomes negligible 24 hours before
cyclogenesis (Fig. 7g). At such short lead times, the cyclone formation was forecasted with confidence to occur in the Ionian Sea, to the southwest of Greece (black contours in Fig. 7g).
Confident forecasts of cyclogenesis should go hand in hand with higher agreement among the EPS members on the location of the PV streamer. Indeed, 24 hours before cyclogenesis, more than 95% of
EPS members agreed on the area of PV streamer intrusion. In contrast, average values (blue contour in Fig. 7g) better match the ones in the ECMWF analysis (green contour in Fig. 2a). The similar
behaviour in the spread of MSLP and PV streamer relies on the direct relationship between the Rossby wave breaking over the Atlantic Ocean and the accurate prediction of Mediterranean
cyclogenesis. This has been highlighted by Chaboureau et al. (2012) and, more recently, by Portmann et al. (2020) and Sherrmann et al. (2023). It has also been discussed in a review paper by Flaounas et
al. (2022).
To get deeper insights into the representation of cyclogenesis among the EPS members, Fig. 8 shows the level of agreement on the cyclone objects (as presented in section 2.2.1). At lead times of 96 hours
(Fig. 8a), cyclone centres are scattered across the central Mediterranean while two members of the EPS do not even predict cyclogenesis. Higher overlapping of cyclone objects among the EPS
members (green shading in Fig. 8) is indeed within the limits of the observed cyclone object as in the ECMWF analysis (black contour in Fig. 8a). In fact, about 30% of the different EPS members
produce overlapping cyclone objects. At forecast lead times of three days, the overlapping of cyclone objects increases abruptly (comparing green shaded areas between Figs 8a and 8c), suggesting a much
higher agreement among the EPS members on the cyclone occurrence within the correct location. The high agreement is retained also for shorter lead times of two and one days (Figs 8e and 8g). A similar
"jump" in the predictability of cyclone occurrence has been shown for several medicane cases by Di Muzio et al. (2019). Most probably, this "jump" is due to the dependence of Mediterranean
cyclogenesis on the preceding Rossby wave breaking and, consequently, on the credible inclusion of this event within the forecast initial conditions.

**4.2 Forecasting cyclone location and intensity at the mature stage**
Figure 1a shows that on 10 September, Daniel was at its mature stage and made landfall on the coasts of Libya. For all different forecast lead times of this event in Fig. 7, the spread of MSLP consistently
retains high values close to the landfalling area (right column of panels in Fig. 7). This is directly relevant to the high MSLP gradients close to the cyclone centre (Fig. 2b) where negligible
displacement of cyclone centres may result in a relatively large standard deviation of MSLP in the EPS. Indeed, Figs. 8b and 8d point to the high certainty of the cyclone occurrence in the EPS, where
most members produce consistent and overlapping cyclone objects (depicted by dark green shading in Figs. 8b and 8d). Such performance comes in contrast to forecasting the stage of cyclogenesis, where
MSLP spread do not have a clear pattern in the left panels of Fig. 7 (green and yellow areas), and cyclone objects present limited overlapping for the same lead times (e.g., comparing Figs 8a and 8b).
The limited agreement among the EPS members on the PV streamer intrusion leads to considerable differences among the EPS members on the location or even the occurrence of Daniel. In contrast, the
predictability of landfall in Libya seems more consistent among the EPS members of ECMWF.
Considering forecast lead times of 72 to 96 hours (i.e., initialization on 6 or 7 September), the cyclone has already formed and was located over the central Mediterranean (spiral part of the track). It is in
the middle of its lifespan and increasing in intensity (MSLP depicted by dot sizes in Fig. 1a). Therefore, the cyclone has been already inscribed in the model's initial conditions. Still, from the
perspective of impacts, the location of landfall and the cyclone's intensity are crucial. Figure 10a shows that even for early lead times of six days (initial conditions of 4 September 2023, 0000 UTC),
the cyclone tracks from all EPS members make landfall on the Libyan coasts. The spread of the tracks



is wide enough to include the actual cyclone track (in blue colour in Fig. 10a); therefore, the forecast may lead to a reliable and timely warning of potential impacts.

Nevertheless, Fig. 10b shows that almost all the EPS members underestimated the cyclone's intensity by forecasting too high MSLP values on 10 September. The intensity of the cyclone is dependent on the baroclinic and diabatic forcing of its development (Flaounas et al., 2021). Therefore, the performance of all EPS members depends on the accurate representation of the parametrized processes, mainly convection close to the cyclone centre and surface fluxes, and the morphology of the PV streamer intrusion. For the present case, Hewson et al. (2024) noted that, while in the development stage, the latent heat released from convection, favoured by the high SST and intense sea surface fluxes, balanced out the tendency for frictional decay, in the last stage a marked upper-level low moving from the west was responsible for a further deepening.

**5. Daniel's impacts in a climatological context**

**5.1 Forecasting climate extremes**
The previous sections focused on the capacity of the EPS to forecast Daniel cyclogenesis as the primary driver of impacts. In this section, we extend this analysis by focusing on the predictability of impacts in a climatological context, namely extreme precipitation and consequent floods. We used the ERA5 reanalysis to diagnose extremes since this product offers a reliable and consistent representation of present-day climate (Hersbach et al., 2020). In this respect, Fig. 9 shows the area affected by extreme daily precipitation on September 5 (in red contour, explained in Section 2.2.1). In addition, Fig. 9 shows the percentage of the EPS members that forecast daily precipitation exceeding the climatological threshold of extremes (in blue shading). At a lead time of 96 hours (Fig. 9a), less than half of the ensemble members predicted the climatological extreme precipitation amounts within the area delimited by the climatological values of ERA5 (red contours). Nevertheless, the area formed by the blue shading in Fig. 9b is consistent with the climatological extremes. Consequently, the members of the EPS that produce extreme precipitation could provide information four days before issuing a warning on the potential occurrence of high-impact weather.

Interestingly, the overlap of extreme precipitation objects among the EPS members might reach up to about 85% in the area of Thessaly in Greece for a lead time of even 96 hours (Fig. 9a). This percentage exceeds by 52% the maximum percentage of overlap between the cyclone objects (Fig. 8a). This suggests that the EPS members have been more consistent in the production of extreme precipitation even if cyclone centres presented a comparably greater spread. For subsequent lead times, the predictability of extreme precipitation strongly increases, showing a high probability for a lead time of even 72 hours. Indeed, almost all members predict extreme precipitation off the coast and in the northeastern part of Greece within the eventually flooded area of Thessaly.

At the time of the landfall of Daniel on the Libyan coasts, the EPS showed a higher predictability, with cyclone objects and associated extreme precipitation being predicted at least four days in advance by several EPS members (Figs. 9b), albeit the location of both cyclone and precipitation objects are still shifted to the west compared to the analysis (Figs 8b and 9b). The probability strongly increased at shorter lead times (Figs 9d, 9f and 9h). This shift is plausibly relevant to the change in Daniel's tracks in EPS members to the west side of the observed one (Fig. 10a), and it is pretty corrected when reaching a lead time of two days with a limited bias in the location of the cyclone centre towards the southwest.

The potential of extreme precipitation leading to substantial socio-economic impacts has also been transferred to hydrologic discharge forecasts. The hydrographs presented in Fig. 11 examine river discharge predictability as forecast by the operational European Flood Awareness System (EFAS) during Daniel. For the Pinios River outlet in Thessaly, the forecast initiated on 1 September underpredicted the peak discharge on 5 September. Nevertheless, extreme discharges were evident for several members five days in advance. The forecast accuracy improved closer to the event, with ensemble members (grey lines) converging towards the peak discharge (perfect forecast - red line).



This trend indicates an increasing reliability of the forecast as the lead time decreases, particularly
within 48 hours of the event. Concerning the predictability of floods at the Wandi Derna River outlet
(Fig. 11, right column), a similar pattern was observed as for the Pinios River outlet. Initial forecasts
are widespread among ensemble forecast members, reflecting high uncertainty. Nevertheless, as the
lead time decreases, the ensemble forecast for 10 September aligns more closely with the actual
discharge.

Figure 4 provides a crucial context by comparing the peak mean daily river discharge during Daniel
with historical records over three decades. The unprecedented magnitude of the event, as shown in
Fig. 4b, underscores Daniel's severity, which is especially evident in the darkest shaded regions where
discharges were at least fivefold higher than in historical records. The ability of EFAS to predict such
extreme events, as illustrated in Fig. 11, highlights its value in anticipating extreme events. Accurately
predicting these unprecedented discharges, especially within a short lead time, suggests that
operational forecast systems like EFAS can capture these events' extremities. The Copernicus
Emergency Management Service (CESM), supported by EFAS and GloFAS, effectively predicts both
the timing and magnitude of extreme hydrologic events, offering vital information that could enhance
preparedness and response strategies in the face of escalating climate extremes. This capability is
crucial for civil protection and mitigating such disasters' socio-economic impacts.

### 5.2 Attribution to climate change

In this section, we discuss the attributability of Daniel and its hazards to climate change, using the
approach used in ClimaMeter (see www.climameter.org and  Faranda et al., 2024). The workflow
used in ClimaMeter consists of looking for weather conditions similar to those that caused the
extreme event of interest . The search of similar past events is based on defining analogues of the
identified surface pressure patterns over the chosen spatio-temporal domain. We split the dataset
1979-Present in two parts of equal length and consider the first half of the satellite era  as "past" and
the second part as "present" separately. We use data from MSWX. We consider the first period as
representative of a past world with a weaker anthropogenic influence on climate than the second
period, which represents the present world affected by anthropogenic climate change.  The analogues
search is only performed on surface pressure data. Results reported for temperature, precipitation and
wind-speed data are always associated with surface pressure analogues. For the landfall in Greece, we
search analogues for 5 September 2023 within the region defined within the domain shown in Fig. 12a
and within the extended autumn season, from September to December. Results are reported in Fig. 12.
Figure 12a-d shows that depressions similar to Daniel landfalling over Greece have about the same
pressure minima in the present than they had in the past . Figure 12e-h shows that temperature during
depressions have increased on the Ionian sea of about 2°C and decreased over Anatolia. Precipitation
analysis (Fig. 12j-l) show that similar events produce heavier precipitation over the Ionian sea and
Albania but  generally lower precipitation over continental Greece and the Peloponnese (between 4
and 12 mm/day). In order to discuss changes in the possible dynamical properties of the event, the
metrics Q, D, and Θ (Figs 12q-s) are computed.  Q is defined as the mean Euclidean distance from the
event to its best analogues, D is a metric of predictability and Θ study the persistence all the cyclones
(see Faranda et al. (2022) for a detailed description of the metric. Figs 12q-s show no significant
changes between the two periods (present and past climate).  We can however infer how distant the
event is from its analogues using the metric Q further highlights that the event has  similar analogues
in past and present periods. Significance of the changes between the distributions of variables during
the past and present periods is evaluated using a two-tailed Cramér-von Mises test at the 0.05
significance level. If the p-value is smaller than 0.05, the null hypothesis that both samples are from
the same distribution is rejected, namely we interpret the distributions as being significantly different
We also find that similar events have become more frequent in December, while they previously
occurred chiefly in October, (Fig. 12t). In order to evaluate the possible role of low-frequency modes
of natural variability in explaining the differences between the composite maps of analogues in the
two periods, we also include in our analysis monthly indices of the El Niño–Southern Oscillation
(ENSO), the Atlantic Multidecadal Oscillation (AMO), and the Pacific Decadal Oscillation (PDO).
We compare the distributions of the ENSO, AMO, and PDO values on the dates of the analogues in
the past and present periods, and we test the statistical significance of the observed differences. For



this case, we find that the AMO and PDO sources of natural climate variability may have influenced the event (Figs 12u-w). This suggests that the changes we see in the event compared to the past may be due to a combination of human-driven climate change and natural variability. Figure 12x shows an increasing trend in the frequency of these events when 30 analogues are searched for the entire period analysed.

Regarding landfall in Libya, we searched for analogues for 10 September 2023 within the region depicted by Fig. 13a, and we searched for analogues for the extended autumn season (SOND). Results are reported in Fig. 13. The MSLP changes (Fig. 13d) show no substantial differences in the areas that have been affected significantly. Precipitation changes (Fig. 13l) show that similar events produced larger amounts of precipitation in the eastern Libyan coast (between 5 and 9 mm/day), severely affected by Daniel's severe precipitation on 10 September 2023. The metrics Q, D, and Θ (Figs 13q-s) show no significant changes between the two periods. Q again highlights that the event does not have suitable analogues in past and present periods. Events have become less frequent in September and November and slightly more common in December (Fig. 13t). Finally, we find that sources of large-scale natural climate variability, namely the AMO and the PDO may have influenced the event (Figs 13u-w). Figure 13x shows changes in the frequency of these events when 30 analogues were searched for the entire period analysed. As in the case of impacts in Greece, a significant increasing trend in frequency is found.

Based on the analyses above, we conclude that Mediterranean depressions like Daniel hitting Greece and Libya show lower MSLP and higher precipitation in the present climate than in the past. We thus interpret Daniel as an event whose characteristics can be ascribed to human-driven climate change. Although not included in our analysis, we hypothesise that the changes we see in precipitation amounts compared to the past may be partially due to human-driven climate change, in keeping with the potential for heavier precipitation in a warmer climate.

## 6. Summary and conclusions

In the last decade, more than 410,000 deaths have been attributed to weather-related disasters, mostly in low-income countries where heatwaves and intense precipitation events are the leading causes of death. Aside from human casualties, 1.7 billion people have been affected in the 2010-2020 decade by these kinds of phenomena. The IFRC World Disasters Report (2020) concluded that the climate acts as a risk multiplier, especially in the case of low-income countries or even at a regional level inside countries. A glaring example of the impact of such disasters has been highlighted by the recent floods in the Mediterranean, especially in Greece and Libya, following the Mediterranean cyclone Daniel.

This study aimed to comprehensively analyse medicane Daniel, which links atmospheric dynamics, predictability, and impacts. Beyond the description of the underlying physical processes, atmospheric dynamics are used here to understand the performance of numerical weather prediction. Impacts, in terms of floods and sea state (for Libya), have also been analysed with respect to numerical weather prediction. All of our analysis has been framed by the climatological context of both cyclone-induced precipitation and relevant impacts that link with climate change attribution of both catastrophic events.

From the perspective of atmospheric dynamics, the processes governing Daniel were similar to those identified for other intense Mediterranean cyclones: cyclogenesis was triggered by the intrusion of an upper-level PV streamer in the Ionian Sea, and thereafter, the cyclone developed into a deep storm, that propagated southwards and towards Libya while it was turning into a well defined mesoscale tropical-like cyclonic system. In terms of impacts, we identified two well-distinct stages: the first is relevant to cyclogenesis, where Daniel was newly formed and affected Greece with severe floods (on 5 September, 2023). In the second stage, Daniel reached maturity while making landfall in Libya, where it inflicted severe socio-economic impacts on 10 September 2023 due to floods (about 5 days after the floods in Greece). In both stages, Daniel produced extreme precipitation amounts, driving a moisture flow towards the areas that experienced the floods. The moisture transport was aligned with the PV streamer and, in particular, the large-scale atmospheric circulation. Large amounts of moisture



that contributed to the catastrophic precipitation in the flooded areas were drawn locally from the Mediterranean Sea, which has been anomalously warm, but also from the continental areas of central
and eastern Europe.
In Greece, the floods occurred in an area that was rather remote from the cyclone centre. On the other hand, floods in Libya occurred close to the cyclone centre. Therefore, the implication of different
cyclone dynamics is important in the prediction of socio-economic impacts at both weather and climate scales. During its first stage over Greece, the predictability of the cyclone formation was
rather poor for lead times of more than four days. In fact, it was a rather challenging issue for the ECMWF EPS to forecast precisely the intrusion of the PV streamer in the Mediterranean. This result
aligns with previous studies that showed rather poor performance in predicting medicane occurrences for lead times of the order of four to five days (Di Muzio et al., 2019). For shorter lead times, the
ECMWF EPS could forecast cyclogenesis, and thus the flooding event, with higher certainty. During its second stage (impacts in Libya), the cyclone intensified quickly, transitioned into a medicane, and
made landfall in Libya within a few days. The predictability of the medicane track -and therefore its landfall- showed higher certainty for lead times of four days. This suggests that numerical weather
prediction is more prone to an erroneous predictability of the stage of cyclogenesis, i.e., after the cyclone formed it is more likely for a forecast model to correctly predict its location in subsequent
times.
Precipitation amounts were found to correspond to climate extremes in both Greece and Libya. In both cases, floods were responsible for unprecedented river discharges that largely exceeded the
climatological maxima of the last 20 years. The numerical weather prediction model was able to forecast these climate extremes (even if thresholds were defined by reanalysis and not by the same
forecast model). This suggests the exceptional potential for information to the public about the severity of imminent high-impact weather. Indeed, framing weather forecasts into a climatological
context, e.g. providing a return period of a precipitation event, would provide the means to the local population for an empirical assessment of the severity of an imminent high-impact weather event. In
this context, we have analysed Daniel with respect to climate change, and we have provided the grounds to interpret Daniel as an event whose characteristics can be ascribed to human-driven climate
change. In these regards, we have performed an analysis based on analogues and indeed several of them were found during winter. The anomalous occurrence of such a storm in September, a warmer
month for SST, could be a reason for enhancing its destructiveness through enhanced precipitation.
In the scientific literature, weather events are typically analysed as case studies with specific objectives that rarely escape the narrow scope of a single scientific discipline. Here, we used Daniel, a
high-impact weather event, as a centrepiece of different approaches in order to provide a deeper understanding of socioeconomic impacts through the prism of both weather and climate scales. We
find such an approach valuable for the bridging of different scientific communities and eventually important for the communication of hazards to the local population. We envisage the use of this
interdisciplinary approach for other weather extremes and regions.
**Author contribution**
All authors provided text and comments and substantially contributed to the final form of the
manuscript. In particular, EF conceptualized the study, organized contributions, wrote parts of the paper and performed editing. SD, KH, and PP contributed to the predictability perspectives of the
study with diagnostic results and writing. AK, CF and SK contributed to the impacts and hazards perspective with diagnostic results and writing. DF contributed to climate change attribution
perspective with diagnostic results and writing. IT and MS both contributed with diagnostic results and writing to moisture sources perspective. Finally, MMM, FP and AH contributed to the manuscript
with writing, commenting and reviewing contributions from all coauthors.
**Competing interests**
Silvio Davolio is a member of the editorial board of Weather and Climate Dynamics.



**Acknowledgements**

This article is based upon work from COST Action CA19109 "MedCyclones", supported by COST – European Cooperation in Science and Technology (http://www.cost.eu, last access: 20 July 2024) and
from project "Earth Observations as a cornerstone to the understanding and prediction of tropical like cyclone risk in the Mediterranean (MEDICANES)", ESA Contract No. 4000144111/23/I-KE.
Georgios Kyros from the National Observatory of Athens/meteo.gr is acknowledged for helping collect the Copernicus Sentinel-2 data in Figure 1. The Israel Science Foundation (grant \#978/23),
provides funding for AH's contribution.

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


















**Figures**





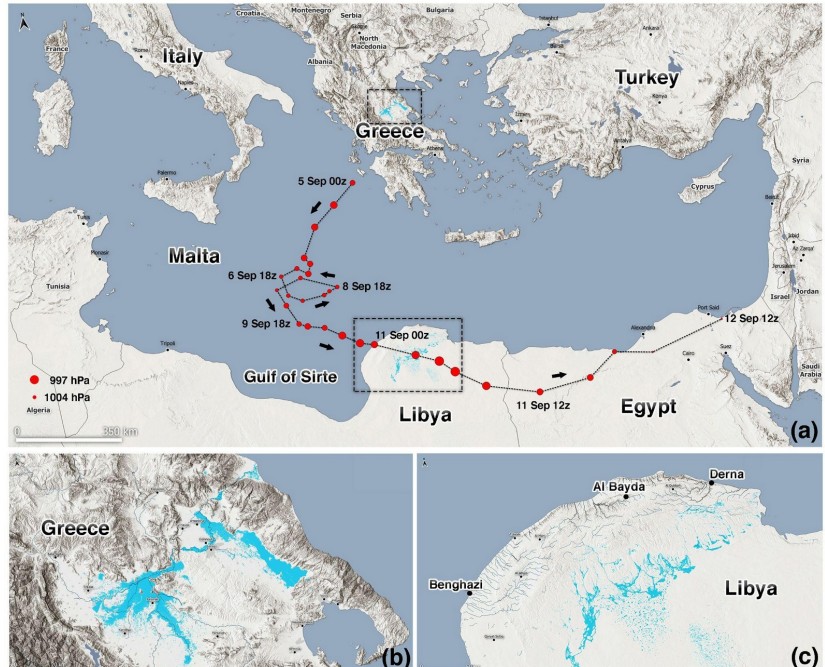


**Figure 1 (a)** Track of Daniel storm where the size of red dots is proportional to cyclone depth in
terms of minimum mean sea level pressure. Flooded areas are shown in cyan and blue tones (acquired
by one of the Copernicus Sentinel-2 satellites on 2 and 12 September). Panels **(b)** and **(c)** zoom over
central Greece and Libya (square boxes in panel **a**).

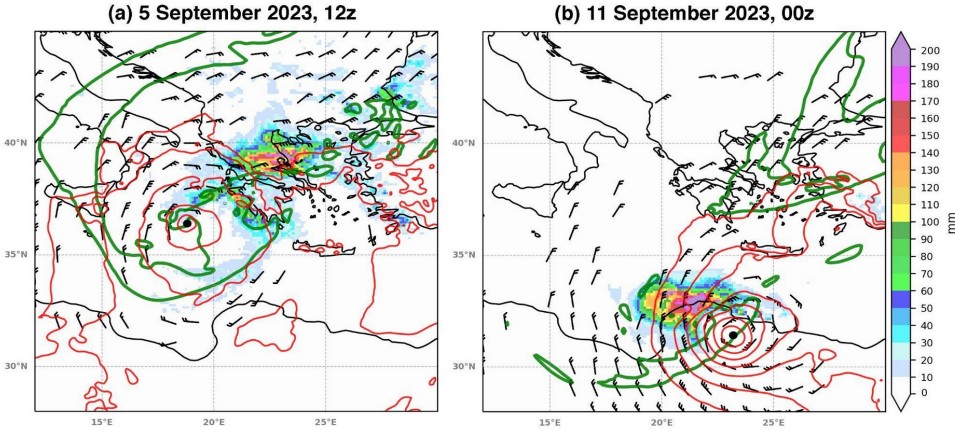


**Figure 2 (a)** Potential Vorticity of 2 PVU at 300 hPa (in green contour) and wind speed higher than
15 knots at 850 hPa (in barbs with full and half bars depicting 10 and 5 knots, respectively) and mean
sea level pressure (in red contours for values lower than 1012 hPa with 2 hPa interval) on September
5, 2023, at 12UTC. 24-hour total accumulated precipitation from 5 to 6 of September 00 UTC is
shown in shading. **(b)** Same as (**a**) but for September 11, 2023, at 00 UTC. In both panels, the black
dot indicates the minimum sea level pressure position.



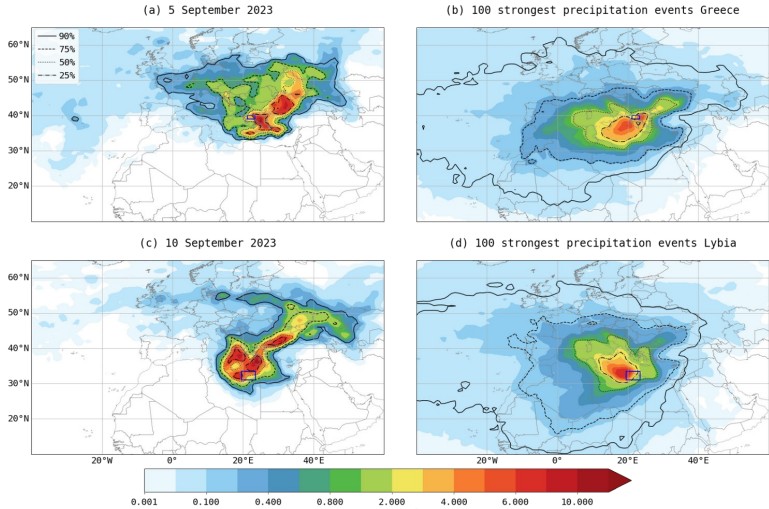


**Figure 3 (a)** Accumulated moisture sources that contribute to the precipitation event in Thessaly
(depicted by the blue rectangle) on 5 September 2023. The black contours outline 90%, 75%, 50%,
and 25% of the total moisture uptake. **(b)** as in **(a)** but for the 100 most extreme daily precipitation
events from 1990 to 2023. **(c)** as in **(a)** but for the precipitation in the study region in Libya (blue
rectangle) on 10 September 2023. **(d)** as in **(c)** but for Libya.

1040



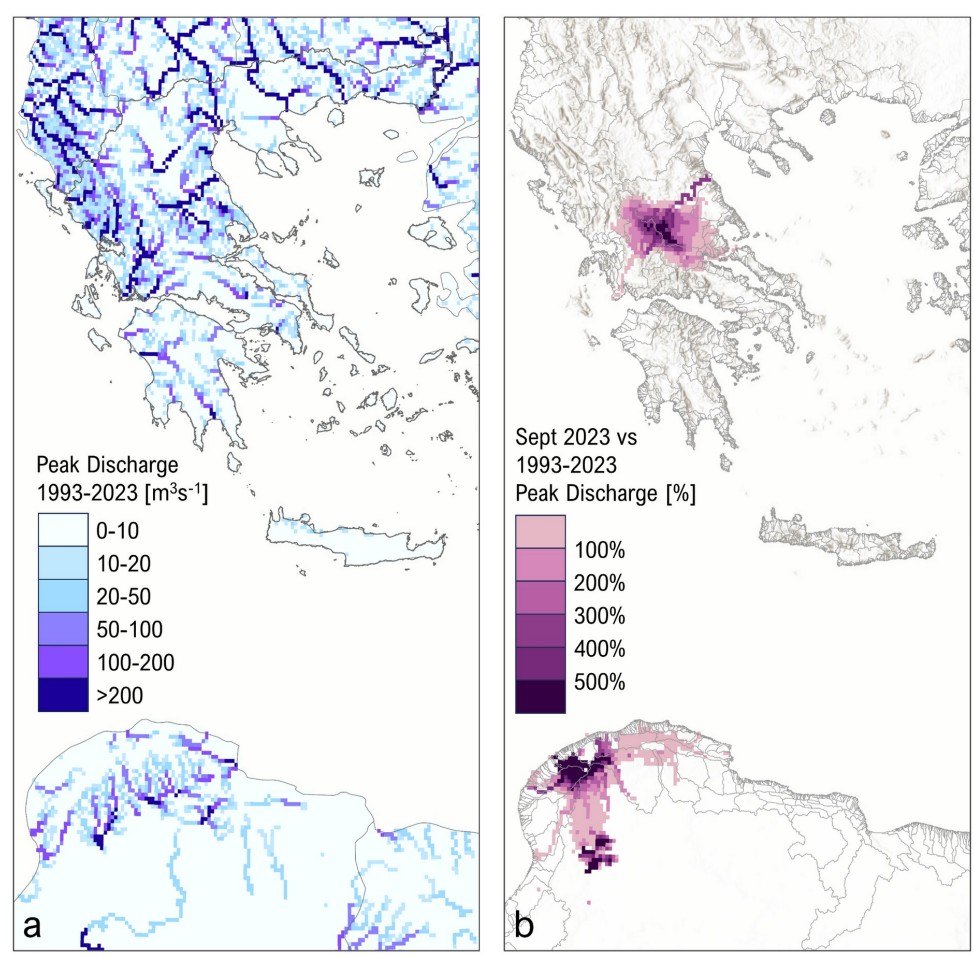

**Figure 4** Peak discharge over three recent decades (1993-2023) versus Daniel storm as represented by the Global Flood Awareness System (**a**) spatial distribution of the maximum peak river discharge from January 1993 to August 2023, (**b**) comparison map for September 2023 illustrating the peak discharge as a percentage increase over the maximum peak discharge during the 30 years from 1993 to August 2023 in **(a)**.



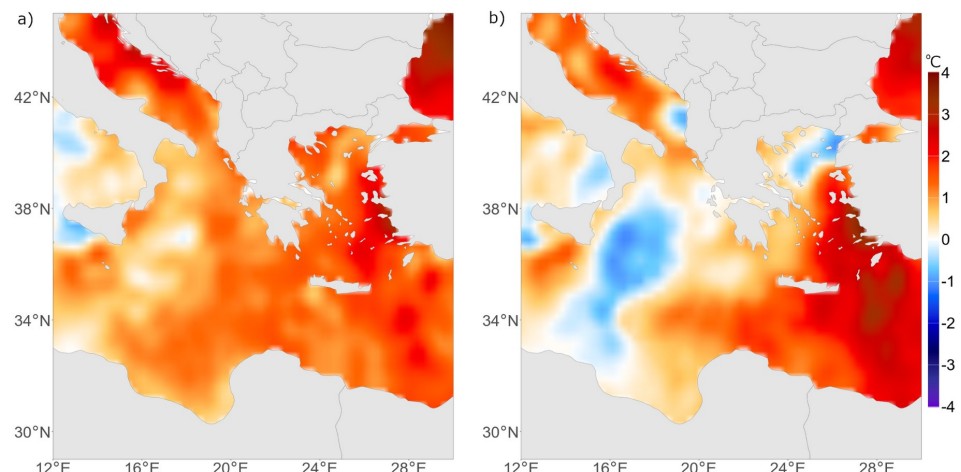

**Figure 5 (a)** Daily SST anomaly for 3 September 2023, and (b) 9 September 2023. The reference climatology for anomaly determination is 1982-2011.

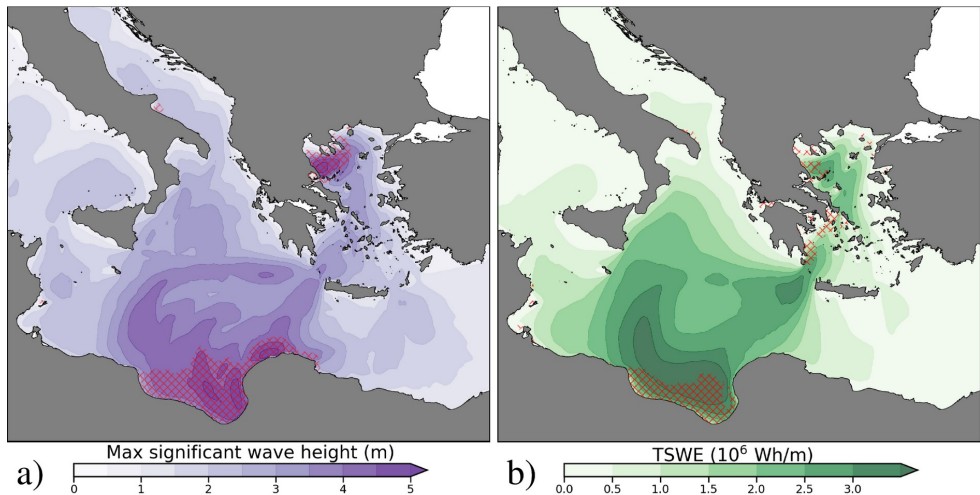

**Figure 6 (a)** Maximum significant wave height. **(b)** Total wave energy of the storm. Red patches mark areas of extreme conditions (above the 99th percentile) determined based on the Mediterranean Sea wave reanalysis.

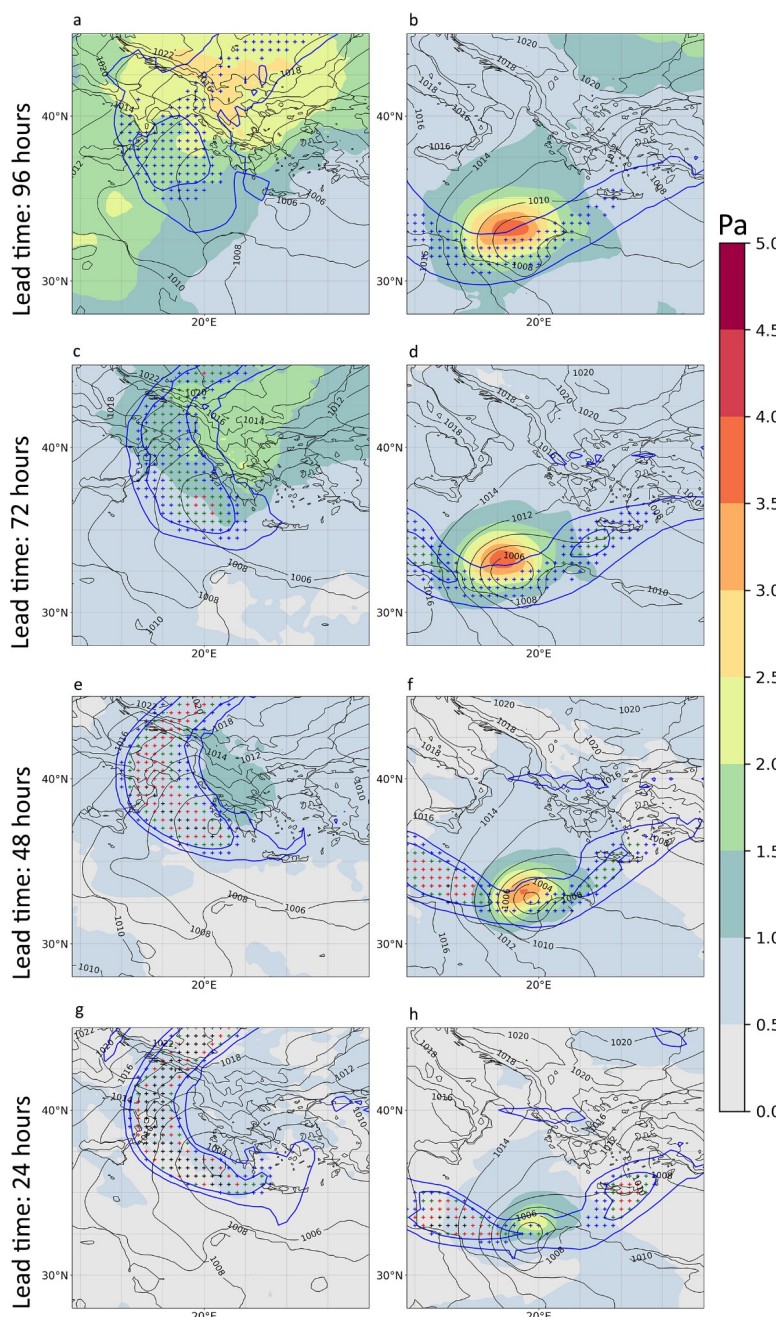

**Figure 7** Standard deviation (in colour) and average (in black contour) MSLP from the 51 ensemble members of the ECMWF EPS. Blue contours enclose areas with an average of 1 and 2 PVU at 300 hPa among all members of the EPS. Blue crosses indicate areas where more than 25% of the members have PV values greater than 2 PVU. Green, red, and black crosses denote member agreement at 50%, 75%, and 95%, respectively. Panels depict different lead forecast times valid on 5 September at 12 UTC (panels **a**, **c**, **e**, **g**) and 10 September at 12 UTC (**b**, **d**, **f**, **h**).




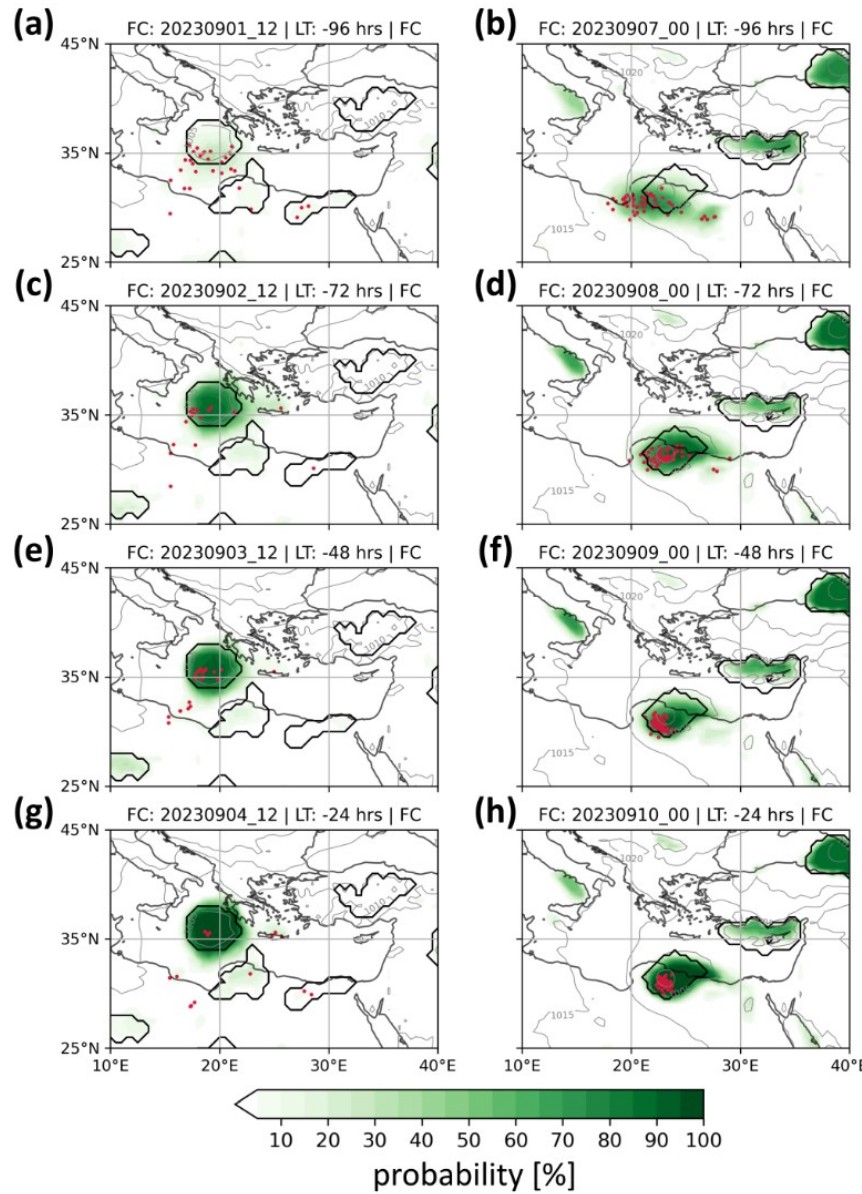

**Figure 8** Percentage of overlapping cyclone objects (green shading) among the ensemble prediction for different lead times valid on 5 September 2023, 1200 UTC (left column panels) and 11 September 2023, 0000 UTC (right column panels). Black contours show cyclone objects in ECMWF analysis (grey contours for MSLP in ECMWF analysis). Red dots depict the location of the minimum pressure of Daniel in the ensemble members.



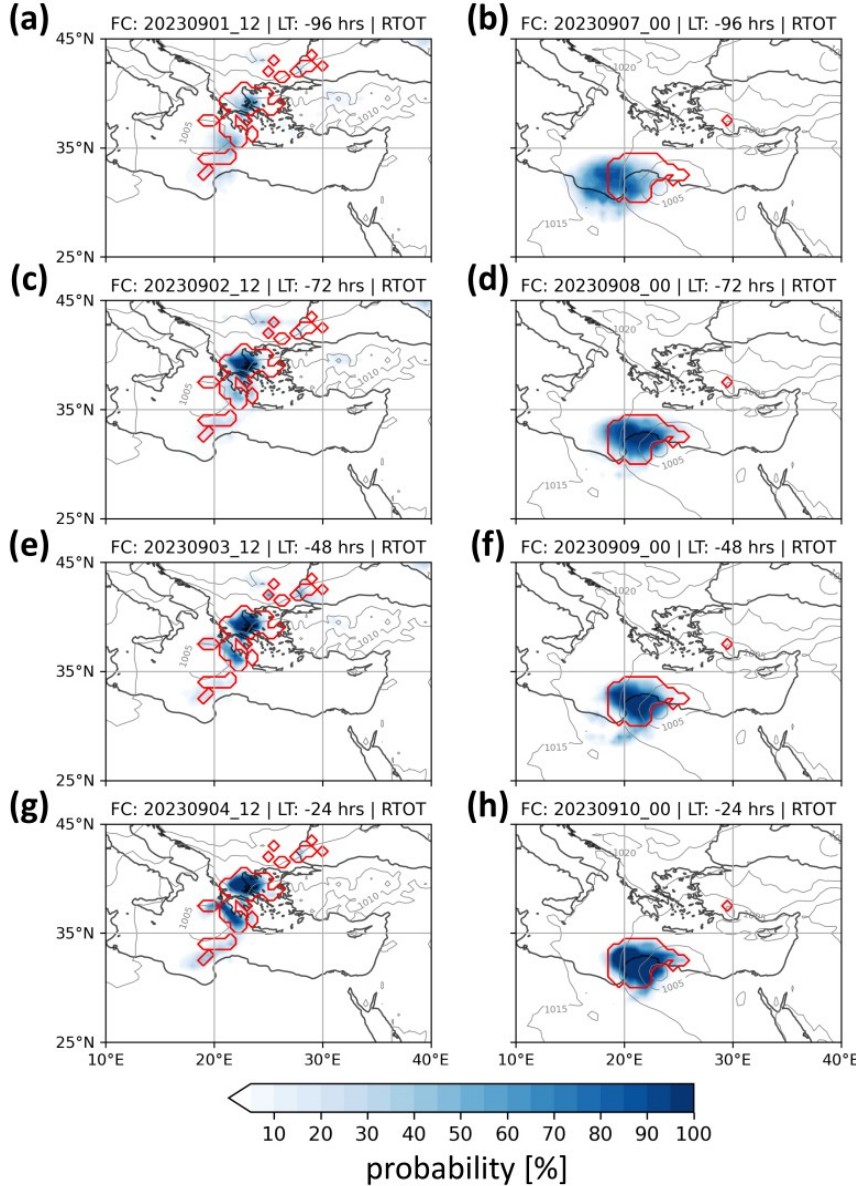

**Figure 9** Percentage of overlapping objects (in blue shading) among the ensemble prediction members for 24-hour accumulation of extreme precipitation for different lead times valid on 5 September 2023, 1200 UTC (left column panels) and 11 September 2023, 0000 UTC (right column panels). Red contours show objects of extreme precipitation determined based on an ERA5 climatology(grey contours for MSLP in ECMWF analysis).

54





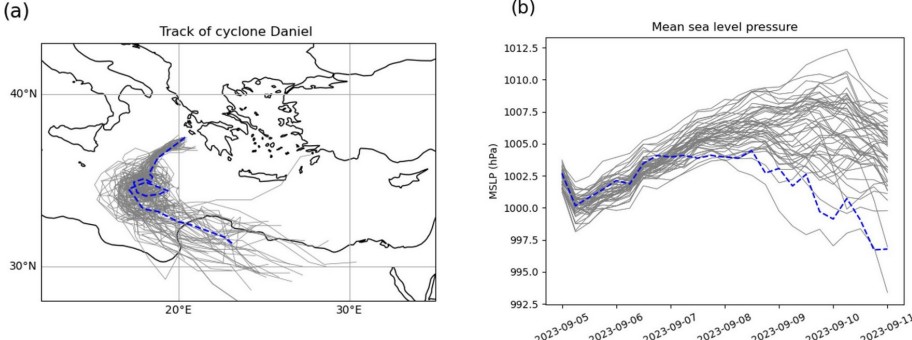

**Figure 10 (a)** MSLP at the centre of cyclone Daniel as represented by the ECMWF analysis (blue dashed line) and by the 50 members of the EPS of ECMWF (grey lines), initialized on 4 September at 0000 UTC. **(b)** As in **(a)** but for tracks of cyclone Daniel.

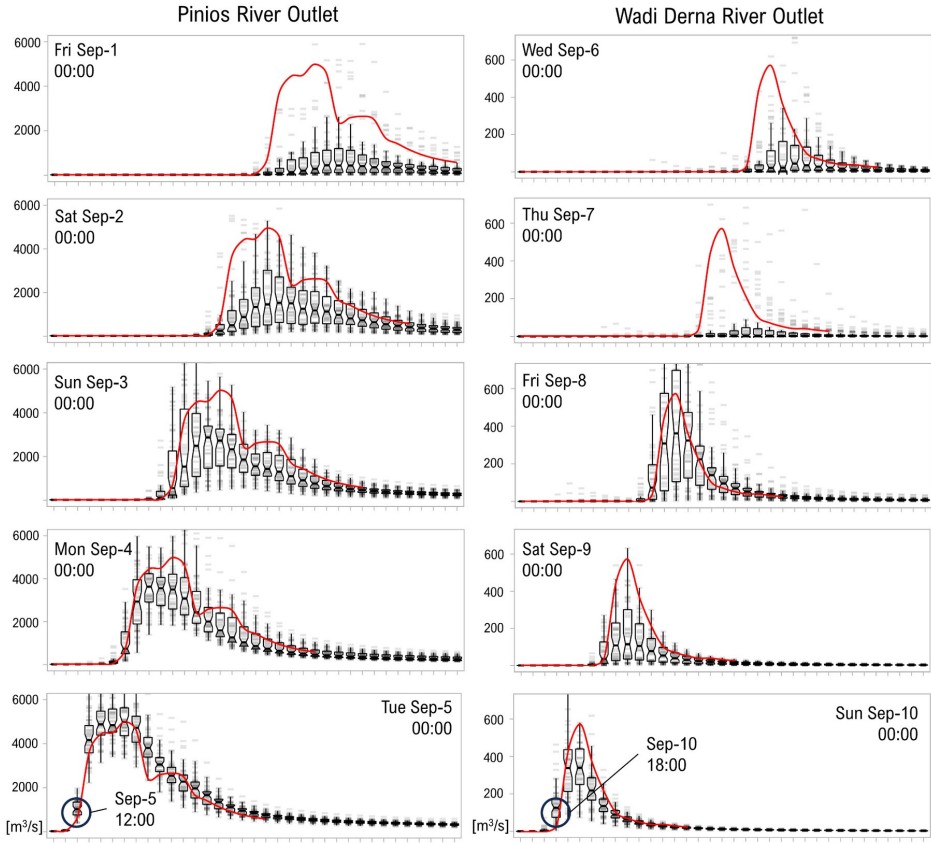

**Figure 11** Six-hourly ensemble forecasts of river discharge by the European Flood Awareness System, shown for different lead times. The red line represents the 'perfect forecast' benchmark, with the observed timing of rising hydrograph limbs marked on September 5th, 12:00 PM local time for the Pinios River in Thessaly, and September 10th, 18:00 local time for the Wandi Derna River. Box plots and grey lines indicate individual ensemble member predictions.



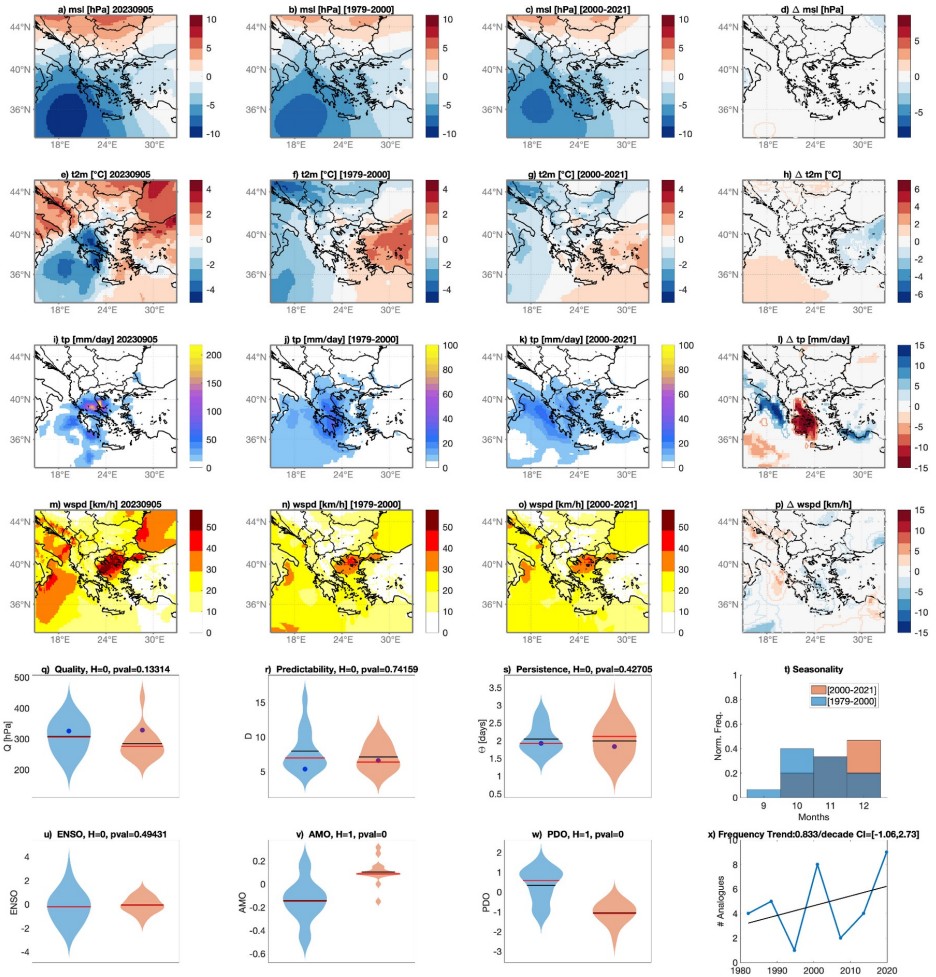

**Figure 12**: Analogues for 5 September 2023 and the region defined by [15°E 33°E 33°N 45°N] and the extended summer season SOND: average surface pressure anomaly (msl) (a), average 2-meter temperature anomalies (t2m) (e), accumulated total precipitation (tp) (i), and average wind-speed (wspd) in the period of the event. Average of the surface pressure analogs found in the counterfactual [1979-2000] (b) and factual periods [2001-2022] (c), along with corresponding 2-meter temperatures (f, g), accumulated precipitation (j, k), and wind speed (n, o). Changes between present and past analogs are presented for surface pressure Δslp (d), 2 meter temperatures Δt2m (h), total precipitation Δtp (i), and windspeed Δwspd (p): color-filled areas indicate significant anomalies concerning the bootstrap procedure. Violin plots for past (blue) and present (orange) periods for Quality Q analogs (q), Predictability Index D (r), Persistence Index Θ (s), and distribution of analogs in each month (t). Violin plots for past (blue) and present (orange) periods for ENSO (u), AMO (v) and PDO (w). Number of the Analogues occurring in each subperiod (blue) and linear trend (black). A blue dot marks values for the peak day of the extreme event. Horizontal bars in panels (q,r,s,u,v,w) correspond to the mean (black) and median (red) of the distributions.



1112

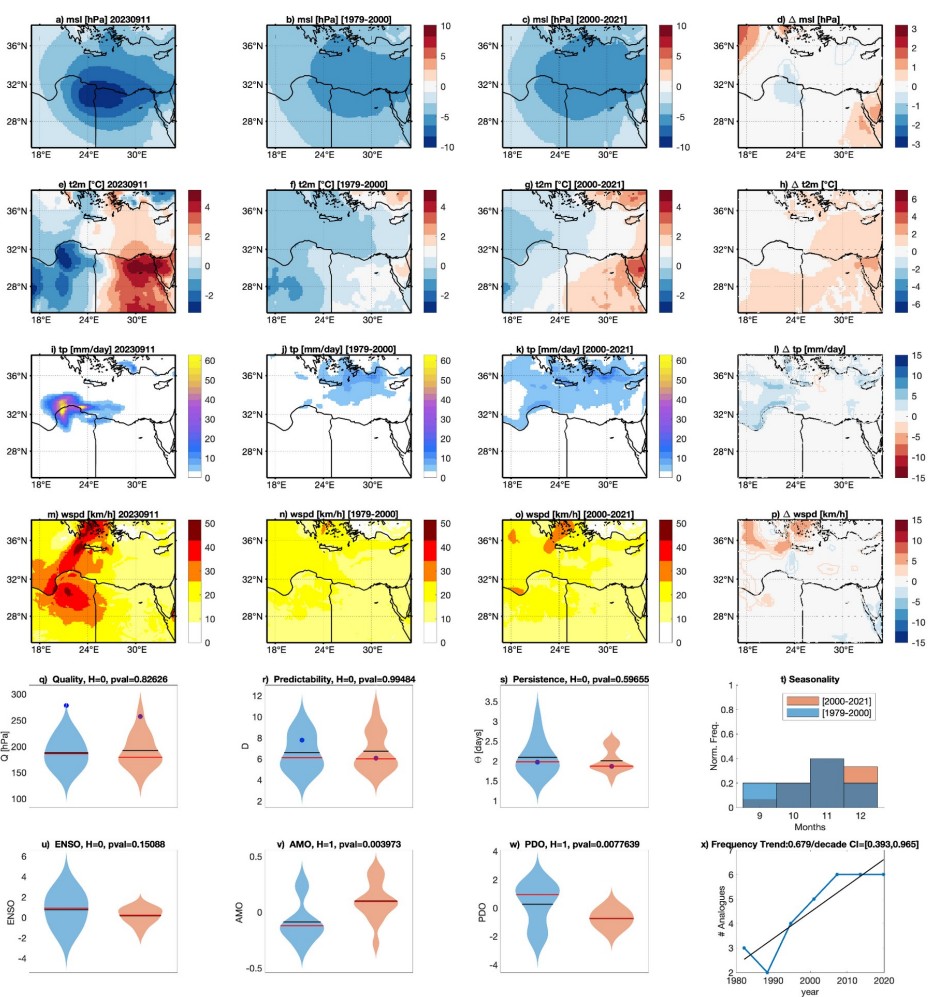

1114

**Figure 13:** As in Fig. 12, but for 10-11/09/2023, the region [17°E 35°E 25°N 38°N] and the extended
1116    autumn season (SOND).