# Peer review of "Dynamics, predictability, impacts, and climate change considerations of the catastrophic 2 Mediterranean Storm Daniel (2023)"

_EGUsphere, 2024_

## Referee Comment (RC2)

**Review**

**Dynamics, predictability, impacts, and climate change considerations of the catastrophic Mediterranean Storm Daniel (2023)**

Flaounas et al.

This manuscript contains a comprehensive analysis of Storm Daniel, choosing not to focus on one single aspect of its lifecycle, dynamics, impacts, drivers or climate change contextualisation but rather to include all those angles. I think this is what the authors mean with the term "holistic". While it is certainly needed a certain degree of effort to go through and understand the results from all the different analyses, in my opinion the authors do a very good job in presenting them in a single article that is successful in highlighting the importance of events like Daniel and describing "how they work".

I think that this manuscript could become a very worthwhile contribution to WCD and I don't have any major issues with it being accepted following revision. As I don't think the overall structure needs fundamental changes, I went into some level of detail with the line-by-line comments. There are quite a few of them (sorry!) but most should be quick and easy to address. I have also a few general comments, that in my opinion are more important. You can find them below. The last of those comments is on the section discussing the attribution to climate change. I understand that my tone there could possibly sound critical and dismissive and I would like to stress that that is not the case. I would be very happy to see the section retained in the manuscript but I think that there are some issues that need addressing. Should you find anything in my comments that is not clear, please feel free to contact me.

**General Comments:**

It is confirmed in the author contributions section that the manuscript was written by several of the authors and it does indeed give an impression of being a little "fragmented". My suggestion to improve readability would be that one of the authors could go through it and try to harmonise the writing style, choice of words and sentence structure, to improve its readability  (see the line-by-line comments).

Abstract: I hope this comment doesn't sound too subjective, but I think that this abstract is quite complex and long-winded. There are many points in which the novelty and importance of this study are highlighted, but their effectiveness is hindered by some sentences being too long, convoluted and not direct enough. Could you make it neater and sharper? See also some specific comments in the "Line-by-line comments" section.

Conclusions: I did not spend the same amount of time on them as I did on the abstract, so I would just recommend that as you improve the abstract you also amend the conclusions accordingly.

Attribution to climate change:
- I have some questions on the methodology used and on describing Daniel as hitting / making landfall over Greece, see in particular comments on Line 580 and 625.
- Coming to the conclusions that can be drawn from the section, it is stated that "Mediterranean depressions like Daniel hitting Greece and Libya show lower MSLP and higher precipitation in the present climate than in the past" and from this it is concluded that "[w]e thus interpret Daniel as an event whose characteristics can be ascribed to human-driven climate change". I understand the argument on heavier precipitation in a warmer climate (although it will have to be expanded without assuming that all readers are familiar with Clausius-Clapeyron and related implications) but no explanation is provided here on the changes in MSLP. As it stands, those conclusions look fairly weakly justified and not adding much to this particular study. For example, the "heavier precipitation in a warmer climate" link could have just been made by looking at the anomalously high SSTs during the event. Could you please make the case as to what the added value of this section is and why is it worth including it?

**Line-by-line comments:**

Line 38: It would be nice if you could remove this "probably" as you will agree that it doesn't sound great in an article abstract. I appreciate that you won't have access to every weather report that has ever been produced in the region but, assuming some prior research ( that you have likely already done), you could replace "probably" with "to our knowledge" or similar, or remove it altogether and add "in recent times" or similar (while also removing "ever").

Line 39-45: I like the creativity in the choice of words, but using simpler and shorter sentences could make it easier for the reader to understand what the aims, perspectives and strategies of this study are.

Line 46: "Our results ... cyclone". This is a bit vague. Are you referring to its structure, its intensity or other properties? Are Medicanes included in this "any other intense Mediterranean cyclone" terminology?

Line 50: "The predictability of  cyclone formation was rather low even  at relatively short lead times…".

Lines 52-53: "Our analysis of impacts shows that numerical weather prediction models are capable to capture the extreme character of …".

Line 58: "while  at its maturity".

Line 61: Another petty comment (sorry…), but the "event" is Daniel, not its impacts.

Lines 83-84: "Daniel was an intense cyclone, preceded by Rossby wave breaking over the Atlantic and the consequent intrusion of an upper-level trough": do you know if any weather report by national agencies was published showing this evolution? It would be nice to see this wave-breaking somewhere.

Line 146: "Can  numerical weather prediction models adequately  simulate climate extremes?"

Lines 192-195: Could you explain why "two-dimensional objects of extreme precipitation" are defined using 99$^{th}$ percentile values of precipitation **and wind**?

Lines 206: "Greece and Libya boxes"

Lines 216-217: I would say "we identified moisture sources in Daniel and in the 100 most extreme daily precipitation events"

Line 219: "Precipitate" or just condensate, given that the criterion only considers specific humidity?

Line 220: "All subsequent moisture uptakes or losses weight a moisture uptake". Could you rewrite this? I don't think I'm understanding what it means. Also, could you clarify what you mean by "weighted" in the following sentence?

Lines 240-242: "To account for the seasonal cycle in surface pressure and temperature data, we remove the average pressure and temperature values for the corresponding calendar days at each grid point and each day". I assume to do this you take advantage of the MSWX ensemble forecasts to increase sample size, given that there are only 22 years in each dataset. Could you give more information on this? For example, how many realisations do you use? At what lead time? ....

Lines 272-274: Could you motivate the choice of considering the role of ENSO, PDO and AMO (all having different frequency) among the many modes of natural variability? Also, as these indices are just diagnostic, what value do they add to the analogue analysis? In other words, does it matter if two cyclones with the same MSLP pattern occur during different phases of ENSO?

Figure 1: There seems to be something wrong with the track. If red dots are shown every 6 hours, then a few are missing between 6 Sep 18Z and 8 Sep 18Z and too many are shown between 11 Sep 00Z and 11 Sep 12Z. Also, the location 11 Sep 00Z red dot is not consistent with that of the black dot in Fig 2b.

Figure 1,caption: It should be "Storm Daniel" and not "Daniel storm" (please correct it here and elsewhere in the manuscript).

Lines 287-288: Are there any articles or reports showing the omega-blocking pattern and anticyclonic wave breaking that you could cite here?

Lines 303-304: Any reference for the NOAAN rainfall observations?

Fig 2: It is difficult to see what the max rainfall values are in both panels and to agree that they are "underestimated by about 50% in the ECMWF analysis" (lines 305-306) as all values above 200mm are purple. I'm sure you'll have already tried many different colour scales and intervals, so one possible alternative solution could be adding a small marker at the location of peak rainfall and annotating the value next to it.

Fig 3: In the caption, should "total moisture uptake" be replaced by "maximum moisture uptake", or am I getting this wrong? This comment is related to those regarding lines 219 and 220 (see above) as I have to admit that the methodology here is not totally clear to me. More generally, whilst I understand the need to show a large domain and thus for the Mediterranean region to be rather small, could you make the panels wider (and try playing with projections and domain edges, coastline and state boundary colours or adding zoomed-in boxes) to facilitate identifying locations in them? At the moment is not very easy to see where "the Aegean and Black Seas" (line 313) are, for example.

Line 315: "concomitant to the upper-level PV streamer". The wind barbs are at 850hPa while the PV streamer is at 300hPa. Why is it relevant that they share location and orientation?

Lines 320-324: Moisture sources for Storm Daniel are "partly in contrast" with the climatology, but also "somewhat overlap" with it. Could you please rewrite these sentences, as the above claims seem to be in conflict with each other?

Fig 4: Please replace "1993-2023" with "Jan 1993 – Aug 2023" in the two panels and in the first line of the caption.

Line 333: Replace "signify" with "indicate"?

Line 344: I don' t think "events" is correct here. Please rephrase.

Line 348: "in the central Mediterranean  was anomalously high,  roughly 2 K  above the average"

Lines 362-364: "While weaker than earlier, the wrap-up of the upper-level PV streamer around the cyclone centre was proposed to be responsible for its intensification just before the cyclone made landfall". Do you see any resemblance with the "low-PV bubble" dynamics highlighted in WCD - The impact of preceding convection on the development of Medicane Ianos and the sensitivity to sea surface temperature ? I am

not suggesting you should cite this work (of which as you know I am coauthor), it's just curiosity.

Lines 354-356: Is it possible to see these features anywhere (papers/reports/publications)?

Line 388: "( World Meteorological Organisation, 2023)"

Lines 400-401: "Eventually, after landfall, Daniel dissipated fast over the Sahara Desert when it reached Egypt on 11 September 2023." Is this sentence relevant here? Also, without adding any more context, could it not be at least partially in contrast with the inland intensification described in Hewson et al. (2024) (included in your references) and your earlier discussion on the importance of the upper-level setting for the intensification near landfall? Please consider removing or clarifying it.

Lines 425-426: "at the initial stage of Daniel, it is the timely prediction of cyclogenesis that would primarily provide useful information to civil protection". If, as you say, "most precipitation was produced in areas remote to the cyclone centre" and caused by moist flow impacting from NE, wouldn't it be that moist flow the key ingredient to be predicted rather than the actual presence of a fairly weak developing cyclone downstream? Can you elaborate on this? (here or in the section that is most suited to this discussion)

Lines 433-434: "numerical weather prediction models" (here and elsewhere).

Lines 452-454: "the direct relationship between the Rossby wave breaking over the Atlantic Ocean and the accurate prediction of Mediterranean cyclogenesis." From what you show, I would say that the direct relationship is between the upper-level PV streamer and the surface cyclogenesis (although the link between the streamer and wave breaking, not shown in this work, can certainly be mentioned and placed in the context of recent literature as currently done at the end of this paragraph).

Figure 7: In my view panels b,d,f,h should refer to 11 Sep 00z, to be consistent with Figs 2,8,9.

Lines 461-462; "while two members of the EPS do not even predict cyclogenesis (not shown)." Or is it shown anywhere?

Line 474: Is it 10 Sep or 11 Sep here? See comment on the date and time issues in Fig 1.

Line 482: "MSLP spread does not have a clear pattern"

Lines 484-486: It would be nice if you could add a discussion here on the agreement between EPS members on the PV streamer for 11 Sep, as it does not seem to be higher than that for 5 Sep. This would be particularly interesting as you previously highlighted its relevance for Daniel's intensification at this stage.

Figure 9: I suggest reconsidering the order of the figures, as (unless I'm missing something )Fig. 10 is mentioned before Fig.9 in the text.

Figure 9, caption: "Percentage of overlapping precipitation objects"

Figure 10: "(b) As in (a) but as a time series for tracks of cyclone Daniel."

Line 522: Shouldn't it be three days, given what you've just said on the limited agreement in Fig 9a and considerable increase in Fig9b? (by the way, I think you mean Fig 9c there)

Lines 528-529: "This suggests that the EPS members have been more consistent in the production of extreme precipitation even if cyclone centres presented a comparably greater spread." This is consistent with my comment on lines 425-426, on the moist flow towards Thessaly being the key ingredient for the prediction of the floods rather than the actual cyclogenesis further downstream.

Lines 539-540: What does "pretty corrected" mean?

Figure 11: Is it time that is indicated on the x-axis? Please specify it (including the interval between ticks). Also, could you add (a),(b),(c) … next to each panel?

Figure 11, caption: "Wandi Derna River" (here and in the text, line 551).

Lines 551-555: The discharge predictability for Wadi Derna River is generally lower than for Pinios (particularly on the 1st, 2nd and 4th rows from the top). Can you elaborate on this?

Line 580: I wouldn't use the expression "landfall over Greece" (here and later in this paragraph) given that the cyclogenesis is SW of Greece and then Daniel moves further away from it. Starting from this trivial comment, there is a more fundamental question that I would like to see discussed. If I understand well the methodology (apologies if this is not the case), the ClimaMeter framework uses single-time surface pressure patterns. This means that a cyclone going in the opposite direction to Daniel (e.g., eventually making landfall over Greece rather than moving away from it) would be considered a suitable analogue provided it has, for at least one time, a pressure pattern similar to Daniel's. This example cyclone could be associated with impacts throughout its evolution that are completely different from those associated with Daniel. I know that ClimaMeter has already been peer-reviewed and I'm not questioning its merits, but I would like to at least see a brief discussion of how the issue presented above can be considered acceptable, in particular in this study. Also, could this issue be avoided, if only partially, by selecting a substantially larger domain (and thus forcing a much larger region to have similar circulation?)

Line 583: I would remind the reader here that 15 analogues for each period are considered in the analysis.

Line 590: "persistence of all the cyclones".

Figure 12, caption: Is "concerning" the correct word in "color-filled areas indicate significant anomalies concerning the bootstrap procedure"?

Lines 592-593: "Figs 12q-s show no significant changes between the two periods (present and past climate)." If significance is evaluated using the test presented at line 595, then I would move its description before this sentence.

Lines 593-594: "We can... present periods". There must be a word missing here. Possibly "that" or "which" after "Q"?

Line 611: Is the period under analysis 10 Sep as written here or 10/11 Sep as in the caption of Figure 13?

Line 614: No description of T2m changes?

Lines 604-606 and 619-621: Sources of variability "may" have influenced the event. Written in this way it sounds like we don't know anything more about it that we didn't before the analysis. Could you rewrite it less vaguely and highlight what the result is?

Line 625: Daniel does not "hit" Greece (although some of the analogues may, see above). I think this choice of words is misleading.

---

## Author Comment (AC1)

**Reply document**

This document includes our reply to Ambrogio Volonte. In the following, the Reviewer's original comments are shown in Italics and our reply in blue-colored text.

*This manuscript contains a comprehensive analysis of Storm Daniel, choosing not to focus on one single aspect of its lifecycle, dynamics, impacts, drivers or climate change contextualisation but rather to include all those angles. I think this is what the authors mean with the term "holistic". While it is certainly needed a certain degree of effort to go through and understand the results from all the different analyses, in my opinion the authors do a very good job in presenting them in a single article that is successful in highlighting the importance of events like Daniel and describing "how they work".*

*I think that this manuscript could become a very worthwhile contribution to WCD and I don't have any major issues with it being accepted following revision. As I don't think the overall structure needs fundamental changes, I went into some level of detail with the line-by-line comments. There are quite a few of them (sorry!) but most should be quick and easy to address. I have also a few general comments, that in my opinion are more important. You can find them below. The last of those comments is on the section discussing the attribution to climate change. I understand that my tone there could possibly sound critical and dismissive and I would like to stress that that is not the case. I would be very happy to see the section retained in the manuscript but I think that there are some issues that need addressing. Should you find anything in my comments that is not clear, please feel free to contact me.*

We would like to thank Ambrogio Volonte for his positive review and the many fruitful comments. Indeed, we wanted to provide a "holistic" view of the event, aiming to link weather and climate perspectives of the same case study from the point of view of different disciplines. In this revised version, we have made substantial changes and made corrections, including the section on climate change attribution.

To make it clear in the introduction, we included the following text to highlight our motivation better:
*"When a high-impact weather event occurs, it encompasses multiple interconnected aspects often studied separately. First, understanding the event's dynamics and physical processes is crucial for assessing short-term forecasting and climate change implications. Second, the associated hazards—such as floods and windstorms—must be assessed concerning the specific conditions of the affected areas. This also raises questions about hazard predictability. Lastly, the event's severity must be placed within a climatological context to determine whether it produced extreme weather conditions and to attribute its intensity to climate change. Despite their interdependence, all these aspects of a specific weather event are rarely examined through an integrated approach. Our motivation is thus to apply a comprehensive framework, using Storm Daniel as the centerpiece of the September impacts in the eastern Mediterranean, and to provide an interdisciplinary assessment of the event (Shirzaei et al., 2025)..."*

***General Comments:***
*It is confirmed in the author contributions section that the manuscript was written by several of the authors and it does indeed give an impression of being a little "fragmented". My suggestion to improve readability would be that one of the authors could go through it and try to harmonise the writing style, choice of words and sentence structure, to improve its readability (see the line-by-line comments).*

Thank you for the suggestion. Indeed, this was a collective effort, aiming to combine the expertise of several colleagues. The new version underwent substantial editing to better harmonize the writing style.

*Abstract: I hope this comment doesn't sound too subjective, but I think that this abstract is quite complex and long-winded. There are many points in which the novelty and importance of this study are highlighted, but their effectiveness is hindered by some sentences being too long, convoluted and not direct enough. Could you make it neater and sharper? See also some specific comments in the "Line-by-line comments" section.*

The abstract has been substantially revised.

*Conclusions: I did not spend the same amount of time on them as I did on the abstract, so I would just recommend that as you improve the abstract you also amend the conclusions accordingly.*

The section on conclusions has also been revised accordingly to match changes in the manuscript.

*Attribution to climate change:*
*- I have some questions on the methodology used and on describing Daniel as hitting / making landfall over Greece, see in particular comments on Line 580 and 625.*

We acknowledge the reviewer's concerns regarding the description of Daniel as "hitting" or "making landfall" over Greece. In our revised text, we will refine our wording to better align with the meteorological characteristics of the event, distinguishing between landfall (typically associated with tropical systems) and the impact of a Mediterranean depression over Greece.

*- Coming to the conclusions that can be drawn from the section, it is stated that "Mediterranean depressions like Daniel hitting Greece and Libya show lower MSLP and higher precipitation in the present climate than in the past" and from this it is concluded that "[w]e thus interpret Daniel as an event whose characteristics can be ascribed to human-driven climate change". I understand the argument on heavier precipitation in a warmer climate (although it will have to be expanded without assuming that all readers are familiar with Clausius-Clapeyron and related implications) but no explanation is provided here on the changes in MSLP. As it stands, those conclusions look fairly weakly justified and not adding much to this particular study. For example, the "heavier precipitation in a warmer climate" link could have just been made by looking at the anomalously high SSTs during the event. Could you please make the case as to what the added value of this section is and why is it worth including it?*

The reviewer notes that while the link between a warmer climate and heavier precipitation is understood, the change in MSLP is not well explained. We clarify that our analysis does not claim a significant decrease in MSLP over Greece but rather identifies trends in analogues of Mediterranean depressions in general. Specifically:
1. The analogue approach allows us to assess whether Mediterranean depressions similar to Daniel have changed over time, including pressure, precipitation, and temperature fields.
2. Our methodology explicitly isolates trends in precipitation, demonstrating increased rainfall associated with similar systems over the Ionian Sea and Albania in the present climate.
3. The significant warming over the Mediterranean Sea (Figure 12e-h) is a key driver of this increase in precipitation, consistent with Clausius-Clapeyron scaling.

To clarify this further, we will expand our discussion of precipitation trends and explicitly state that the observed MSLP changes in Greece are less pronounced than in Libya. Instead, we emphasize that the increase in precipitation and frequency of similar systems are robust signals of climate change.

The reviewer also questions whether our conclusions could have been drawn solely from the observed high SSTs during the event. While high SSTs are important, our methodology provides added value by contextualizing Daniel within historical atmospheric patterns. The key contributions of this section include:

1. Demonstrating that similar Mediterranean depressions have become more frequent in the present climate, particularly in December (Figure 12t).
2. Identifying statistically significant changes in precipitation and temperature patterns associated with these systems, independent of a single SST anomaly.
3. Showing that while large-scale climate variability modes such as ENSO, AMO, and PDO can influence atmospheric conditions, our analysis does not establish a direct causal link between these modes and the development of Daniel; rather, this assessment is exploratory, highlighting potential associations without making definitive attributions given the limitations of a 40-year dataset.
4. By comparing the event against a database of past analogues, we provide a broader climatological perspective rather than relying on one event's SST anomaly alone. We will clarify this in the revised text to highlight why this approach strengthens the attribution analysis.

In response to the reviewer's concerns, we have:

- Improved the phrasing of Daniel's impact on Greece to avoid misleading terminology.
- Explicitly stated that significant changes in MSLP are not evident in Greece, but changes in precipitation are.
- Expanded the discussion on precipitation changes, connecting them more clearly to Clausius-Clapeyron without assuming reader familiarity.
- Highlighted the added value of the analogue approach compared to SST-based reasoning alone.

We hope these clarifications address the reviewer's concerns and strengthen the clarity of our attribution findings.

***Line-by-line comments:***

*Line 38: It would be nice if you could remove this "probably" as you will agree that it doesn't sound great in an article abstract. I appreciate that you won't have access to every weather report that has ever been produced in the region but, assuming some prior research ( that you have likely already done), you could replace "probably" with "to our knowledge" or similar, or remove it altogether and add "in recent times" or similar (while also removing "ever").*

Indeed, we did not find a more catastrophic flood event in the Mediterranean and therefore, we rephrased "*to our knowledge*".

*Line 39-45: I like the creativity in the choice of words, but using simpler and shorter sentences could make it easier for the reader to understand what the aims, perspectives and strategies of this study are.*

The abstract has been revised.

*Line 46: "Our results … cyclone". This is a bit vague. Are you referring to its structure, its intensity or other properties? Are Medicanes included in this "any other intense Mediterranean cyclone" terminology?*

Thank you for this comment. Indeed, the formation mechanism of medicanes is not expected to differ from the one involved in any other intense Mediterranean cyclone. We clarified this in the abstract:
*"Daniel initially developed like any other intense Mediterranean cyclone, including medicanes, due to upper tropospheric forcing followed by Rossby wave breaking."*

*Line 50: "The predictability of the cyclone formation was rather low even in at relatively short lead times…".*
*Lines 52-53: "Our analysis of impacts shows that numerical weather prediction models are capable to capture the extreme character of …".*
*Line 58: "while in at its maturity".*
*Line 61: Another petty comment (sorry…), but the "event" is Daniel, not its impacts.*

The abstract has been revised.

*Lines 83-84: "Daniel was an intense cyclone, preceded by Rossby wave breaking over the Atlantic and the consequent intrusion of an upper-level trough": do you know if any weather report by national agencies was published showing this evolution? It would be nice to see this wave-breaking somewhere.*

We added the reference of Couto et al. (2024), who examined the large-scale dynamics prior to the formation of Daniel.

*Line 146: "Can Are numerical weather prediction models adequately for the prediction simulate climate extremes?"*

Done.

*Lines 192-195: Could you explain why "two-dimensional objects of extreme precipitation" are defined using 99th percentile values of precipitation and wind?*

Thank you for pointing out a typo. The two-dimensional objects of extreme precipitation are, of course, defined by using the 99th percentile values of precipitation only. This has been corrected in the revised version of the manuscript.

*Lines 206: "Greece and Libya boxes"*

Done.

*Lines 216-217: I would say "we identified moisture sources in Daniel and in the 100 most extreme daily precipitation events"*

Done.

*Line 219: "Precipitate" or just condensate, given that the criterion only considers specific humidity?*

If specific humidity decreases during the last time step and relative humidity is higher than 90%, we define this as a "precipitating air parcel". This follows the definition by Sodemann et al. (2008) and assumes that the missing humidity during this time step will precipitate. Thanks to this question, we noticed a bug in our script during review, which led, however, to irrelevant changes in the moisture source distributions.

We adjusted the text to include this information:

*"This method involves the tracking of changes in specific humidity along all trajectories that precipitate upon arrival, which are defined as air parcels showing a decrease in specific humidity during the last time step before arrival and a relative humidity larger than 90% upon arrival (following Sodemann et al., 2008)."*

*Line 220: "All subsequent moisture uptakes or losses weight a moisture uptake". Could you*

*rewrite this? I don't think I'm understanding what it means. Also, could you clarify what you mean by "weighted" in the following sentence?*

Thank you for pointing out that more explanation is needed for the moisture source diagnostic. We reformulated this part and added a reference to the method's paper:

*"Along each trajectory, an increase in specific humidity is interpreted as a moisture uptake, and a decrease in specific humidity is interpreted as a moisture loss. Each moisture loss reduces all previous moisture uptakes, weighted by their uptake amount. For a detailed description of the moisture source diagnostic, see Sodemann et al. (2008)."*

*Lines 240-242: "To account for the seasonal cycle in surface pressure and temperature data, we remove the average pressure and temperature values for the corresponding calendar days at each grid point and each day". I assume to do this you take advantage of the MSWX ensemble forecasts to increase sample size, given that there are only 22 years in each dataset. Could you give more information on this? For example, how many realisations do you use? At what lead time? ….*

Thanks for this remark, MSWX does not provide ensemble forecasts. It only uses a single realization for producing its real time product.

*Lines 272-274: Could you motivate the choice of considering the role of ENSO, PDO and AMO (all having different frequency) among the many modes of natural variability? Also, as these indices are just diagnostic, what value do they add to the analogue analysis? In other words, does it matter if two cyclones with the same MSLP pattern occur during different phases of ENSO?*

The choice of these modes of variability comes from the fact that they are the main oceanic modes that influence the weather at scales longer than one year. The use of these indices allows building other counterfactual worlds, namely those with positive or negative phases of the indices, and therefore allows to determine whether the changes observed could be linked to natural variability instead of greenhouse gas emissions

*Figure 1: There seems to be something wrong with the track. If red dots are shown every 6 hours, then a few are missing between 6 Sep 18Z and 8 Sep 18Z and too many are shown between 11 Sep 00Z and 11 Sep 12Z. Also, the location 11 Sep 00Z red dot is not consistent with that of the black dot in Fig 2b.*

Thanks for noting this. Figure 1 has been modified, correcting the mistakes in dates and time indicated in the original figure. To accommodate the Reviewer's 1 comments, we have added more information on the plot.

*Figure 1,caption: It should be "Storm Daniel" and not "Daniel storm" (please correct it here and elsewhere in the manuscript).*

Corrected.

*Lines 287-288: Are there any articles or reports showing the omega-blocking pattern and anticyclonic wave breaking that you could cite here?*

We added the reference of Couto et al. (2024).

*Lines 303-304: Any reference for the NOAAN rainfall observations?*

This reference has already been included:

*"Lagouvardos, K., Kotroni, V., Bezes, A., Koletsis, I., Kopania, T., Lykoudis, S., Mazarakis, N., Papagiannaki, K., and Vougioukas, S.: The automatic weather stations NOANN network of the National Observatory of Athens: operation and database, Geoscience Data Journal, 4, 4–16, https://doi.org/10.1002/gdj3.44, 2017."*

*Fig 2: It is difficult to see what the max rainfall values are in both panels and to agree that they are "underestimated by about 50% in the ECMWF analysis" (lines 305-306) as all values above 200mm are purple. I'm sure you'll have already tried many different colour scales and intervals, so one possible alternative solution could be adding a small marker at the location of peak rainfall and annotating the value next to it.*

We have added some additional information about the differences between the IFS and observation maximum 24-hr accumulated precipitation:

*"The NOAAN surface stations recorded more than 750 mm of daily rainfall and up to 1,235 mm within four days in eastern parts of the Thessaly region (flooded areas are shown in cyan colours in Fig. 1b). Notably these peak values are underestimated by about 40% in the ECMWF analysis (max IFS 24-h accumulated rainfall equal to 434 mm on 6 September 2023 00 UTC)."*

*Fig 3: In the caption, should "total moisture uptake" be replaced by "maximum moisture uptake", or am I getting this wrong? This comment is related to those regarding lines 219 and 220 (see above) as I have to admit that the methodology here is not totally clear to me. More generally, whilst I understand the need to show a large domain and thus for the Mediterranean region to be rather small, could you make the panels wider (and try playing with projections and domain edges, coastline and state boundary colours or adding zoomed-in boxes) to facilitate identifying locations in them? At the moment is not very easy to see where "the Aegean and Black Seas" (line 313) are, for example.*

Thank you for pointing this out. The caption of Figure 3 should be "relative moisture uptakes" as they give the relative contribution per km^2 of each grid cell to the total moisture that contributes to the precipitation event for the respective day. The contour lines show another unit, which represents the percentage of the total moisture uptake, thereby summing up from largest to smallest relative moisture contribution per grid point. We've adjusted the caption to make this more straightforward. Concerning questions on the methods, please see the answer to the question regarding lines 219-220. Finally, we've adjusted Figure 3 so that it is easier to identify moisture uptake hot spots. This also includes adding the land/ocean fraction of the moisture uptakes and a comparison with results from a recent study (Argüeso et al, 2024). The text has been adjusted in the following way:

*"Further moisture mainly originated from central to eastern Europe and the North Atlantic Ocean. These source regions are in general agreement with a recent study (Argüeso et al., 2024), which investigated moisture sources of rainfall over Greece from 3 to 9 Sep 2023 using a Eulerian moisture source diagnostic. Our moisture source analysis shows larger contributions from land (54.7%) than in Argüeso et al. (2024) (27%). The Lagrangian method used in our study shows relatively large moisture contributions from north of the Black Sea because most of the air parcels arriving on 5 Sep 2023 descended and took up moisture in this region before moving southwestward along the western flank of the PV streamer. The differences in the land fraction between the two methods might originate from different periods used for the moisture source calculations, different handling of moisture uptakes above the boundary layer, a lower explained fraction of the total moisture sources (84%) with the Eulerian compared to the Lagrangian diagnostic (explained fraction of 90%), and general differences in Eulerian versus Lagrangian approaches. An ongoing comparison study of moisture source diagnostics is investigating differences in these methods in detail and will*

*shed more light on disagreements between various moisture source diagnostics."*

*Line 315: "concomitant to the upper-level PV streamer". The wind barbs are at 850hPa while the PV streamer is at 300hPa. Why is it relevant that they share location and orientation?*

Concomitant might be a strong word for the context. We revised it to "*having a similar orientation with the upper-level PV streamer."*

*Lines 320-324: Moisture sources for Storm Daniel are "partly in contrast" with the climatology, but also "somewhat overlap" with it. Could you please rewrite these sentences, as the above claims seem to be in conflict with each other?*

We revised as follows:
*"The moisture sources shown in Fig. 3a largely overlap with the climatological moisture sources of extreme precipitation in the same area. However, for Daniel, they are mostly concentrated over the Aegean Sea and areas to the northeast. In contrast, the typical moisture sources for extreme precipitation in Thessaly extend further over the central Mediterranean (Fig. 3b)."*

*Fig 4: Please replace "1993-2023" with "Jan 1993 – Aug 2023" in the two panels and in the first line of the caption.*

We replaced it with "Jan 1993 – Aug 2023" in the figure and caption.

*Line 333: Replace "signify" with "indicate"?*

We replaced it with "indicate."

*Line 344: I don't think "events" is correct here. Please rephrase.*

We removed "events."

*Line 348: "in the central Mediterranean has been was anomalously high, by roughly 2 K respect to above the average"*

This part has been revised.

*Lines 362-364: "While weaker than earlier, the wrap-up of the upper-level PV streamer around the cyclone centre was proposed to be responsible for its intensification just before the cyclone made landfall". Do you see any resemblance with the "low-PV bubble" dynamics highlighted in WCD - The impact of preceding convection on the development of Medicane Ianos and the sensitivity to sea surface temperature ? I am not suggesting you should cite this work (of which as you know I am coauthor), it's just curiosity.*

It may be that the outflow of deep convection hinders the eastward progression of the upper-level PV streamer and the subsequent interaction with the cyclone similarly to the case of Medicane Ianos. However, without dedicated sensitivity tests to factors controlling convective intensity as in Pantillon et al. (2024) and Sanchez et al. (2024), assessing such a complex feedback is challenging. Here we prefer not to speculate on the detailed dynamical mechanisms and rather emphasize the anomalous intensification during landfall.

*Lines 354-356: Is it possible to see these features anywhere (papers/reports/publications)?*

Unfortunately, we could find publications supporting our analysis, but our statements are

based on various open-source data that we decided not to include in the main manuscript.

Regarding deep moist convection and lighting activity, we have attached Figures A1 and A2, showing the total lightning activity that was recorded by the Blitzortung network (https://www.lightningmaps.org/blitzortung/europe/index.php). On 6 Sep 2023, the largest part of the Ionian Sea and Greece were affected by lightning activity (Fig. A1), whereas on 9 Sep 2023, lighting was limited close to the cyclone centre (Fig. A2).

[Figure]

Figures A1-A2 24-h accumulated total lightning detections (strokes) on 6 and 9 September 2023 by the Blitzortung network.

Figures A3 and A4 support our statement about the morphological changes in the wind field close to the sea surface. On 6 Sep 2023, the highest wind speeds that were estimated by the ASCAT instrument onboard MetOp-C satellite (https://manati.star.nesdis.noaa.gov/datasets/ASCATCData.php) were found far away from the cyclone centre (Fig. A3), and on 9 Sep 2023 the winds close to the cyclone centre were much stronger, with the highest wind speeds a few kilometers northwest from the Storm Daniel's centre.

[Figure]

Figures. A3-A4 Advanced Scatterometer (ASCAT METOP-C) near surface wind speed on 6 and 9 September 2023.

*Line 388: "(Weather World Meteorological Organisation, 2023)"*

Done

*Lines 400-401: "Eventually, after landfall, Daniel dissipated fast over the Sahara Desert when it reached Egypt on 11 September 2023." Is this sentence relevant here? Also, without adding any more context, could it not be at least partially in contrast with the inland intensification described in Hewson et al. (2024) (included in your references) and your earlier discussion on the importance of the upper-level setting for the intensification near landfall? Please consider removing or clarifying it.*

Thank you for spotting this. Indeed, the phrase was wrong, and thus it was removed.

*Lines 425-426: "at the initial stage of Daniel, it is the timely prediction of cyclogenesis that would primarily provide useful information to civil protection". If, as you say, "most precipitation was produced in areas remote to the cyclone centre" and caused by moist flow impacting from NE, wouldn't it be that moist flow the key ingredient to be predicted rather than the actual presence of a fairly weak developing cyclone downstream? Can you elaborate on this? (here or in the section that is most suited to this discussion)*

Thank you for this insightful comment. We revised these lines as follows:
*"Therefore, accurately forecasting the time and location of cyclone formation (as shown in Fig. 8) may play a secondary role in predicting its impacts in Greece. In this context, the reliable simulation of moisture inflow—which appears to be more closely linked to large-scale circulation, as previously discussed—by the EPS members could be more crucial for impact prediction."*

*Lines 433-434: "numerical weather prediction models" (here and elsewhere).*

Done

*Lines 452-454: "the direct relationship between the Rossby wave breaking over the Atlantic Ocean and the accurate prediction of Mediterranean cyclogenesis." From what you show, I would say that the direct relationship is between the upper-level PV streamer and the surface cyclogenesis (although the link between the streamer and wave breaking, not shown*

*in this work, can certainly be mentioned and placed in the context of recent literature as currently done at the end of this paragraph).*

Thank you for this comment. This part has been revised:

*"The similar behaviour in the cyclone and PV streamer predictability relies on the direct relationship between the Rossby wave breaking over the Atlantic Ocean and the accurate prediction of Mediterranean cyclogenesis. This has been highlighted by Chaboureau et al. (2012) and Pantillon et al. (2013) for the case of the extratropical transition…."*

*Figure 7: In my view panels b,d,f,h should refer to 11 Sep 00z, to be consistent with Figs 2,8,9.*

The Figure has been revised as suggested.

*Lines 461-462; "while two members of the EPS do not even predict cyclogenesis (not shown)." Or is it shown anywhere?*

This part of the text has been revised following the changes in the lead times shown in Figs. 7-9.

*Line 474: Is it 10 Sep or 11 Sep here? See comment on the date and time issues in Fig 1.*

Thank you for spotting this. "Mature stage" has been deleted. The 10th of September refers to the day of landfall.

*Line 482: "MSLP spread does not have a clear pattern"*

Done.

*Lines 484-486: It would be nice if you could add a discussion here on the agreement between EPS members on the PV streamer for 11 Sep, as it does not seem to be higher than that for 5 Sep. This would be particularly interesting as you previously highlighted its relevance for Daniel's intensification at this stage.*

We added the following at the end of section 4.2:
*"Upper tropospheric forcing is crucial in accurately predicting cyclone intensity in this context. While Fig. 7b —unlike Fig. 7a— shows that some EPS members align with the location of this upper tropospheric feature (blue crosses), an average of 2 PVU and an agreement above 50% among the EPS members near the cyclone center is only evident at a lead time of approximately three days (Fig. 7f, depicted by green crosses)."*

*Figure 9: I suggest reconsidering the order of the figures, as (unless I'm missing something ). Fig. 10 is mentioned before Fig.9 in the text. Figure 9, caption: "Percentage of overlapping precipitation objects"*

The order has been changed.

*Figure 10: "(b) As in (a) but as a time series for tracks of cyclone Daniel."*

Done.

*Line 522: Shouldn't it be three days, given what you've just said on the limited agreement in Fig 9a and considerable increase in Fig9b? (by the way, I think you mean Fig 9c there)*

Thank you for the corrections. This part has been revised accordingly.

*Lines 528-529: "This suggests that the EPS members have been more consistent in the production of extreme precipitation even if cyclone centres presented a comparably greater spread." This is consistent with my comment on lines 425-426, on the moist flow towards Thessaly being the key ingredient for the prediction of the floods rather than the actual cyclogenesis further downstream.*

Thank you for this insightful comment. We revised these lines as follows:
"Therefore, accurately forecasting the time and location of cyclone formation (as shown in Fig. 8) may play a secondary role in predicting its impacts in Greece. In this context, the reliable simulation of moisture inflow—which appears to be more closely linked to large-scale circulation, as previously discussed—by the EPS members could be more crucial for impact prediction."

*Lines 539-540: What does "pretty corrected" mean?*

The phrase has been corrected to
"The probability strongly increases at shorter lead times (Figs 10f and 10h) mostly and all EPS members tend to converge to similar cyclone locations when reaching a lead time of one day (Fig. 8h)."

*Figure 11: Is it time that is indicated on the x-axis? Please specify it (including the interval between ticks). Also, could you add (a),(b),(c) … next to each panel?*

The x-axis indicates the lead time for each panel (initialization) in 6h time intervals. This was added in the revised figure, as well as letters [(a),(b),(c) etc.] in each panel. Figure caption has been extended for clarity.

*Figure 11, caption: "Wandi Derna River" (here and in the text, line 551).*
We used "Wadi Derna" throughout the manuscript, ensuring alignment with the accepted terminology in the literature.
*Lines 551-555: The discharge predictability for Wadi Derna River is generally lower than for Pinios (particularly on the 1st, 2nd and 4th rows from the top). Can you elaborate on this?*

This section has been thoroughly revised to explain the differences in predictability, addressing the challenges specific to Wadi Derna. Below is the revised text:

"The forecasts for the Wadi Derna River outlet (Fig. 11, right panels) exhibit significant variability and fail to converge during the earlier forecast initialization dates as well as at shorter lead times. This persistent lack of convergence can be attributed to distinct challenges of both temporal scales. For earlier forecast initialization dates, the primary source of variability lies in the westward displacement of extreme precipitation predicted by the EPS (Figs. 10b and 10d). For example, forecasts initialized on 9 September, during a critical period for implementing preventative measures, display a wide spread and a shortfall in the median forecast compared to the benchmark (red line). This variability persists even for forecasts initialized on 10 September. The failure to converge at shorter lead times is compounded by challenges inherent to the Wadi Derna catchment. The resolution of the precipitation forcings used in the forecasts combined with the relatively small size (575 km2) and flash-flood-prone nature of this basin amplify the uncertainties in predicting discharge, particularly in response to localized extreme rainfall."

*Line 580: I wouldn't use the expression "landfall over Greece" (here and later in this paragraph) given that the cyclogenesis is SW of Greece and then Daniel moves further away*

*from it. Starting from this trivial comment, there is a more fundamental question that I would like to see discussed. If I understand well the methodology (apologies if this is not the case), the ClimaMeter framework uses single-time surface pressure patterns. This means that a cyclone going in the opposite direction to Daniel (e.g., eventually making landfall over Greece rather than moving away from it) would be considered a suitable analogue provided it has, for at least one time, a pressure pattern similar to Daniel's. This example cyclone could be associated with impacts throughout its evolution that are completely different from those associated with Daniel. I know that ClimaMeter has already been peer-reviewed and I'm not questioning its merits, but I would like to at least see a brief discussion of how the issue presented above can be considered acceptable, in particular in this study. Also, could this issue be avoided, if only partially, by selecting a substantially larger domain (and thus forcing a much larger region to have similar circulation?)*

We appreciate the reviewer's careful consideration of the methodology and their suggestion to refine our description of Daniel's evolution. We agree that "landfall over Greece" is not the most accurate phrasing, as the cyclogenesis occurred southwest of Greece before Daniel moved further away. We will revise this wording to reflect the storm's trajectory better.

Regarding the broader methodological question, the reviewer correctly identifies that ClimaMeter uses single-time surface pressure patterns to identify analogues. This means that a cyclone moving in the opposite direction to Daniel could, in principle, be considered an analogue if it exhibited a similar pressure pattern at a given moment. However, this limitation is mitigated in our study in several ways. First, while our analogue search is based on surface pressure alone, the subsequent analysis examines associated temperature, precipitation, and wind speed patterns to ensure that the analogues share broader dynamical similarities with Daniel. This helps to filter out cases where the identified analogue might have evolved in a vastly different manner.

Second, the issue of analogues with different tracks is partly addressed by the regional domain selection. While a significantly larger domain could, as the reviewer suggests, constrain the analogues further by ensuring that a broader area exhibits similar circulation patterns, it would also risk including patterns that match at a large scale but deviate in local storm dynamics. The current domain size represents a balance between capturing the key features of the Mediterranean depressions and avoiding overly restrictive constraints that could reduce the analogue sample size.

To acknowledge this point, we added a brief discussion in the manuscript outlining this trade-off and explaining that while our approach prioritizes pressure pattern similarity, the additional analysis of precipitation and wind fields ensures that the identified analogues remain meteorologically relevant. We will also clarify that while increasing the domain size could help filter out analogues with very different storm trajectories, it would not necessarily resolve all limitations. This transparency will ensure that readers understand both the strengths and potential constraints of the analogue-based approach used in ClimaMeter for this study.

*Line 583: I would remind the reader here that 15 analogues for each period are considered in the analysis.*
*Line 590: "persistence of all the cyclones".*

Thank you. We have rephrased this part.

*Figure 12, caption: Is "concerning" the correct word in "color-filled areas indicate significant anomalies concerning the bootstrap procedure"?*

Thank you. We have changed the text for clarity.

*Lines 592-593: "Figs 12q-s show no significant changes between the two periods (present and past climate)." If significance is evaluated using the test presented at line 595, then I would move its description before this sentence.*

Thank you. We have rephrased this part.

*Lines 593-594: "We can… present periods". There must be a word missing here. Possibly "that" or "which" after "Q"?*

Thank you. We have rephrased this part.

*Line 611: Is the period under analysis 10 Sep as written here or 10/11 Sep as in the caption of Figure 13?*

Thank you. We have rephrased this part.

*Line 614: No description of T2m changes?*

Thank you, we have added the temperature changes.

*Lines 604-606 and 619-621: Sources of variability "may" have influenced the event. Written in this way it sounds like we don't know anything more about it that we didn't before the analysis. Could you rewrite it less vaguely and highlight what the result is?*

We have rephrased this part, also acknowledging the exploratory nature of this analysis.

*Line 625: Daniel does not "hit" Greece (although some of the analogues may, see above). I think this choice of words is misleading.*

We have used "impacted" now.

**References:**

Argüeso, D., Marcos, M. and Amores, A.: Storm Daniel fueled by anomalously high sea surface temperatures in the Mediterranean. npj Clim Atmos Sci 7, 307 (2024). https://doi.org/10.1038/s41612-024-00872-2

Sodemann, H., Schwierz, C., and Wernli, H.: Interannual variability of Greenland winter precipitation sources: Lagrangian moisture diagnostic and North Atlantic Oscillation influence, J. Geophys. Res., 113, 2007JD008503 (2008) https://doi.org/10.1029/2007JD008503

---

## Author Comment (AC2)

**Reply document**
This document includes our reply to the comments of the first Reviewer. In the following, the original comments are shown in Italics and our reply in blue-colored text.

*Overview*

*This manuscript constitutes a case study for cyclone/medicane "Daniel" which, at different stages in its lifecycle, delivered devastating weather and impacts in parts of both Greece and Libya, in early September 2023. Four aspects are examined, as listed at the start of the title.*

*The most novel and publication-worthy features of the study are the moisture source analyses, for both the Greek and Libyan floods, the sea wave analysis and the use of cyclone and precipitation objects.*

*The remainder of the study does not add much to previous published literature on this case (notably Hewson et al, 2024) which is cited and latterly Couto et al (2024), which focusses on broadscale aspects. Admittedly the Couto paper, available here: https://www.mdpi.com/2073-4433/15/10/1205, has only just appeared so was probably unavailable to the authors pre-submission. In some respects these two papers go much further than the one under review, particularly with regard broadscale patterns, local details of the extreme weather and considerations with regard to high impact warnings. Given that a standard requirement for publication, in any journal, is that one adds to previously established knowledge (rather than detracting from it) it is clear, in the opinion of this reviewer, that a very substantial reworking of the paper's content would be required for acceptance.*

*There is a clear reluctance to include observations in this paper – notably rainfall measurements. If numerical model analyses were perfect, this might be acceptable, but given that they are not, particularly with regard to rainfall, which is the impact centrepiece of this study, this is a major omission.*

*Furthermore, reviewing the paper has been a frustrating process due to the many inconsistencies in different segments of the text, inconsistencies between what the figures show and what the text says, simple errors, poorly explained figures, and unsubstantiated conclusions. Rather than go through absolutely everything which is of concern, which would take a very long time and replicate the checks the authors themselves should have carried out before submission, I will instead go through the figures, which are the bedrock of the paper, and highlight the key issues via those.*

We thank the reviewer for taking the time to comment on our manuscript in detail. We answered all the comments and suggestions below.

*Main points*

*Figure 1a: Some of the red spots are missing (assuming the time interval is 6 hours, which should anyway be stated); some of the labels show the wrong time, and the size-for-mslp legend is hard to interpret. At one stage in the text it is stated that the cyclone was intense early in its lifecycle – by normal measures 1004mb is not intense – and indeed elsewhere in the text this statement is contradicted. Somewhere else in the text it says that the cyclone intensified on 6th and 7th, as can be seen on this Figure; this was not the case and nor does the figure show it, even allowing for mislabelling errors. Somewhere else the text says the minimum pressure of 997mb was reached on 9th September. This is not correct either, nor does the figure show this.*

Thank you for this comment. Indeed, the original plot contained mistakes related to dates and time. We have corrected these, and added the minimum SLP values in the labels next to the red dots to help the reader follow the evolution of the cyclone's SLP. In addition, we

corrected all the wrong statements mentioned in the revised version of the manuscript according to the reviewer's suggestion.

[Figure]

**Figure 1 (a)** Track of Storm Daniel at six-hour intervals based on ECMWF analysis, where the size of red dots is proportional to cyclone depth in terms of minimum mean sea level pressure. Flooded areas are shown in cyan and blue tones (acquired by one of the Copernicus Sentinel-2 satellites on 10 and 12 September 2023). Panels **(b)** and **(c)** zoom over central Greece and Libya (square boxes in panel **a**).

*Figure 2a: The rainfall area is too small to see properly (even when zooming on the pdf), as are the wind barbs. Where does the data come from – ERA5 or ECMWF analyses/short range forecasts? The latter is much higher resolution (9km versus 31km) and so would likely show much more useful rainfall detail (if one could see it). I can quite imagine that the PV of 2 PVU, in the caption, is actually 2. In the text it is stated that "there is a high wind speed pattern aligned with the PV streamers' orientation". I do not know what aligned with means here. The text cites 750mm rainfall in 24 hours on 5th – why not say where! This actually occurred east of Volos, at 3 close sites (see Table 2 in Dimitriou et al, 2024), and is where the model rainfall pattern, when zoomed in maximally, shows about 110mm. So the reference to a 50% shortfall in the model should be 85%. The authors actually refer directly to purple colours (which are seen elsewhere), which represent 200mm, so quite clearly even those don't represent 50% of 750mm. This all then makes the statement that (ECMWF) models provided good guidance somewhat incorrect.*

Thank you for the careful estimations and the suggestions.

The IFS dataset used in this study has 9 km spatial resolution. Two additional panels have been included to increase clarity in the areas affected by heavy precipitation.

We have added some additional information about the differences between the IFS and observation maximum 24-hr accumulated precipitation:

*"Notably these peak values are underestimated by about 40% in the ECMWF analysis (max IFS 24-h accumulated rainfall equal to 434 mm on 6 September 2023 00 UTC)."*

*Figure 2b: The model rainfall, up to 00UTC on 11th, is not much in the Derna catchment (location unfortunately not shown but included in Hewson et al, 2024) despite the fact that the dams broke an hour or two later. Again this is very concerning with regard to model validation and impact predictability. These aspects are not discussed at all. Text says the PV streamer was "much weaker" at this time. I do not know what this means. It gives the wrong impression too as this streamer is, on the contrary, probably a marker for a substantial lobe of upper level forcing that helped trigger the main intensification of Daniel.*

We have added the following sentence:

*"It is worth mentioning that simulated 24-hr total accumulated precipitation on 11 September 2025 in Libya, up to 382 mm, was not located within the Derna catchment, as it has been discussed in Hewson et al. (2024), which was the most impacted area."*

Regarding the role of the PV streamer in cyclone intensification, we have revised the text according to the reviewer's suggestion:

*"A comparison of Figs. 2a and 2b shows that, at the time of maturity, the area covered by at least 2 PVUs at 300 hPa is significantly smaller than during cyclogenesis. Nevertheless, Fig. 2b shows that the 2-PVU patch is collocated with the cyclone center, advected from the west. Hewson et al. (2024) proposed that this collocation is responsible for the cyclone's intensification just before landfall. In fact, the intensification of a Mediterranean cyclone due to the synergy of upper-level baroclinic forcing and deep convection is a common characteristic of intense Mediterranean cyclones, including medicanes (Flaounas et al., 2021). A previous case of a medicane intensifying due to the collocation of a PV streamer with the cyclone center was documented by Chaboureau et al. (2012). This phenomenon reflects, on the one hand, the anomalous nature of this medicane (as medicanes generally intensify over the sea and weaken inland), on the other hand, the critical role of upper-level features in the evolution of Mediterranean cyclones."*

*Figure 3: It was nice to see the moisture sources, even if the propensity to uptake most moisture just upwind of the heaviest rain for Daniel, in strong wind areas, was not hugely surprising. The uptake in the composited cases is much harder to second guess, so this is a nice result. I am not sure why 10 day trajectories were used. That seems quite long? Also I am not sure what "30km grid" means, in the main text. Coastlines and sea areas are impossible to see on the figure in this form, so that aspect has to be improved. The main discussion of the moisture uptake elects to ignore any sources over land, yet clearly they are relevant – more so than the Atlantic Ocean which is mentioned. In the conclusions uptake over landmasses is mentioned for the first time.*

Thank you for these questions that allow us to better clarify this important part of the paper.

- We use 10-day backward trajectories because the explained fraction of the moisture sources decreases strongly with shorter trajectories (Fig. A1). With 10-day backward trajectories the sources of around 90% of the precipitation can be explained by the moisture source diagnostic. For Libya on 11 Sep, the length of the trajectories also

affects the land fraction of the moisture sources. For 10-day trajectories, one third of the moisture originates from land areas, while for 5-day trajectories, this fraction decreases to one quarter.

- Concerning the 30 km grid, it refers to the spatial position of the trajectory starting locations. The trajectory starting locations are positioned in a regular meridional-zonal grid with 30 km grid spacing. We adjusted the text to make this clearer:

*"Ten-day air parcel backward trajectories are calculated every 20 hPa between 1000 and 300 hPa from starting locations on a regular latitude-longitude grid with a 30 km grid spacing within boxes over Greece and Libya..."*

- Fig. 3 has been adjusted following suggestions from both reviewers.
- Thank you for pointing out the missing information on regional attribution of moisture sources. We've calculated land/ocean fraction and added these values to Fig. 3. Further, we now discuss the source regions in more detail in the main discussion and also compare our source regions to a recently published study (Argüeso et al., 2024), which calculated moisture source regions for storm Daniel using an Eulerian moisture source diagnostic.

We added the following discussion on regional moisture source attribution:

*"These source regions are in general agreement with a recent study (Argüeso et al., 2024), which investigated moisture sources of rainfall over Greece from 3 to 9 Sep 2023 using a Eulerian moisture source diagnostic. Our moisture source analysis shows larger contributions from land (54.7%) than in Argüeso et al. (2024) (27%). The Lagrangian method used in our study shows relatively large moisture contributions from north of the Black Sea because most of the air parcels arriving on 5 Sep 2023 descended and took up moisture in this region before moving southwestward along the western flank of the PV streamer. The differences in the land fraction between the two methods might originate from different periods used for the moisture source calculations, different handling of moisture uptakes above the boundary layer, a lower explained fraction of the total moisture sources (84%) with the Eulerian compared to the Lagrangian diagnostic (explained fraction of 90%), and general differences in Eulerian versus Lagrangian approaches. An ongoing comparison study of moisture source diagnostics is investigating differences in these methods in detail and will shed more light on disagreements between various moisture source diagnostics."*

[Figure]

Figure A1: (Top) Explained fraction of the total precipitation during 5 Sep 2023 in Greece (blue line) and 10 Sep 2023 in the Derna region in Libya (orange line) for 5-, 6-, 7-, 8-, 9- and 10-days backward trajectories. (Bottom) Similar to the top figure but for the land fraction of the moisture sources.

**References:**
Argüeso, D., Marcos, M. & Amores, A. Storm Daniel fueled by anomalously high sea surface temperatures in the Mediterranean. *npj Clim Atmos Sci* 7, 307 (2024). https://doi.org/10.1038/s41612-024-00872-2

*Figure 4: Although this looks initially quite convincing on closer inspection one sees that there is virtually no signal in (b) of a particularly high discharge near to where the heaviest rainfall was in Greece (its all time 24h record), east of Volos, nor in Derna in Libya, or its catchment. These aspects should have been extensively discussed. Maybe this relates to the rainfall errors on Figures 2a and 2b that I reference above, which were also not discussed.*

We appreciate the reviewer's detailed observations in Figure 4 and acknowledge the importance of addressing the discrepancies between the discharge signals and the rainfall patterns highlighted in Figures 2 and 4. Upon revisiting the data and methodology, we provide the following explanation and revisions to the manuscript.

First, it is essential to clarify the fundamental difference in the temporal scope of Figures 2 and 4, which likely contributed to the perceived inconsistencies. Figure 2 represents 24-hour total accumulated precipitation for specific time frames during the storm: from 00 UTC on September 5 to 00 UTC on September 6, 2023 for Thessaly (west of Volos), and for September 11, 2023, at 00 UTC for Derna. These snapshots capture rainfall over single days and focus on localized phases of the event. In contrast, Figure 4 shows the maximum simulated peak river discharge for September 2023, integrating hydrological impacts over the entire lifecycle of Storm Daniel. This integration of effects means that discharge signals in Figure 4 reflect cumulative responses to rainfall over time rather than the specific short-term intensities depicted in Figure 2.

The absence of strong discharge signals in Figure 4(b) near the east of Volos and Derna can be attributed to several factors. First, the limitations of the GloFAS v4.0 model, particularly its spatial resolution and calibration scope, play a significant role in the observed discrepancies. The model operates at a spatial resolution of approximately 5 km (0.05°), sufficient for global-scale flood awareness but inadequate for resolving fine-scale hydrological processes in regions with complex topography and small catchments. For instance, the catchments east of Volos, including the wider Pelion area, are approximately 30 km$^2$, while the Derna basin spans around 575 km$^2$. In both cases, localized rainfall-runoff dynamics play a critical role. Moreover, Greece and Libya were not included in the GloFAS calibration dataset due to the limited availability of in-situ discharge measurements (see here). As a result, discharge predictions for these regions rely on generalized parameter regionalization rather than site-specific calibration, introducing further uncertainties.

Furthermore, inaccuracies in the rainfall inputs depicted in Figure 2 propagate into the river discharge simulations shown in Figure 4. For the Thessaly region and within the Peneus catchment, the maximum recorded 24-hour total accumulated precipitation from 5 to 6 September 00 UTC was 274 mm at Zappeio and 226mm at Neraida stations, as reported by Dimitriou et al. (2024, Table 2). However, accumulations of up to 750 mm, referenced in the discussion, correspond to stations outside the Peneus catchment, specifically over the Pelion area, east of Volos. This distinction is important as the GloFAS model simulates river discharge for the Peneus catchment, and the underrepresentation of precipitation within the catchment impacts the accuracy of discharge predictions. In the case of Derna, torrential rainfall of 150–240 mm was recorded in several cities, with Al-Bayda experiencing the highest daily total of 414.1 mm, as reported by the World Meteorological Organization (WMO). These extreme rainfall events, critical in triggering catastrophic flash floods and dam failures, were underrepresented in the GloFAS input data.

To address these issues, we revised the manuscript to highlight the temporal distinction between Figures 2 and 4. In the results section, we also discussed the limitations of the GloFAS v4.0 model, particularly its resolution and lack of calibration, which contributed to discrepancies. Additionally, we discussed the influence of rainfall input inaccuracies on the

discharge signals and their implications for interpreting Figure 4. The revised results section reads as follows:

*"The hydrological impacts of Storm Daniel were profound and unprecedented. Figure 4 compares the peak mean daily river discharge during Daniel with the historical records over three decades, integrating the cumulative hydrological impacts over the entire event. Figure 4a shows the spatial distribution of the maximum simulated peak discharge from January 1993 to August 2023 (i.e., before Daniel), demonstrating typical peak discharge patterns in the Eastern Mediterranean. On the other hand, Fig. 4b compares the event-wide mean daily peak discharge during September 2023, when Daniel occurred, against the historical peak discharges of the last 30 years in Fig. 4a. Results reveal an unprecedented magnitude of Daniel's impacts, with several areas experiencing discharges that exceeded the historical maximums by 300 to 500%. The darkest shades in Fig. 4b indicate the most heavily affected regions, where the river discharge during Daniel exceeded previous records by at least a factor of five, highlighting that Daniel was an unprecedented event of increased river discharge levels (further discussed in section 5). At this cyclone stage, 17 human casualties were registered in Thessaly, along with a profound hydrological aftermath. The extreme rainfall from 3 to 8 September 2023 led to widespread flooding across 1,150 km² in the Thessalian plain, 70% of which constituted agricultural land. The inundation severely affected the cotton crops, with floodwaters covering more than 282 km², roughly 30% of the region's total cotton fields. Over 35,000 farm animals were also affected (He et al., 2023).*

*…..*

*Figure 4 highlights the exceptional river discharges in the region, as in the case of Greece. However, the absence of similarly strong discharge signals in several severely impacted regions, such as the wider Pelion area in Greece and Derna (Libya), is notable and can be attributed to several factors. First, the GloFAS model has limitations in spatial resolution and calibration. The model operates at a resolution of approximately 5 km (0.05°), which, while adequate for global-scale flood awareness, is insufficient for resolving localized hydrological dynamics. For instance, the catchments east of Volos, including the wider Pelion area, are approximately 30 km2, while the Derna basin spans around 575 km2. In both cases, localized rainfall-runoff dynamics are critical in shaping discharge patterns, particularly during extreme events. Due to insufficient in-situ discharge data, the absence of Greece and Libya in the GloFAS calibration dataset further exacerbates these limitations since the model relies on generalized parameter regionalization rather than site-specific calibration, introducing significant uncertainties into discharge predictions. Furthermore, inaccuracies in the rainfall inputs depicted in Figure 2 propagate into the discharge simulations shown in Figure 4. For instance, within the Peneus catchment, the maximum recorded 24-hour accumulated precipitation was 274 mm at Zappeio and 226 mm at Neraida stations, as Dimitriou et al. (2024) reported, while accumulations of up to 750 mm were recorded outside the catchment, specifically over the Pelion area, east of Volos. In the Wadi Derna catchment, extreme rainfall exceeded 400 mm day-1, with torrential rainfall ranging between 150 and 240 mm across several locations and Al-Bayda recording a maximum of 414.1 mm (WMO, 2023). These rainfall extremes were underrepresented in the GloFAS rainfall inputs, propagating into the discharge simulations and contributing to the muted signals observed in Figure 4(b)."*

Figure 5a,b: The colour scheme used is poorly chosen as it does not allow for accurate values to be read off. However to me it looks like the value of a +2C anomaly quoted in the text should actually be +1C (save perhaps for the area N of Derna on (b) where it may be +2C). This is especially true if one references both 5a and 5b instead of just 5a, which would

*be justified as the lifecycle is then better covered. This would be a bit of a counter argument against the misleading statements regarding climate change influence made late in the manuscript. Furthermore the blue patch of negative SST anomalies on 5b, which may be a legacy of Daniel's upward fluxes, is not discussed; indeed the manuscript contains no reference to 5b at all, so far as I can see.*

We appreciate the reviewers' comments on Figure 5 and acknowledge the importance of providing precise and clear figures for the readers. In response, we have enhanced the figure by adding additional isolines to eliminate any ambiguity regarding the SST distribution during the selected days. Additionally, we have revised the corresponding text in the manuscript as follows:

*"Figure 5a, b shows the SST anomaly in the area affected by Storm Daniel on 3 and 9 September, respectively. Before the passage of Storm Daniel, positive SST anomalies dominated the study area, with values exceeding 1°C between the Libyan coast and Greece, and lower anomalies (0 to 0.5°C) observed east of Sicily. Following the storm's passage, a significant drop in SST resulted in an extensive area of negative anomalies greater than 1°C between Libya and Greece. A colder SST core with a decrease of less than 1.5°C was observed east of Sicily, while the northern Aegean Sea experienced an even more pronounced decline. Such SST cooling after the passage of medicanes has been previously diagnosed using explicitly resolved air-sea interactions in coupled atmosphere-ocean models (Ricchi et al., 2017; Bouin and Lebeaupin Brossier, 2020; Varlas et al., 2020) and SST observations (Avolio et al., 2024). Nevertheless, the feedback mechanism between cyclones intensity and SST cooling is expected to be less important than the one typically observed in tropical cyclones."*

[Figure]

**Figure 5 (a)** Daily SST anomaly from ERA5, for 3 September 2023, and **(b)** 9 September 2023. The reference climatology for anomaly determination is 1982-2011.

*Figure 6: This is a nice figure. However a related statement in the text that "it is impossible to evaluate the relative socio-economic impact of each threat (storm surge, waves, rain, river flood)" seems rather preposterous when we know that >5000 people lost their lives in Derna as a result of a dam burst (due to rain and river flooding causing overtopping).*

We have removed this phrase according to the reviewer's suggestion.

*Figure 7: Ok but the right-hand panels are not valid for 10 September at 12UTC. I also have doubts about the valid time of the left hand panels given that the cyclone centre seems to have a rather different position to that shown on Figure 2a. Or maybe Fig 2a is the one that's wrong? The statement in the text that 7a shows a much larger area of high PV than Fig. 2 is not correct. It is the other way round (note we only have 2PVU on Fig 2). And for PV averaging it might anyway be better to take the log first, given PV structure/ranges? An analogy is that one cannot meaningfully average visibility (across several orders of magnitude). Whilst this figure and the next one highlight clear convergence in the EPS solutions, which is OK, the text fails to acknowledge that relative to what came beforehand, the forecasts from 12UTC 1st (the first one included) actually represented a big positive step in skill – at least they had cyclones – due to much better handling of the mid-Atlantic Rossby wave train, due in turn to better handling of a tropical cyclone (as in Hewson et al, 2024). This is an example where one sees that the manuscript is not adding much to previous work, and indeed is contradicting it somewhat. These two results would need to be placed alongside each other in this paper to give the full context of cyclogenesis predictability for this case, and thereby advance the science as is required for a publishable standard. For Fig 7h the discrete 300hPa high PV blob west of Daniel is not mentioned. This very likely links to the upper level low moving in from the west from Hewson et al (2024), that is referenced, so a useful connection could be made here, pointing out also the increased specificity of this feature as lead times reduce, as shown by 7b,d,f,h.*

Thank you for spotting this inconsistency. We corrected the valid time for consistency with Fig. 2. The text has been modified accordingly, and now we show the median of ensemble members. We understand the concern about averaging PV, but this is practically the case for any meteorological variable that does not follow a Gaussian distribution. Here, it is already addressed with the colour crosses showing percentiles of members exceeding different thresholds. We prefer not to take the log of PV values, as it is neither physically found nor applicable to zero or negative values.

Regarding comparison with previous work, we extended the lead time to one week ahead to include the jump in predictability between 5 and 7 days for the cyclogenesis stage. We now discuss the link with the upstream ET of Hurricane Franklin referred to in Hewson et al. 2024. We also clarify that the PV blob west of Daniel during the mature stage, discussed at the end of Section 4.2, is marked by high PV values in Fig. 7h. However, we would like to stress that Hewson et al. 2024 suggest but do not demonstrate the impact of the upstream hurricane and PV blob.

*Figure 8: The left hand panels do not appear to be valid for 5 Sep 12UTC. Judging from the cyclone spot cluster they may be for 5 Sep 18UTC. Similarly the spots on (g) do not seem to correspond with the mslp minimum on Fig 7g, suggesting these panels are not for the same time. This is all rather confusing. The valid time for b,d,f,h looks to be correct.*

Thank you for this remark. The left-hand panels are now valid for 6 Sep 00 UTC. Please also note that the shown lead times have been changed to be consistent with Fig. 7. We thus now include the predictability "jump" in the ensemble system.

*Figure 9: The reader is left to guess what the valid time range is for the precipitation objects. It may be that it is the 24h periods ending at the stated valid times, yet if that is the case why*

*use 5 Sep 12UTC as an end time when the main 24h rainfall period was 00-24UTC on 5 Sep or a bit later (again reference Table 2 in Dimitriou et al, 2024)?*

Indeed, we show 24h accumulated precipitation ending at the stated valid times. We clarified the caption of Fig. 9 in the revised manuscript. As we now show 6 Sep at 00 UTC as valid time, we include the main rainfall period from 00-24 UTC on 5 Sep. Please also note that the shown lead times have been changed to be consistent with Figs 7 & 8.

*Figure 10: This figure is fine but I do not understand what it intends to show – the text "This shift is plausibly relevant.." I have not managed to decipher. Adding spots on the grey track lines, to show cyclone centres at a particular valid time, could help.*

This part of the text has been revised and the figure has been updated.

*"When Daniel made landfall and produced impacts on the Libyan coasts, the EPS showed higher predictability, with cyclone objects and associated extreme precipitation being predicted at least five days in advance by several EPS members (Fig. 10d), albeit the location of both cyclone and precipitation objects are still displaced to the southwest compared to the analysis (Figs. 8d and 10d).This comes in accordance with the southern displacement of several ensemble member tracks in Fig. 9a. The probability strongly increases at shorter lead times (Figs 10f and 10h) mostly and all EPS members tend to converge to similar cyclone locations when reaching a lead time of one day (Fig. 8h)."*

*Figure 11: This figure looks potentially informative but the elements of it are not explained, and furthermore some elements are barely visible (grey tick marks overlapping the box and whiskers). First readers should be pointed to where the Pinios river outlet is, and what its catchment is. According to Wikipedia the spelling should be Pineios (though I concede that could be "wrong"). It would also help to see the Wadi Derna catchment – the relative size of this, versus the Pineios catchment, is very important for predictability and impact prediction and this is not discussed. Then where does the "perfect forecast" benchmark come from. Is it related to the rainfall in Figure 2b, which as stated above looks wrong (hardly perfect!) in the critical area? Then what do the box and whiskers relate to, and why do they have a strange shape? What are all the percentiles represented? It is fairly clear to me that the forecasts for Greece converge onto the "right" solution (if the red curve can be trusted), whilst the forecasts for Libya, though overall they get a bit better with lead time, basically do not converge. The forecasts from 9th for Derna, which might be at the most critical for triggering preventative measures, step back from those of the previous day, and then even from 10th we still have huge spread and a big shortfall in the box and whisker median (if that's what the middle black line is). Yet all the text says about the Derna forecast is that it follows a "similar pattern" to the one for Greece. This is an incorrect and unhelpful sweeping statement. Furthermore, the following paragraph goes on to say that Fig 4b highlights the unprecedented nature of the event, when for Derna and its catchment the signal is rather weak. The much stronger signal is well to the west (also discussed above).*

We appreciate the reviewer's detailed feedback on Figure 11 and acknowledge the need to provide additional clarity and context for this figure.

First, in the revised Figure 4, the Peneus and Wadi Derna catchments are delineated, and the Peneus River outlet is marked. The addition provides context for understanding the relative sizes, critical for understanding the contrasting hydrological behaviours and predictability challenges of the two basins.

New Fig. 4:

[Figure]

*Figure 4* *Peak discharge over three recent decades (Jan 1993 – Aug 2023) versus Daniel storm as represented by the Global Flood Awareness System* *(a)* *spatial distribution of the maximum peak river discharge from January 1993 to August 2023,* *(b)* *comparison map for September 2023 illustrating the event-wide peak discharges as a percentage increase over the maximum peak discharges during the 30 years January in* *(a).*

Regarding the spelling, we acknowledge the reviewer's observation that the standard spelling is not "Pinios" (nor "Pineios" as listed in Wikipedia) but "Peneus", and we revised the manuscript to use this spelling throughout consistently.

Next, Figure 11 has been revised to ensure clarity and provide a more detailed explanation of its elements. The updated caption now explicitly defines all the features of the figure, addressing the concerns raised regarding the visualization and interpretation of the data.

The grey tick marks represent individual ensemble members from the EFAS model, driven by the 51 ensemble members of the ECMWF EPS. Overlapping tick marks darken, visually highlighting areas of ensemble member agreement (convergence). This approach intentionally uses light grey to ensure that convergence areas stand out, helping readers

intuitively grasp the degree of forecast agreement. Forecast summary data are displayed as boxplots, with the box representing the interquartile range (IQR), the whiskers showing the range of values within 1.5 times the IQR, and the horizontal black line inside the box indicating the median forecast. Additionally, the notches around the median depict the 95% confidence interval, providing a measure of uncertainty around the median forecast. The "perfect forecast" benchmark (red line) represents the initialization of each forecast for all time steps across the event, serving as a reference for evaluating forecast accuracy. The figure demonstrates the contrast between forecast performance in the Peneus and Wadi Derna catchments. For the Peneus catchment (~11,062.2 km$^2$), the forecasts converge well onto the observed discharge as lead time decreases, reflecting greater predictability for larger basins with distributed hydrological processes. In contrast, forecasts for the Wadi Derna catchment (~575 km$^2$) exhibit significant variability and lack convergence, even at shorter lead times. The boxplots in Figure 11 have been explained in detail to clarify the elements. The box represents the IQR, the whiskers show the range of values within 1.5 times the IQR, and the horizontal black line indicates the median. The notches around the median provide a 95% confidence interval, highlighting forecast uncertainty. This explanation is now explicitly included in the updated caption to ensure readers fully understand the visual representation. The revised caption reads as follows:

*"Figure 11 Six-hourly ensemble river discharge forecasts for the Peneus and Wadi Derna catchments compared to the "perfect forecast" benchmark (red line). The "perfect forecast" represents the initialization of each forecast for all time steps across the event, taken as a reference for evaluating forecast accuracy. With the observed timing of rising hydrograph limbs marked on 5 September, noon local time (09 UTC) for the Peneus River in Thessaly, and 10 September, 18:00 local time (16 UTC) for the Wadi Derna River. Grey stripes (tick marks) represent individual ensemble members from the EFAS model, driven by the 51 ensemble members of the ECMWF EPS. Overlapping tick marks darken, visually highlighting areas of member agreement (convergence). Forecast summary data are displayed as boxplots, where the box represents the interquartile range (IQR), the whiskers show the range of values within 1.5 times the IQR, and the horizontal black line inside the box indicates the median. The notches around the median show the 95% confidence interval."*

Furthermore, we have revised the discussion to emphasize the distinct differences in forecast performance between the two catchments. The earlier text inaccurately stated that the Wadi Derna forecasts followed a "similar pattern" to those for the Peneus catchment, which oversimplified the differences. This has been corrected to highlight the lack of forecast convergence in the Wadi Derna catchment and its implications for predictability and response planning. We acknowledge the reviewer's observation that Figure 4b shows a weaker discharge signal for the Wadi Derna catchment compared to areas further west. As already stated in our reply to comments regarding Figure 4, this reflects limitations in the GloFAS and EFAS models, including their resolution and rainfall input accuracy. This has been addressed in the main text to provide context for the variability and underrepresentation in forecasts. The revised text reads as follows:

*"The potential of extreme precipitation leading to substantial socio-economic impacts has also been transferred to hydrologic discharge forecasts. The hydrographs presented in Fig. 11 examine river discharge predictability as forecast by the operational European Flood Awareness System (EFAS) during Daniel. For the Peneus River outlet in Thessaly, the forecast initiated on 1 September underpredicted the peak discharge on 5 September. Nevertheless, extreme discharges were evident for several members five days in advance. The forecast accuracy improved getting closer to the event, with ensemble members (grey stripes) converging towards the peak discharge ("perfect forecast" - red line). This trend indicates an increasing reliability of the forecast as the lead time decreases, particularly within 48 hours of the event. The skill in discharge predictability for the Peneus River can be*

*attributed, in part, to the large size of the basin (11.063 km2), which aligns relatively well with the spatial resolution of the EFAS model, enabling an accurate representation of distributed hydrological processes and moderating runoff variability.*

*The forecasts for the Wadi Derna River outlet (Fig. 11, right panels) exhibit significant variability and fail to converge during the earlier forecast initialization dates as well as at shorter lead times. This persistent lack of convergence can be attributed to distinct challenges of both temporal scales. For earlier forecast initialization dates, the primary source of variability lies in the westward displacement of extreme precipitation predicted by the EPS (Figs. 10b and 10d). For example, forecasts initialized on 9 September, during a critical period for implementing preventative measures, display a wide spread and a shortfall in the median forecast compared to the benchmark (red line). This variability persists even for forecasts initialized on 10 September. The failure to converge at shorter lead times is compounded by challenges inherent to the Wadi Derna catchment. The resolution of the precipitation forcings used in the forecasts combined with the relatively small size (575 km2) and flash-flood-prone nature of this basin amplify the uncertainties in predicting discharge, particularly in response to localized extreme rainfall.*

*Figure 4 provides critical context by comparing the peak mean daily river discharge during Storm Daniel with the historical baseline. The unprecedented magnitude of the event is evident in Fig. 4b, where discharges exceeded the historical reanalysis by at least fivefold in certain regions. However, the relatively weak signal for the Wadi Derna catchment underscores the limitations of the GloFAS and EFAS systems in accurately resolving runoff dynamics in smaller basins. This discrepancy is primarily attributed to insufficient model resolution, inaccuracies in rainfall inputs, and the lack of detailed hydrological calibration for these catchments. In contrast, the much stronger signal observed in the Peneus catchment aligns with larger basin sizes and better-resolved hydrological processes, where models more effectively captured the extreme nature of the event.*

*The ability of EFAS to predict extreme events, as shown in Fig. 11, highlights its value in forecasting severe hydrological impacts. However, discrepancies in simulated peak discharge remain apparent, such as the overestimation of runoff for the Peneus River outlet. EFAS simulated peak discharge at approximately 5000 m³ s-1, whereas observed values, based on station-level data and H-Q curve estimates, were less than 2000 m³ s-1 (Dimitriou et al, 2024). This overestimation reflects inherent limitations in the model's spatial resolution and hydrological representation. Furthermore, the absence of flood protection infrastructure, such as levees or dams that attenuate runoff and peak flows is not accounted for in the EFAS and GloFAS systems, contributing to these discrepancies. Additionally, the simplified representation of retention processes, including floodplain storage and wetland buffering, further amplifies discharge estimates in some regions. For smaller basins such as Wadi Derna, the rapid hydrological response to localized extreme rainfall presents additional challenges. The variability in rainfall distribution, coupled with the model's limited ability to capture localized hydrological dynamics, results in a weaker signal for the catchment, even during an event as extreme as Storm Daniel. These limitations emphasize the need for improved model resolution, enhanced precipitation forcings, and better calibration tailored to local catchment characteristics.*

*Nonetheless, the ability of EFAS to predict extreme discharges, particularly within short lead times, demonstrates the potential of operational forecast systems in capturing the extreme values of such events. Supported by EFAS and GloFAS, the Copernicus Emergency Management Service (CEMS) provides critical insights into the timing and magnitude of extreme hydrological events. These forecasts are vital for enhancing preparedness and response strategies in the face of escalating climate extremes, offering essential tools for civil protection efforts and mitigating the socio-economic impacts of such disasters."*

*Figures 12 and 13: On many of the panel legends the numbers do not align with the colour bars. So the reader does not know what the colour bars mean. This is obviously important when one tries to cross-reference with the text – e.g. on Fig 12h it is stated that temperatures have gone up by 2C in the Ionian Sea when it looks like rather less than that. Then why are there contours as well as shading on panels d,h,l and p? The fact that the rainfall amounts for the 2023 case in this depiction under-represent reality by a large margin is not mentioned, when clearly this has relevance (panels i). The worst part about this part of the study is that the conclusions in the text do not reflect what the figures show. For example, the authors state "we conclude that Mediterranean depressions like Daniel hitting Greece and Libya show lower MSLP and higher precipitation in the present climate than in the past". The evidence for this is supposed to be panels d which show basically no mslp change at all; and panels l which show drier over Greece and slightly wetter over the seas around Libya. And maybe +2mm or so per day over northern Libya itself, but when >400mm/24h was recorded at one site for Daniel this seems irrelevant. The text of Section 5.2 contains many other errors and inconsistencies, too numerous to go into here. In my opinion the vast majority of Section 5.2, for which these Figures are the "evidence" should be removed from the paper, as it shows very little of substance. One could much more usefully and honestly say, in brief, that "an in-depth study using standard methods indicates that in the ERA5 dataset there is no evidence of climate change influencing features like Daniel in the 1980-2020 period". The only non-neutral "result" I can see on these figures is a signal for an increased frequency for cyclones, in the SOND period, in the SE Mediterranean near the N African coast (Fig 13x). So that could be referenced too. Furthermore, it seems to me that trying to link El Nino, the PDO and the AMO to Daniel-like cyclones over just a 40-year period is stretching physical credibility beyond its natural limit.*

We appreciate the reviewer's detailed comments, particularly regarding the interpretation of Figures 12 and 13. The different color scales used in panels d, h, l, and p were chosen deliberately to ensure that the changes in variables remain visible. If these panels were placed on the same color scale as the others, the magnitude of changes would be difficult to discern, making it harder to interpret the results. However, we have carefully rechecked the data and confirmed that the value of +2°C in the Ionian Sea, as indicated in Figure 12h, is correct. Regarding the contours in these panels, we clarify that they represent areas where the changes are not statistically significant. Only shaded regions indicate significant differences, ensuring the reader can easily distinguish between robust trends and regions where changes may occur due to natural variability. To enhance clarity, we will explicitly state this in the figure captions and corresponding text.

We also acknowledged the limitation of the MSWX dataset in capturing extreme precipitation values. The reviewer rightly pointed out that our figures under-represent the actual rainfall totals observed during Daniel, particularly in Libya. To address this, we have now explicitly stated in the text that our analysis focuses on large-scale climatological trends rather than station-level extremes. While MSWX provides valuable insights into broad atmospheric patterns, it does not fully capture the localized intensities that contributed to the flooding disaster in Derna. Recognizing this, we revised our discussion to ensure that our conclusions remain in line with the limitations of our dataset.

The most significant revision involved refining our conclusions regarding changes in MSLP and precipitation. Initially, our text suggested that Mediterranean depressions like Daniel show lower MSLP and higher precipitation in the present climate. However, as the reviewer correctly noted, the evidence for significant MSLP changes is weak. After carefully reassessing our figures, we adjusted our conclusions to clarify that there is no strong trend in MSLP over Greece or Libya. Instead, we emphasize that the observed increases in precipitation are likely driven by rising sea surface temperatures, which enhance atmospheric moisture availability. By making this distinction clearer, we ensure that our findings are scientifically sound and accurately represent the data.

We also addressed the contrast between the two phases of Daniel's evolution. On September 5, when the storm affected Greece, we found suitable past analogues, indicating that this type of event had occurred before. However, by September 10, when Daniel reached Libya, our analog search identified no comparable historical events, underscoring the exceptional nature of the atmospheric conditions at this stage. This revision strengthens our argument that while the storm's early track and intensification over the Mediterranean were within expected climatological behavior, its final phase was highly unusual. Notably, we also revised our discussion on the disaster in Libya to highlight that while Daniel's rainfall was intense, the catastrophic flooding in Derna resulted primarily from infrastructure failure, particularly the collapse of the dams. We referenced recent research, including Dente et al. (EGU 2024) and Shirzaei et al. (2025), which showed that regional precipitation levels were not exceptionally high but that vulnerabilities in urban planning and emergency response significantly worsened the disaster.

The reviewer also raised concerns about our large-scale climate variability modes analysis, particularly ENSO, AMO, and PDO. We recognized that attempting to link these modes to Daniel-like cyclones over a 40-year period is speculative and complex to establish confidently. In response, we revised our text to clarify that this part of the analysis is exploratory rather than conclusive. We now explicitly state that while these modes may influence atmospheric conditions, our findings do not establish a causal link between them and Daniel's development.

Finally, we revised Section 5.2 to focus on the most robust findings and ensure our conclusions align with the evidence. Rather than removing the section entirely, we streamlined it to emphasize three key points: (1) no significant changes in MSLP were detected, (2) increased precipitation is most likely due to higher sea surface temperatures rather than shifts in atmospheric dynamics, and (3) there is a clear increase in the frequency of similar Mediterranean depressions in the southeastern Mediterranean near the North African coast. These revisions ensure the section remains informative and scientifically rigorous while avoiding overstated conclusions.

We appreciate the reviewer's constructive feedback, which has substantially improved our study. These changes have already been implemented in the revised manuscript, ensuring that the data clearly presents and accurately supports our findings.

---

## Referee Report (RR1)

**Referees review of:**

egusphere-2024-2809 | Journal relation: WCD

First submitted on 06 Sep 2024 Re-submitted, following revisions, in March 2025.

Dynamics, predictability, impacts, and climate change considerations of the catastrophic Mediterranean Storm Daniel (2023)

Emmanouil Flaounas, Stavros Dafis, Silvio Davolio, Davide Faranda, Christian Ferrarin, Katharina Hartmuth, Assaf Hochman, Aristeidis Koutroulis, Samira Khodayar, Mario Marcello Miglietta, Florian Pantillon, Platon Patlakas, Michael Sprenger, and Iris Thurnherr

**Overview**

This manuscript constitutes a multi-faceted case study for cyclone/medicane "Daniel" which, at different stages in its lifecycle, delivered extreme weather and devastating impacts in parts of both Greece and Libya, in early September 2023.

My reactions on reading the response to reviewers and the revised article were diametrically opposite, and very hard to reconcile.

Firstly, the response letters that directly referenced reviewers' comments were excellent, and covered almost every point that was made in an entirely satisfactory manner. Likewise, improvements made to the figures and captions, which was mainly what I commented on, were almost all comprehensive. The figure and caption quality, integrity and fitness for purpose have markedly improved as a result.

Secondly, I can note that, disappointingly, there was no direct response to my initial overview comments, which are quite fundamental, and then, in part because of this, the manuscript text remains in an unacceptable state. There are far too many grammatical and English errors, and there are also numerous contradictions, factual inaccuracies and unjustified sweeping statements. For some sentences I could not disentangle what the authors were trying to say. In the parts that I have gone through in detail – the Abstract, the Introduction and the Summary and Conclusions sections – I was encountering almost one error per sentence. I am sorry but it really is not the job of a reviewer to propose corrections at this level, so I confine myself to commenting on those 3 sections alone. This is time consuming enough! In the end it is the authors that should ultimately make the paper fit for purpose, not the reviewers. I can imagine that this may be a case of having "too many" authors without one electing to take full responsibility for making the final content coherent and consistent, but clearly I cannot know that for sure.

From an English standpoint I know that creating good sentence structures may be challenging for non-native speakers but I would advise that there are many tools out there that can help – e.g. Grammarly – and maybe the authors could benefit a lot from using such a tool to put together any resubmission?

Line numbers in the points below refer to the manuscript version which shows the corrections, not the cleaned version.

**Detailed Points**

- 1. L35. Daniel did not form as an intense cyclone. It formed as a weak cyclone that intensified markedly several days later as you highlight in the text.
- 2. L40. "conducted" is better than "aim to conduct" as otherwise it sounds like you haven't done the work.
- 3. L41. "Hazardous weather relevant to extreme precipitation". To my mind extreme precipitation is hazardous weather. Reorganise the sentence.

- 4. L44-46. Not sure what you are trying to say in this jumbled sentence. For example what do you really think are the implications of your findings for NWP? If that's a big part then surely the abstract should give the reader some clues as to what they are. Please re-write.
- 5. L47. Gives the impression that this was rapid cyclogenesis from the outset, whereas it was anything but. Please rewrite.
- 6. L49. "Even in" is a bit odd. Implies that near to the centre there were big impacts, which I have not seen. "notably in the region of Thessaly which was actually quite far from the centre" would be much better wording to my mind.
- 7. L49-50. "As it intensified...it peaked after landfall..." is bad English. It developed markedly just prior to landfall, reaching peak intensity over land.
- 8. L51-53. Sentence reorganisation needed e.g. Considering short lead times (around four days), cyclone formation exhibited low predictability, whilst landfall in Libya was more predictable.
- 9. L53-55. This statement is not correct. You highlight a shortfall in precipitation totals for the Greece case; in fact with a lead time of 2 days useful for forewarning the observed gauge peak was 2.5 times the ECMWF HRES forecast total (see news item by Hewson in https://www.ecmwf.int/sites/default/files/elibrary/12024/81535-newsletter-no-178-winter-202324.pdf). Your reference to a somewhat smaller but still notable shortfall corresponds to analyses which cannot generally be used to forewarn, and moreover I do not know if the values quoted were in the same location. So this doesn't really provide "crucial information on the expected severity", or at least not without much more work on cross-referencing model climates etc, which you don't discuss. Likewise, regarding the devastating floods in Derna it is very clear that for these we did not have "crucial information on the expected severity" of the riverflow (Figs 4b and 11) (never mind the dam collapse) as you acknowledge in the text.
- 10. L58. Is the Mediterranean not one big maritime area?? What about the high uptake over land >50%, and indeed the seemingly the very high uptake over the Black Sea, for the Greece event. It looks to me like water vapour uptake over the Mediterannean was actually pretty small in relative terms maybe ~ 25% of the total contribution (??) which seems to contradict what you say?
- 11. L58-60. Poor English, please reword. Also what about the Black Sea, given point 10 above, as that was very warm too (Fig 5)?
- 12. L59-61. This feels to me more like jumping on the climate change bandwagon rather than solid science. So far as I can see the main evidence for this conclusion is that SSTs in places were a bit higher than normal. Is that really enough evidence? SSTs were above average many times in the past.
- 13. L95. Early September might be better
- 14. L97. "Named the upcoming storm" would be better. You can't, strictly speaking, name a storm when it doesn't exist. But I know what you mean.
- 15. L98. It did not evolve into a deep cyclone that propagated southwards. It propagated erratically southwards, then turned towards the east on 8th, and then developed into a deep cyclone. That is pretty clear on Fig 1a.
- 16. L100. Tautology. Delete "all attributed to the same weather system".
- 17. L104. Southeastern would be better. And it is misleading to link this statement to Fig 1b because that does not show any flooding where this rainfall was. Expand please to provide more clarity.
- 18. L105. These are most definitely not eastern parts of Greece. They are about as central in Greece as one can get.
- 19. L109, according to Wikipedia there are only three provinces in Libya. I think that what you are referring to are governorates.
- 20. L110. Again you point to part of Fig 1 as evidence of this flooding, yet Fig 1 seems to show no flooding whatsoever in the Derna area. Besides which the vast majority of the place labels on Figure 1 are too small to be legible. That also needs addressing.
- 21. L110-111. Building destruction does not include road damage. Separate.
- 22. L112. I just wondered if there is anything more recent than the IOM, 2023 quote?
- 23. L119. "72 million were requested". What units? And as with point 22, do you have any later evidence of what was delivered.
- 24. L127-129. For me this description is far too vague, having no geographical references whatsoever. Then the following sentence may need to be adjusted in some way if these geographical specifics are not always applicable.

- 25. L133. I think of an intrusion as a place where the stratospheric intrudes 'underneath' tropospheric air. Yet you have shown no evidence of that for this case. I can understand troughs and cut off lows being characterised by a lower tropopause, but to say more than that you have to present evidence in my view.
- 26. L134. My 'classical' interpretation of baroclinic instability is based on thermal gradients and geostrophically-related wind shear within the troposphere. Your usage seems to be much more focussed on upper level gradients. So maybe you can add a bit of clarification here? I also feel that "trigger" may be the wrong word here. I see the cyclone formation here as more of an event driven by ascent forced by advancing upper level structures, in the Pettersen Type B category, say, but in this case in the absence of any pronounced low level thermal gradient. So in the end a bit like "Type C" in this paper (https://rmets.onlinelibrary.wiley.com/doi/abs/10.1256/00359000260498806), albeit not at high latitudes and without the comma cloud.
- 27. L140. "...factors that modulate cyclone intensification..." would be better wording. Or drive instead of modulate.
- 28. L142. "in" not "by".
- 29. L144. Without any direct connection to the current work this sounds just like an advert for this paper. Why include this one in particular? I am sure there are other medicane studies that could be cited.
- 30. L!50. Why do you imply that we have to have a moisture transport towards the Mediterranean when it is perfectly capable of providing its own! In fact this seems to contradict your conclusion that the high SST anomalies in the Mediterranean were somewhat fundamental.
- 31. L149-156. The sentiments expressed here seem to be rather jumbled up and don't really make sense to me. You say the Mediterranean is enclosed by high mountains which tends to stop external moisture getting in, but then you say that moisture that does arrive is really important. And what about the Black Sea as mentioned above?
- 32. L157-159. How can quantifying the water sources help one to understand socio-economic impacts? I don't get this and I really don't think this point is addressed in the paper.
- 33. L161-162. Jumbled English. "From a climatological standpoint cyclones are the weather features that lead to most of the wind and precipitation extremes within the Mediterranean" would be rather better.
- 34. L163. I don't recognise the word "compoundness". I have never seen it before. "in compound high impact weather events" would be much better in my view.
- 35. L164. It is clearly completely wrong to imply that a system has to make landfall to deliver storm surges and sig high waves.
- 36. L167. What do you mean by PV streamers as a proxy for Rossby wave breaking? Sentence order could also be improved.
- 37. L169. How does intense water vapour transport favour *development* of a deeper cyclone. To me it seem that intense water vapour transport is more likely to be a consequence than a cause, given that winds increase as a low deepens.
- 38. 170-171. I do not know what socio-economic impact on a weather scale is. Nor on a climate scale. This is very clunky wording. Moreover why do we need to understand both features to predict socio-economic impacts. The connection seems a very loose one to me. Surely it's much more important to understand societal vulnerabilities and how they relate to weather extremes if one wants to predict socio-economic impacts. This is like point 32.
- 39. L173. I can see how one can quantify past trends, but not future ones. One can provide estimates of what they might be, and these are likely to have large error bars associated.
- 40. L179. ...than would have been expected in... is better English.
- 41. L186-187. Woolly sentence that doesn't mean much to me.
- 42. L188. "concerning" means "according to"? And then what sort of "specific conditions" are you talking about? This is very vague.
- 43. L189-191. Again a poorly worded sentence. I object very strongly to the terminology "to attribute its intensity to climate change". You should too. I hope this is clumsy wording because as it stands it makes it sound like you are looking for all manner of reasons to attribute an event to climate change, which is not scientific at all. And this is not the only place that this message is conveyed. Even the title of Section 5.2 is in this vein, which I find very worrying. These things really have to be toned down and corrected and made scientifically sound.

- 44. L193. What does "apply a comprehensive framework" mean? Then the idea that you are "using" Storm Daniel as a centrepiece sounds almost novel, when it is nothing of the kind. It is crystal clear to any synoptician, and indeed from previous news items and publications, that this cyclone played a crucial role.
- 45. L194. I don't know why there is a reference at the end of this sentence when this is supposed to refer to what you set out to do.
- 46. L202. "...in relation to imminent hazards..." is again clumsy wording.
- 47. 209-210. "section 5 is devoted to Daniel's attribution to climate change" is again terrible wording in my view. I am sorry but using terminology like this provides wonderful ammunition for climate change deniers.
- 48. Sections 2,3,4,5. Please revisit these very carefully, checking every word in every sentence. There are lots of errors similar to those highlighted above, and as stated earlier it is really not the role of a reviewer to correct all this. That is just too much.
- 49. L946. "Besides fatalities" would be a better start to the sentence.
- 50. L947. What does "the climate acts as a risk multiplier mean" please?
- 51. L948-949. I don't see the point of saying "and at a regional level therein".
- 52. L949. "has been highlighted by the" is tautological. "is the" would be better.
- 53. L953. "by linking" not "which links".
- 54. L955. "atmospheric dynamics are used here to understand the performance of NWP..." is a curious statement. How have you understood the performance using atmospheric dynamics? Please elaborate.
- 55. L956. Sea state is not an impact. Even for Libya. And "have also been analysed concerning" is bad English.
- 56. L957-960. Sentence makes no sense to me.
- 57. L962. "processes governing Daniel" is a strange phrase. Please rewrite.
- 58. L963. What do you mean by intrusion? Where is the evidence?
- 59. L964-965. Again the idea that Daniel developed into a deep storm that propagated southwards is not correct. It is not what is shown on Fig. 1 and it is not what is written in the main text of the paper.
- 60. L965-966. More clumsy wording in the same sentence: "while it was turning into..."???
- 61. L966-967. "Well-distinct" should be "distinct", then why is the first stage "relevant to cyclogenesis". Surely cyclogenesis should be the stage itself?
- 62. L968. Not sure how you define maturity but I would be inclined to go with the time of lowest pressure, which is after landfall.
- 63. L969. The primary floods, that killed 5000+ people, in Derna were clearly on 11 Sep, not 10th.
- 64. L970-976. Very jumbled and somewhat unintelligible. Please rewrite.
- 65. L978. Change "relatively remotely" to "in regions that were quite remote from".
- 66. L980-981. Again sentence does not make sense and needs rewriting.
- 67. L982. Change "the predictability of the" to "predictions of".
- 68. L983. Why particularly the ECMWF EPS?! I guess that's not what you mean. Therefore please clarify.
- 69. L985. Occurrences or genesis? It may be the former, but I am just checking.
- 70. L986-988. Aside from the fact that this sentence needs re-ordering are you really sure that getting a modest cyclogenesis event correct was critical for getting the remote moist inflow from the Aegean, that drove the floods, correct too, given how far away that was?
- 71. L989-990. Once again there are errors in the manuscript timings, which for a reviewer is pretty frustrating! Daniel did not make landfall in Libya "within a few days". It was much sooner than that.
- 72. L990. Predictions of not the predictability of.
- 73. L992. "more prone to an erroneous predictability of" is again bad English. Please correct.
- 74. L993-994. How about "correctly predict its location" instead of "correctly predict correctly its evolution in terms of location"?
- 75. L998. Floods are not responsible for high discharges. It's the other way round.
- 76. L998. Largely??! Unprecedented means bigger than anything that happened before, not just in the last 20 years!
- 77. L999-1001. The discrepancy in resolution is pretty important here. This should be discussed/addressed.
- 78. L1001-1002. "exceptional potential for information to the public..". Bad English again.
- 79. L1002-1005. Gobbledegook. Besides which return periods are not exactly new.

- 80. L1006. "the grounds"? not sure. As stated above the main evidence seems to be SST and to me that's not much. And that's hardly rocket science. Then a slightly later comment about it being September seems to imply by chance rather than by climate change.
- 81. L1014-1017. Grandiose words but it is not clear to me that you have really done this in any convincing way. Once all the text gets tidied up I might be in a better position to judge. But in any case whilst the words "socio-economic impacts" are used a lot, there is little of substance behind them, it seems, so I remain sceptical.
- 82. L1017. Linking not bridging?
- 83. L1018. What does eventually mean here?

---

## Author Response (AR2)

**Reviewer 1**

This manuscript constitutes a multi-faceted case study for cyclone/medicane "Daniel" which, at different stages in its lifecycle, delivered extreme weather and devastating impacts in parts of both Greece and Libya, in early September 2023.

My reactions on reading the response to reviewers and the revised article were diametrically opposite, and very hard to reconcile.

Firstly, the response letters that directly referenced reviewers' comments were excellent, and covered almost every point that was made in an entirely satisfactory manner. Likewise, improvements made to the figures and captions, which was mainly what I commented on, were almost all comprehensive. The figure and caption quality, integrity and fitness for purpose have markedly improved as a result.

Secondly, I can note that, disappointingly, there was no direct response to my initial overview comments, which are quite fundamental, and then, in part because of this, the manuscript text remains in an unacceptable state. There are far too many grammatical and English errors, and there are also numerous contradictions, factual inaccuracies and unjustified sweeping statements. For some sentences I could not disentangle what the authors were trying to say. In the parts that I have gone through in detail – the Abstract, the Introduction and the Summary and Conclusions sections – I was encountering almost one error per sentence. I am sorry but it really is not the job of a reviewer to propose corrections at this level, so I confine myself to commenting on those 3 sections alone. This is time consuming enough! In the end it is the authors that should ultimately make the paper fit for purpose, not the reviewers. I can imagine that this may be a case of having "too many" authors without one electing to take full responsibility for making the final content coherent and consistent, but clearly I cannot know that for sure.

From an English standpoint I know that creating good sentence structures may be challenging for non-native speakers but I would advise that there are many tools out there that can help — e.g. Grammarly — and maybe the authors could benefit a lot from using such a tool to put together any re-submission?

Line numbers in the points below refer to the manuscript version which shows the corrections, not the cleaned version.

We thank the Reviewer for their comments and for the appreciation that the responses to the previous comments were excellent. All new queries have been addressed, and the text underwent substantial editing.

**Detailed Points**

1. L35. Daniel did not form as an intense cyclone. It formed as a weak cyclone that intensified markedly several days later as you highlight in the text.

Corrected.

2. L40. "conducted" is better than "aim to conduct" as otherwise it sounds like you haven't done the work.

**Corrected.**

3. L41. "Hazardous weather relevant to extreme precipitation". To my mind extreme precipitation is hazardous weather. Reorganise the sentence.

Cyclone systems can produce hazardous weather due to windstorms, storm surges, sea waves, hail, etc. This was relevant to precipitation, floods, and significant sea wave activity. We changed to "hazardous weather conditions related to extreme precipitation"

4. L44-46. Not sure what you are trying to say in this jumbled sentence. For example what do you really think are the implications of your findings for NWP? If that's a big part then surely the abstract should give the reader some clues as to what they are. Please re-write.

Changed to: "Given the climatologically extreme precipitation produced by Daniel, we examine the capacity of numerical weather prediction models to capture such extremes, and we finally investigate potential links to climate change."

5. L47. Gives the impression that this was rapid cyclogenesis from the outset, whereas it was anything but. Please rewrite.

This comment is not entirely clear to us. We reexamined the phrase in the abstract and believe our statement is correct. If any additional corrections may be necessary, we shall implement them as required.

6. L49. "Even in" is a bit odd. Implies that near to the centre there were big impacts, which I have not seen. "notably in the region of Thessaly which was actually quite far from the centre" would be much better wording to my mind.

Corrected to "At this stage, it produced significant socioeconomic impacts in Greece, i.e., in areas far from its center."

7. L49-50. "As it intensified...it peaked after landfall..." is bad English. It developed markedly just prior to landfall, reaching peak intensity over land.

**Corrected as suggested.**

8. L51-53. Sentence reorganisation needed – e.g. Considering short lead times (around four days), cyclone formation exhibited low predictability, whilst landfall in Libya was more predictable.

**Corrected as suggested.**

9. L53-55. This statement is not correct. You highlight a shortfall in precipitation totals for the Greece case; in fact with a lead time of 2 days – useful for forewarning - the observed gauge peak was 2.5 times the ECMWF HRES forecast total (see news item by Hewson in

https://www.ecmwf.int/sites/default/files/elibrary/12024/81535-newsletter-no-178-winter-2023 24.pdf). Your reference to a somewhat smaller but still notable shortfall corresponds to analyses which cannot generally be used to forewarn, and moreover I do not know if the values quoted were in the same location. So this doesn't really provide "crucial information on the expected severity", or at least not without much more work on cross-referencing model climates etc, which you don't discuss. Likewise, regarding the devastating floods in Derna it is very clear that for these we did not have "crucial information on the expected severity" of the riverflow (Figs 4b and 11) (never mind the dam collapse) as you acknowledge in the text.

Extremes of precipitation and river discharge are statistically treated from model outputs despite their bias from observations. We show that the extreme character of the event was fairly well captured in ECMWF NWP and GLOFAS. Consequently, we support the idea that information on the severity of the weather event could be deduced from the forecasts. We decided to leave the phrase as it is.

10. L58. Is the Mediterranean not one big maritime area?? What about the high uptake over land >50%, and indeed the seemingly the very high uptake over the Black Sea, for the Greece event. It looks to me like water vapour uptake over the Mediterranean was actually pretty small in relative terms — maybe ~ 25% of the total contribution (??) — which seems to contradict what you say?

This is correct that the Black Sea and continental Europe contribute significantly to the moisture sources. On 5 September, the Black Sea and continental Europe accounted for 75% of the moisture sources, and on 10 September for 40%. We revised to "Our findings indicate that large-scale atmospheric circulation was the primary driver, drawing substantial water vapor from the eastern Mediterranean, Black Seas and continental Europe."

11. L58-60. Poor English, please reword. Also what about the Black Sea, given point 10 above, as that was very warm too (Fig 5)?

Revised to: "The intensification of storm Daniel was likely driven by anomalously warm SST in the Mediterranean and Black Sea, enhancing evaporation and contributing to the extreme precipitation along the Lybian coast."

12. L59-61. This feels to me more like jumping on the climate change bandwagon rather than solid science. So far as I can see the main evidence for this conclusion is that SSTs in places were a bit higher than normal. Is that really enough evidence? SSTs were above average many times in the past.

The climate change attribution does not rely only on SST, thus the content of the two phrases the reviewer is referring to is not to be mixed. To clarify, we added "Combining multiple lines of evidence, as customary in attribution studies, we can deduce that..".

13. L95. Early September might be better

Corrected as suggested.

14. L97. "Named the upcoming storm" would be better. You can't, strictly speaking, name a storm when it doesn't exist. But I know what you mean.

Corrected as suggested.

15. L98. It did not evolve into a deep cyclone that propagated southwards. It propagated erratically southwards, then turned towards the east on 8th, and then developed into a deep cyclone. That is pretty clear on Fig 1a.

Corrected as suggested.

16. L100. Tautology. Delete "all attributed to the same weather system".

Corrected as suggested.

17. L104. Southeastern would be better. And it is misleading to link this statement to Fig 1b because that does not show any flooding where this rainfall was. Expand please to provide more clarity.

We deleted the reference to Fig. 1b.

18. L105. These are most definitely not eastern parts of Greece. They are about as central in Greece as one can get.

We revised to: "Thessaly experienced..."

19. L109, according to Wikipedia there are only three provinces in Libya. I think that what you are referring to are governorates.

Corrected as suggested.

20. L110. Again you point to part of Fig 1 as evidence of this flooding, yet Fig 1 seems to show no flooding whatsoever in the Derna area. Besides which the vast majority of the place labels on Figure 1 are too small to be legible. That also needs addressing.

We agree that Figure 1 does not visibly show flooding in the Derna area. This is due to the timing of the available satellite overpasses, which captured the area after the peak of the flash flood had already passed. As a result, the imagery reflects the post-event conditions rather than the flooding itself.

To improve clarity, we have updated Figure 1 to enhance the legibility of the place labels that are mentioned in the text, ensuring they are readable at the current scale.

21. L110-111. Building destruction does not include road damage. Separate.

We removed "including".

22. L112. I just wondered if there is anything more recent than the IOM, 2023 quote?

We changed to "Global Data Institute of the UN International Organization for Migration (IOM). 2023. Libya — Storm Daniel Flash update 8 (13 October 2023)" (see references).

23. L119. "72 million were requested". What units? And as with point 22, do you have any later evidence of what was delivered.

Updated to US dollars.

24. L127-129. For me this description is far too vague, having no geographical references whatsoever. Then the following sentence may need to be adjusted in some way if these geographical specifics are not always applicable.

We revised as follows: "Daniel was an intense cyclone, preceded by Rossby wave breaking over the Atlantic Ocean, which led to the formation of an omega blocking pattern (Couto et al., 2024) and the subsequent intrusion of an upper-level trough in the Mediterranean. This scenario is commonly observed before the formation of intense Mediterranean cyclones, including medicanes (Raveh-Rubin and Flaounas, 2017)."

25. L133. I think of an intrusion as a place where the stratospheric intrudes 'underneath' tropospheric air. Yet you have shown no evidence of that for this case. I can understand troughs and cut off lows being characterised by a lower tropopause, but to say more than that you have to present evidence in my view.

Any filament of stratospheric air extending southwards in a constant pressure level (or other vertical level metric) can be perceived as an intrusion. We understand that the Reviewer refers to tropopause folding, which is not the case here. We therefore chose to leave the text as is.

26. L134. My 'classical' interpretation of baroclinic instability is based on thermal gradients and geostrophically-related wind shear within the troposphere. Your usage seems to be much more focussed on upper level gradients. So maybe you can add a bit of clarification here? I also feel that "trigger" may be the wrong word here. I see the cyclone formation here as more of an event driven by ascent forced by advancing upper level structures, in the Pettersen Type B category, say, but in this case in the absence of any pronounced low level "Type C" in thermal gradient. So in the end а bit like this paper (https://rmets.onlinelibrary.wiley.com/doi/abs/10.1256/00359000260498806), albeit not at high latitudes and without the comma cloud.

Thank you for proposing this study. We changed the phrase according to the comment: "Such an anomaly forces ascent by advancing upper level PV structures."

27. L140. "...factors that modulate cyclone intensification..." would be better wording. Or drive instead of modulate.

Corrected as suggested.

28. L142. "in" not "by".

**Corrected as suggested.**

29. L144. Without any direct connection to the current work this sounds just like an advert for this paper. Why include this one in particular? I am sure there are other medicane studies that could be cited.

We are now more explicit on the reason for including this particular reference: "...., while a recent thorough analysis of the interplay and synergies between baroclinic and diabatic forcing of another intense cyclone in the central-eastern Mediterranean (lanos, 2020) is provided by Pantillon et al. (2024)."

30. L!50. Why do you imply that we have to have a moisture transport towards the Mediterranean when it is perfectly capable of providing its own! In fact this seems to contradict your conclusion that the high SST anomalies in the Mediterranean were somewhat fundamental.

We see no contradiction. There are moisture sources from the Mediterranean Sea and other regions, as is typically in several cyclone cases. We changed "essential to "contributes", we also added the reference of Flaounas et al (2019), who have shown moisture sources extending far from the Mediterranean Sea for a short climatology of intense Mediterranean cyclones producing heavy precipitation:

"Regardless of whether precipitation is stratiform or convective, the large-scale atmospheric circulation contributes by transporting water vapour toward the Mediterranean and thus "feeding" the cyclone-induced precipitation (Flaounas et al., 2019; Hochman et al., 2024)."

31. L149-156. The sentiments expressed here seem to be rather jumbled up and don't really make sense to me. You say the Mediterranean is enclosed by high mountains which tends to stop external moisture getting in, but then you say that moisture that does arrive is really important. And what about the Black Sea as mentioned above?

We changed "high mountains" to "continental areas". This should be clearer now: "Indeed, the Mediterranean basin is composed of a relatively closed sea surrounded by high mountains. Consequently, Mediterranean cyclones have fewer water sources than their counterparts in the storm tracks over the open oceans."

32. L157-159. How can quantifying the water sources help one to understand socio-economic impacts? I don't get this and I really don't think this point is addressed in the paper.

Thanks for noticing this. Indeed, the sentence was not clear enough. We therefore revised it as follows: "Hence, identifying and quantifying the contribution of water sources to heavy precipitation is a key step for improving our ability to forecast these events in the Mediterranean and anticipate their possible socio-economic impacts (Hochman et al., 2022a)".

33. L161-162. Jumbled English. "From a climatological standpoint cyclones are the weather features that lead to most of the wind and precipitation extremes within the Mediterranean" would be rather better.

**Corrected as suggested.**

34. L163. I don't recognise the word "compoundness". I have never seen it before. "in compound high impact weather events" would be much better in my view.

The term is used in the scientific literature. We therefore decided to leave it as is.

35. L164. It is clearly completely wrong to imply that a system has to make landfall to deliver storm surges and sig high waves.

We changed to: "... also considering that rather compact systems close to the coast additionally contribute to impacts with storm surges and significantly high waves (Patlakas et al., 2021; Ferrarin et al., 2023a; Ferrarin et al., 2023b)."

36. L167. What do you mean by PV streamers as a proxy for Rossby wave breaking? Sentence order could also be improved.

We removed PV streamers to avoid confusion: "Especially in the case of precipitation, recent results have shown that intense water vapour transport and Rossby-wave breaking are two of the main features that lead to extreme Mediterranean events (de Vries, 2021; Hochman et al., 2023)."

37. L169. How does intense water vapour transport favour development of a deeper cyclone. To me it seem that intense water vapour transport is more likely to be a consequence than a cause, given that winds increase as a low deepens.

It can be both. Abundant moisture may lead to more intense and longer-lasting convection, which will keep deepening the cyclone. We revised as follows: "Both of these large-scale atmospheric features favour the development of cyclones into deep, low-pressure systems: the former through baroclinic forcing and the latter through diabatic forcing by intensifying convection (e.g., Davolio et al., 2020)."

38. 170-171. I do not know what socio-economic impact on a weather scale is. Nor on a climate scale. This is very clunky wording. Moreover why do we need to understand both features to predict socio-economic impacts. The connection seems a very loose one to me. Surely it's much more important to understand societal vulnerabilities and how they relate to weather extremes if one wants to predict socio-economic impacts. This is like point 32.

Risk assessment (or alternatively the prediction of impacts) is a function of hazards (e.g. windstorms, floods), vulnerability, and exposure. These three components are to be considered independent and equally important. The accurate modelling of the hazard component is essential for the accurate estimation of impacts in both weather (forecasts) and climate (climate predictions) time scales. We leave as is.

39. L173. I can see how one can quantify past trends, but not future ones. One can provide estimates of what they might be, and these are likely to have large error bars associated.

Trends refer to tendencies in time series. A future climate simulation can serve this purpose.

40. L179. ...than would have been expected in... is better English.

Corrected as suggested.

41. L186-187. Woolly sentence that doesn't mean much to me.

Changed to: "First, understanding the event's dynamics and physical processes is crucial for assessing weather forecasting performance and climate change attribution."

42. L188. "concerning" means "according to"? And then what sort of "specific conditions" are you talking about? This is very vague.

Changed to: "Second, the associated hazards—such as floods and windstorms—must be assessed according to the specific weather conditions, as well as the vulnerability and exposure of the affected areas."

43. L189-191. Again a poorly worded sentence. I object very strongly to the terminology "to attribute its intensity to climate change". You should too. I hope this is clumsy wording because as it stands it makes it sound like you are looking for all manner of reasons to attribute an event to climate change, which is not scientific at all. And this is not the only place that this message is conveyed. Even the title of Section 5.2 is in this vein, which I find very worrying. These things really have to be toned down and corrected and made scientifically sound.

We thank the reviewer for this important comment and fully agree that attribution studies must be communicated with scientific precision and caution. We have carefully revised the wording throughout the manuscript to avoid any misleading or overly deterministic language regarding the role of climate change. In particular, we no longer use formulations such as "to attribute its intensity to climate change." Instead, we now consistently describe our aim as analyzing the potential contributions of both natural variability and human-induced climate change to the large-scale conditions that may have influenced the development of Storm Daniel.

We have also revised the title of Section 5.2 and several key sentences to ensure that our conclusions are appropriately cautious. Additional explanations have been included to clarify the scope and limitations of our analysis, emphasizing that our approach is exploratory and focuses on the likelihood of circulation patterns, not the deterministic attribution of specific storm features.

44. L193. What does "apply a comprehensive framework" mean? Then the idea that you are "using" Storm Daniel as a centrepiece sounds almost novel, when it is nothing of the kind. It

is crystal clear to any synoptician, and indeed from previous news items and publications, that this cyclone played a crucial role.

Changed to "Our motivation is thus to apply a comprehensive framework to provide an interdisciplinary assessment of the Storm Daniel event".

45. L194. I don't know why there is a reference at the end of this sentence when this is supposed to refer to what you set out to do.

We removed the reference.

46. L202. "...in relation to imminent hazards..." is again clumsy wording.

We removed "imminent hazards".

47. 209-210. "section 5 is devoted to Daniel's attribution to climate change" is again terrible wording in my view. I am sorry but using terminology like this provides wonderful ammunition for climate change deniers.

We agree that precise and careful wording is essential, especially when discussing attribution and its implications. We have revised the sentence in question to avoid any misleading phrasing and now write: "Section 5 discusses the potential contribution of natural variability and human-induced climate change to the characteristics and evolution of storm Daniel." We have similarly reworded and toned down many sentences related to attribution all over the manuscript.

48. Sections 2,3,4,5. Please revisit these very carefully, checking every word in every sentence. There are lots of errors similar to those highlighted above, and as stated earlier it is really not the role of a reviewer to correct all this. That is just too much.

Language has been improved.

49. L946. "Besides fatalities" would be a better start to the sentence.

Corrected as suggested.

50. L947. What does "the climate acts as a risk multiplier mean" please?

Indeed, this was not clear enough. We rephrased the sentence: "The IFRC World Disasters Report (2020) concluded that climate change serves as a risk multiplier, i.e., intensifying existing vulnerabilities, particularly in low-income countries".

51. L948-949. I don't see the point of saying "and at a regional level therein".

We removed this part of the phrase.

52. L949. "has been highlighted by the" is tautological. "is the" would be better.

Corrected as suggested.

53. L953. "by linking" not "which links".

Corrected as suggested.

54. L955. "atmospheric dynamics are used here to understand the performance of NWP..." is a curious statement. How have you understood the performance using atmospheric dynamics? Please elaborate.

This is done in subsequent paragraphs. We removed the phrase here to avoid raising earlier questions on the topic.

55. L956. Sea state is not an impact. Even for Libya. And "have also been analysed concerning" is bad English.

56. L957-960. Sentence makes no sense to me.

The text has been updated to: "Impacts—including flooding and coastal sea-state conditions in Libya—were also evaluated with numerical weather-prediction models. We placed these findings in a broader climatological context of cyclone-driven precipitation, underscoring how the observed impacts connect to climate-change attribution for both catastrophic events."

57. L962. "processes governing Daniel" is a strange phrase. Please rewrite.

Changed to: "... the processes governing Daniel's development were..."

58. L963. What do you mean by intrusion? Where is the evidence?

This is thoroughly discussed in the text, and Fig. 2 explicitly shows the PV streamer. We decided to leave the phrase as is.

59. L964-965. Again the idea that Daniel developed into a deep storm that propagated southwards is not correct. It is not what is shown on Fig. 1 and it is not what is written in the main text of the paper.

60. L965-966. More clumsy wording in the same sentence: "while it was turning into..."????

Corrected as earlier in comment 15.

61. L966-967. "Well-distinct" should be "distinct", then why is the first stage "relevant to cyclogenesis". Surely cyclogenesis should be the stage itself?

We removed "well". "Stages" refers to cyclogenesis and the phase with severe impacts of the cyclone, respectively. This latter part was kept as is.

62. L968. Not sure how you define maturity but I would be inclined to go with the time of lowest pressure, which is after landfall.

Changed as suggested.

63. L969. The primary floods, that killed 5000+ people, in Derna were clearly on 11 Sep, not 10th.

Changed as suggested.

64. L970-976. Very jumbled and somewhat unintelligible. Please rewrite.

Changed to "During both stages, Storm Daniel produced extreme precipitation by transporting moist air toward the flood-affected regions. The moisture transport followed the large-scale atmospheric circulation and drew on two main sources: an anomalously warm Mediterranean Sea and the continental areas of central and eastern Europe. Together, these reservoirs supplied the water vapor that fueled the catastrophic rainfall."

65. L978. Change "relatively remotely" to "in regions that were quite remote from".

66. L980-981. Again sentence does not make sense and needs rewriting.

Changed to: "In Greece, the floods occurred during the stage of cyclogenesis, in regions that were quite remote from the cyclone centre. On the other hand, floods in Libya occurred close to the cyclone centre and close to the stage of its maximum intensity."

67. L982. Change "the predictability of the" to "predictions of".

Changed as suggested.

68. L983. Why particularly the ECMWF EPS?! I guess that's not what you mean. Therefore please clarify.

In the paper, we used the ECMWF EPS, and this is what we comment on.

69. L985. Occurrences or genesis? It may be the former, but I am just checking.

"Occurrences" is more adequate here. Left as is.

70. L986-988. Aside from the fact that this sentence needs re-ordering are you really sure that getting a modest cyclogenesis event correct was critical for getting the remote moist inflow from the Aegean, that drove the floods, correct too, given how far away that was?

We do not discuss moisture sources here. We discussed in the text that the model showed higher confidence in the occurrence of extreme precipitation than in the exact location of cyclogenesis.

71. L989-990. Once again there are errors in the manuscript timings, which for a reviewer is pretty frustrating! Daniel did not make landfall in Libya "within a few days". It was much sooner than that.

We clarified "During its second stage (impacts in Libya), the cyclone transitioned into a medicane, making landfall in Libya within a few days after its formation."

72. L990. Predictions of not the predictability of.

Changed as suggested.

73 & 74. L992. "more prone to an erroneous predictability of" is again bad English. Please correct. L993-994. How about "correctly predict its location" instead of "correctly predict correctly its evolution in terms of location"?

We changed to: "These results indicate that numerical weather-prediction models are less skillful at predicting cyclogenesis; however, once the cyclone has formed, the models could become more reliable at forecasting its subsequent track."

75. L998. Floods are not responsible for high discharges. It's the other way round.

Changed as suggested.

76. L998. Largely??! Unprecedented means bigger than anything that happened before, not just in the last 20 years!

We removed "unprecedented".

77. L999-1001. The discrepancy in resolution is pretty important here. This should be discussed/addressed.

This has already been discussed in section 3. We prefer to keep the last section in the form of a summary.

78. L1001-1002. "exceptional potential for information to the public..". Bad English again.

Changed to "This underscores the exceptional potential to give the public timely, accurate warnings about the severity of impending high-impact weather events."

79. L1002-1005. Gobbledegook. Besides which return periods are not exactly new.

We removed this part because it is repetitive of the previous sentence.

80. L1006. "the grounds"? not sure. As stated above the main evidence seems to be SST and to me that's not much. And that's hardly rocket science. Then a slightly later comment about it being September seems to imply by chance rather than by climate change.

We changed "grounds" to "means".

81. L1014-1017. Grandiose words but it is not clear to me that you have really done this in any convincing way. Once all the text gets tidied up I might be in a better position to judge. But in any case whilst the words "socio-economic impacts" are used a lot, there is little of substance behind them, it seems, so I remain sceptical.

We believe this concluding sentence is just a summary of what has been shown in the paper, therefore we decided to leave it as it was.

82. L1017. Linking not bridging?

Corrected as suggested.

83. L1018. What does eventually mean here?

We removed "eventually".

**Reviewer 2**

Many thanks for carefully considering and addressing my comments. I would have been happy to recommend for this manuscript to be accepted in its current form, but there are (in my opinion) still outstanding issues with Section 5.2, "Attribution to climate change".

I appreciate the effort made by the authors in revising the section after the comments from me and the other reviewer, and I am sure that more improvements could be made fairly quickly by addressing minor issues such as:

We sincerely thank the reviewer for the constructive and detailed feedback. We have carefully considered each of the remaining concerns regarding Section 5.2 and have revised the manuscript accordingly. We have already changed the section title to "The Role of Natural Variability and Human-Driven Climate Change in Changing Daniel Dynamics" to reflect better that we investigate possible sources of changes in Daniel's dynamics, without being overly assertive about the role of climate change. Please find below our point-by-point responses:

- ensuring consistency by considering the same number of analogues throughout all the panels of each figure (instead of having, e.g., 30 for Fig12x vs 15 for all other panels of Fig12);

We confirm that all panels in Figure 12 (and Figure 13) use a total of 30 analogues. If this was unclear due to labeling or presentation, we have now made it explicit in the figure captions and clarified the text accordingly to avoid any confusion.

- removing or better motivating the analysis of slow modes of variability, as the only result stated, even in this revised version, is that they "may have influenced the development of the MSLP pattern associated with the storm", without providing any hypothesis on how that could have happened;

We agree that the original sentence was vague and insufficiently motivated. We have now clarified the rationale for including the analysis of slow modes of variability by explicitly stating our hypothesis: that multidecadal modes such as the Atlantic Multidecadal Oscillation (AMO) and the Pacific Decadal Oscillation (PDO) may influence the background state of the

atmosphere over the Mediterranean region. This, in turn, could modulate the occurrence and intensity of high-impact weather systems like Storm Daniel. We support this hypothesis by referencing recent literature highlighting teleconnections between these slow modes and Mediterranean climate variability.

- checking that all statements in the text are consistent with results shown in the figures (I can't see any precipitation increase over Albania in Fig12I);

Thank you for flagging this inconsistency. The statement about Albania has been revised to match the data shown in Fig. 12.

- explaining how frequency changes can be deduced if the same number of analogues is selected for each of the two periods;

We thank the reviewer for the comment and would like to clarify a misunderstanding regarding the analogue selection. The figure is based on the 30 best analogues selected over the entire period, not 15 analogues per sub-period. This means the analogues are drawn from historical and recent periods purely based on circulation similarity, without imposing a fixed number per time slice. This approach allows for an objective assessment of frequency changes: if circulation patterns like the one associated with storm Daniel have become more likely in the recent climate, we expect a disproportionate number of the top analogues to fall within the recent period. Conversely, if such configurations were more common in the past, the analogues would cluster in the earlier period. Thus, by keeping the total number of analogues fixed, we can infer frequency shifts from the distribution of analogues across time, without introducing biases from an arbitrary time-based quota. We have added: "The increase in the frequency of circulation analogues to storm Daniel in recent decades suggests that the synoptic conditions conducive to such extreme Mediterranean cyclones are becoming more common. This shift implies a heightened background risk for similar high-impact events under present-day climate conditions."

- revising the text to remove clumsy expressions such as "About impacts in Greece, we search...." (other poorly phrased sentences can be found in other sections of the paper, but on their own they wouldn't warrant further revisions as I'm sure they would be fixed during the typesetting/proofreading stage).

We thank the reviewer for highlighting this. We have conducted a careful language revision of Section 5.2 and corrected all awkward or imprecise formulations. In the case cited, we now say: "To assess the impacts in Greece, we investigate..."

Unfortunately, there are some more fundamental issues with the methodology, which in its current form does not look to be best suited for attributing the strength of Storm Daniel to climate change. The two main sticking points for me are:

- defining analogues using just one surface variable (MSLP) at a single level and at a single time (particularly as Daniel is a cyclone undergoing substantial changes in its three-dimensional structure during its lifecycle);

We agree this is a limitation of the current analogue-based framework. As we clarified in the revised text, the goal of our approach is not to capture the full three-dimensional structure of the storm (as would be required in a full dynamical attribution), but rather to identify comparable large-scale circulation patterns. This provides a statistical context for understanding whether such circulation patterns—and the associated impacts—are becoming more common under climate change. We now emphasize this distinction clearly in the text and highlight that our method complements, rather than replaces, high-resolution model-based attribution studies. The analogue selection based on MSLP is a deliberate simplification, enabling rapid assessments across multiple events.

- using only a 40-year dataset and, as the authors acknowledged, with limitations on small scales (which are key to the extreme impacts and to the very nature of the event under study, as Daniel was a compact Mediterranean cyclone, later becoming a Medicane, and not a synoptic-scale mid-latitude cyclone).

Given the above, it's hard to see how the results shown in Section 5.2 justify the claim that the authors "provided the grounds to interpret Daniel as an event whose characteristics can be ascribed to human-driven climate change". The authors state (referring to part of the analysis in the section) that "this assessment is exploratory, highlighting potential associations without making definitive attributions". Considering the section as a whole, I acknowledge the merits of this exploratory analysis, but I am not convinced it is, in its current state, publishable.

We fully agree that Daniel's small-scale and Medicane-like nature challenges any attribution approach relying on coarser reanalyses. This is now explicitly acknowledged in the revised text. We stress that our results do not attempt to reconstruct the mesoscale structure of Daniel, but rather assess whether circulation patterns associated with such compact events are more likely in the current climate. We also include a cautionary note that small-scale features and rapid intensification, which are critical to Daniel's severity, may be underrepresented in our framework, and hence our attribution remains exploratory.

Having said that, I really don't want to sound antagonistic and to be a stumbling block preventing the acceptance of this overall very good manuscript. If the editor does not share my judgement on the issues of Section 5.2 and thinks that it is indeed worth publishing as it is, I would be happy to see the manuscript accepted without further revision.

We thank the reviewer for this constructive stance. We hope that the revisions we have implemented address the concerns raised to a satisfactory degree, while maintaining the clarity and usefulness of the ClimaMeter methodology for transparent and reproducible assessments.

Dear authors,

Many thanks for the detailed and thorough revision of the manuscript, addressing all the reviewers' concerns. I would like to also thank here each of the reviewers for their detailed and constructive reviews of both versions of the manuscript.

The manuscript integrates together innovative and relevant aspects of storm Daniel and its impacts in Greece and Libya and considers them together using an impressive combination of tools and approaches. Importantly, the paper also places Daniel in a climate context, referring to its extremeness by considering the variability in the observed period and using an attribution tool based on analogues.

As you can see from the reviewers' reports, both acknowledge the improvements made, but are still critical about the results concerning the climate change attribution, and/or the way it is conveyed in the text. I share this concern, please see below.

Therefore, this outstanding issue should be addressed before the manuscript can be accepted for publication. Given the limited scope of the analogue chapter in the current manuscript, I suggest that if there is evidence to support the main conclusions it should be shown directly in the manuscript (e.g., relation of precipitation to SST, or SST and humidity signals in the analogue analysis). Alternatively, the interpretation and conclusions should be significantly reduced and toned down. Additionally, the analogue analysis can be better framed as a way to detect the exceptionality of the event and its hazards and understand possible long-term changes of similar cases in the observed period. The framing of the analysis as attribution to anthropogenic climate change can be too confusing or even misleading or misused.

When revising the manuscript, please take the opportunity to further improve the text of the manuscript following the suggestions by the reviewer, also to reflect better the answers to reviewers in the main text.

Many thanks again and best wishes, Shira
* * *
(line numbers refer to the new clean version):

Framing of climate change attribution:

The framing of the analysis as attribution to anthropogenic climate change is not explicit. This aspect is not mentioned in the Methods section, but only in the Results (line 677: "attributing observed shifts to anthropogenic climate change"). This attribution aspect can only be inferred indirectly after reading the last paragraph in the Methods Section 2.2.3 noting that the modes of climate variability are considered together as the "natural variability". However, it is conceptually unclear if the remaining variability is assumed to be attributed to anthropogenic forcing? To me this is not trivial. There can be other factors at play, as well as large case-to-case variability that influence the two samples of analogues. Therefore, the rationale behind the analogue analysis should be more clearly framed. At its

current form, the concluding statement in lines 804-5: "we ... provided the grounds to interpret Daniel as an event whose characteristics can be ascribed to human-driven climate change" – is not backed by the methodology and results and should be removed.

Climate modes of variability (ENSO, AMO, PDO):

The results for both stages show that the analogue difference signals can be attributed to significantly different modes of AMO and PDO. Following on the comment above – this means that the signals are not necessarily attributed to climate warming already given the current analysis. This aspect remains quite fuzzy and not explained further, as noted also by the reviewer.

Rather than using the analysis for climate change attribution, I find more value in using the analogue approach for learning if/how exceptional Daniel was in its various aspects (dynamics, hazards...). In this context it will be useful to add a dot to mark the position of Daniel in the distribution of the mode states in panels u,v,w in both figures.

Results concerning impact in Greece and Libya and relation to SST:

In line 728 it is stated that "the increase in precipitation over the region is most likely linked to higher sea surface temperatures (SSTs), which provide more moisture to the atmosphere". I find this conclusive statement to be somewhat detached from the results shown. First, there is clear decrease in precipitation over Greece that emerges from the analogue results, but a (more moderate) increase over the Ionian Sea (but not Albania as stated), so the results are not simply "increase of precipitation in the region". Second, the relation to SST and humidity content is not shown. Instead, Fig. 12 presents 2-m temperature and winds, when SST and humidity fields would have been better fields to examine for the purpose.

Similarly, lines 735-6 state that "increase in rainfall over Libya was also likely driven by warmer SSTs and a warmer atmosphere, which can hold more water (Clausius-Clapeyron relationship) rather than a shift in atmospheric dynamics patterns". Here, there is indeed increase in precipitation, but the only results pertaining to SST (Fig. 5b) show a mixed picture of negative SST anomalies in the Ionian Sea on 9 Sep as a result of the cyclone passage in the previous days, with moderate positive anomalies closer to northeastern Libya. In fact, 2-m temperature there shows a negative anomaly, contradicting the statement in the text. Therefore, the connection to SST is not substantiated by the results but rather speculated or based on literature on other medicanes and global aspects of climate change. As such, the thermodynamical climate considerations are not shown and cannot establish the main conclusion (as also stated in the conclusions, lines 807-8).

In fact, looking closely at Fig. 12m, it is possible that the anomalously high wind intensities above the Black Sea (along with high SST anomalies there) are related to this region serving as an important moisture source for precipitation (Fig. 3). However, this aspect is not discussed in the manuscript.

**Additional typos:**

Fig. 6: please clarify in the caption that there are no data for the Black Sea.

Lines 305, 580: 5 -> 6 September

Line 433: 2025 -> 2023

Line 1482: summer -> autumn
Correct normal script to superscripts in all units

**Answer**

Dear Shira,

We thank you sincerely for your careful and thoughtful editorial comments, and for acknowledging the overall value of our manuscript and the improvements made. We are especially grateful to both reviewers for their thorough and constructive evaluations, and for their critical insights which have helped us to considerably improve the quality and clarity of our work.

We have carefully revised the manuscript in line with your suggestions and those of the reviewers (including the typos). Below we summarise the main revisions and how we have addressed the outstanding concerns:

1. Framing of climate change attribution We agree that the framing of the climate change attribution was insufficiently clear in the previous version. We have now revised the Methods section (Section 2.2.3) to explicitly define how we interpret natural variability and to clarify that our analogue-based approach does not attempt to separate anthropogenic forcing as a residual, but instead investigates shifts in the frequency of circulation analogues. We have removed the concluding statement in lines 804-805 (see authors' track changes), and we now clearly frame our results as exploratory evidence of possible long-term changes in the frequency of analogue situations.

Additionally, we have revised the title of Section 5.2 to: "The Role of Natural Variability and Long-term Trends in the Context of Storm Daniel", and throughout the section we have adopted more cautious language. We no longer refer to direct attribution to anthropogenic climate change, but instead focus on the statistical detection of shifts in analogue frequency and the broader context of thermodynamic and dynamical changes.

**2. Use of ENSO, AMO, PDO modes:** We now clearly state in Section 5.2 that the different distributions of analogues across phases of AMO and PDO highlight that these slow modes may modulate the occurrence of Daniel-like configurations. We do not interpret these results as anthropogenic attribution per se, but as part of the natural variability context. We have added brief explanations of the potential teleconnection mechanisms and cited relevant studies to support this framing.

We considered the suggestion to include a dot indicating the position of storm Daniel in Figures 12 and 13. However, we ultimately chose not to implement this modification. Adding such a marker would not significantly enhance the interpretation of the results, given that the rarity and distribution of the event are already clearly discussed in the text. Moreover, since the figures summarize climatological distributions rather than case-specific diagnostics, including an event-specific marker risks over-interpreting a single data point within a broader statistical framework. While this addition offers a useful visual cue regarding the rarity of the event under the observed modes, we note that th underlying data and interpretations remain unchanged; the information added is therefore illustrative rather than substantive. This

provides the reader with a clearer understanding of the rarity of the event under the observed modes.

**3. Relationship to SST and precipitation** We have toned down statements in lines 728 and 735-736 regarding the linkage between SST and precipitation. We now clarify that the increase in precipitation over Libya (but not over Greece) may be partly related to warmer SSTs and a warmer atmosphere, but we no longer assert this as a primary conclusion. Instead, we explicitly acknowledge the limitations of our dataset (2-m temperature rather than SST or humidity fields) and highlight the need for further analysis.

We have also revised the text to reflect the mixed signal in the SST anomaly fields and clarified that while Clausius-Clapeyron scaling provides a general theoretical framework, our analysis does not directly quantify these thermodynamic effects.

The potential contribution of the Black Sea as a moisture source is now explicitly discussed, in line with Fig. 3 and Fig. 12m.

---

## Author Response (AR3)

**Authors' Reply**

We would like to thank Shira Raveh-Rubin for guiding our work through the peer-reviewed process. All comments have been addressed in the revised version.